# Global IWV trends and variability in atmospheric reanalyses and GPS observations

Ana C. Parracho[1, 2], Olivier Bock[1], Sophie Bastin[2]

[1]IGN LAREG, Université Paris Diderot, Sorbonne Paris Cité, Paris, 75013, France,
[2]LATMOS/IPSL, UVSQ, Université Paris-Saclay, Sorbonne Universités, UPMC, Univ., Paris 06, CNRS, Guyancourt, France

*Correspondence to*: Ana C. Parracho (ana.parracho@etu.upmc.fr)

**Abstract.**

This study investigates the means, variability, and trends in Integrated Water Vapour (IWV) from two modern reanalyses
(ERA-Interim and MERRA-2) from 1980 to 2016 and ground-based GPS data from 1995 to 2010. It is found that the mean distributions and inter-annual variability in IWV in the reanalyses and GPS are consistent, even in regions of strong gradients. Inter-annual variability is dominated by ENSO with variations as large as 20% IWV in the tropics and the mid to high northern latitudes in winter. ERA-Interim is shown to exhibit a slight moist bias in the extra-tropics and a slight dry bias in the tropics (both in the order of 0.5 to 1 kg m$^{-2}$) compared to GPS. ERA-Interim is also generally drier than MERRA-2 over the ocean
and within the tropics. Differences in variability and trends are pointed out at a few GPS sites, which might be due to representativeness errors, for sites located in coastal regions and regions of complex topography, gaps and inhomogeneities in the GPS series, due to equipment changes, and potential inhomogeneities in the reanalyses, due to observing system changes. Trends in IWV and surface temperature in ERA-Interim and MERRA-2 are shown to be consistent, with positive IWV trends generally correlated with surface warming, but MERRA-2 presents a more general global moistening trend compared to ERA-
Interim. Inconsistent trends are found between the two reanalyses over Antarctica and most of the southern hemisphere, and over central and northern Africa. The uncertainty in current reanalyses remains quite high in these regions where few in-situ observations are available and the spread between models is generally important. Interannual and decadal variations in IWV are also shown to be strongly linked with variations in the atmospheric circulation, especially in arid regions, such as North Africa and Western Australia, which add uncertainty in the trend estimates over the shorter period. In these regions, the
Clausius-Clapeyron scaling ratio is found not to be a good humidity proxy for interannual variability and decadal trends.

## 1 Introduction

Water vapour is a key component of the Earth's atmosphere and plays a key role in the planet's energy balance. It is the major greenhouse gas in the atmosphere and accounts for about 75 % of the total greenhouse effect globally (Kondratev, 1972). The
total amount of water vapour is mainly controlled by temperature, closely following the Clausius-Clapeyron (C-C) equation

(Schneider et al., 2010). Water vapour is thus an important part of the response of the climate system to external forcing, constituting a positive feedback in global warming (Held and Soden, 2006). However, at a regional scale, deviations from C-C law are observed and the strength of the feedback can vary, also because the radiative effect of absorption by water vapour is sensitive to the fractional change in water vapour, not to the absolute change (O'Gorman and Muller, 2010).

The spatial structure of global IWV variability at seasonal and longer time scales evidences patterns that result from interactions between the atmospheric circulation and the land and ocean surfaces and are dominated by El Ninõ Southern Oscillation (ENSO) (Trenberth et al., 2005).

Although El Niño events are associated with increasing temperatures in the eastern and central Pacific with impact on the global weather and climate, it is not well known if global warming will lead to more frequent or intense El Niño events (Collins

et al., 2010). Another strong cause of variability in the northern hemisphere is the North Atlantic Oscillation (NAO). A high positive NAO is associated with warmer winters in the Eurasian landmass, due to the stronger westerly and south-westerly airflow that brings in warmer maritime air. However, it is not clear how the phase or intensity of NAO has been, or will be, affected by climate change (Visbeck et al., 2001).

All these parameters, and the fact that the time of residence of water vapour in the atmosphere is short, make IWV a highly

variable component and its study in terms of variability and trends is rather challenging. Sherwood et al., 2010, compared the long-term IWV trends reported in several studies using different datasets. Although there appears to be a global positive trend in the overall IWV data, which is consistent with a global warming trend, it is difficult to compare results from different studies, as they refer to different data sources, time periods and different sites and spatial coverage. Trenberth et al. (2005) found major problems in the means, variability, and trends from 1988 to 2001 for the National Centers for Environmental

Prediction (NCEP) reanalyses 1 and 2, and for the 40-year European Centre for Medium-Range Weather Forecasts (ECMWF) reanalysis (ERA-40) over the oceans. The reanalyses showed reasonable results over land where they are constrained by radiosonde observations. Only the reprocessed IWV data from the special sensor microwave imager (SSM/I) appeared to be realistic in terms of means, variability, and trends over the oceans. Their work points to two important issues. First, the reanalyses generally lack assimilation of water vapour information and suffer from model biases and, in the case of ERA40,

problems in bias corrections with new satellites (namely after major volcano eruptions). Second, they highlight the need for the reprocessing of data, and point to the shortcomings in reanalyses due to the changing observing system. The bias correction of new satellite radiances in the ECMWF reanalysis system has recently been improved using a variational bias-correction scheme, including the detection of instrument calibration errors and long term drifts, as well as volcano eruptions (Dee and Uppala, 2009). Reduction of model biases and enhanced assimilation capabilities of satellite data (e.g. rain-affected radiances)

have generally improved the water cycle in modern reanalyses. Reanalyses data agree thus generally well at representing the short-term variability (e.g. ENSO) but their ability for detecting climate trends is still debated (Dessler and Davis, 2010; Thorne and Vose, 2010; Trenberth et al., 2011; Robertson et al., 2014; Schröder et al., 2016).

In this study we will focus on analysing the mean distributions, inter-annual variability, and decadal trends from two recent reanalyses, ECMWF reanalysis ERA-Interim (Dee et al., 2011), referred to as ERAI, and the 2nd Modern-Era Retrospective

analysis for Research and Applications, MERRA-2 (Gelaro et al., 2017). MERRA-2 benefits from recent developments in NASA's Goddard Earth Observing System (GEOS) model suite intended to address the impact of the changes in observing system (Gelaro et al., 2017). As a result, atmospheric water balance and variability in MERRA-2 are more realistic, though variations of IWV with temperature are weaker in the main satellite data reanalyses (namely ERA-Interim and MERRA-2)

compared to microwave satellite observations over the oceans (Bosilovich et al., 2017). Here, the IWV contents of the two reanalyses are intercompared and compared to a global, homogeneously reprocessed Global Positioning System (GPS) dataset over ocean and land. The ground-based GPS observations are independent from the reanalyses as they are so far not assimilated and constitute a valuable validation data for atmospheric reanalyses (Bock et al., 2007) and satellite data (Mears et al., 2015). A critical assessment of the homogeneity of GPS dataset itself is provided throughout the study as previous work detected

small offsets associated to GPS equipment changes (Vey et al., 2009; Ning et al., 2016).

Furthermore, to add new insights in both the evaluation of ERA-interim and MERRA-2 reanalyses and in the understanding of IWV trends and variability, we separate the analysis into seasons, and consider trends and interannual variability of seasons. This analysis by seasons is rarely provided in other studies, although it helps to better identify regions with higher uncertainty and to understand the physical processes involved in different seasons (e.g. the dynamical component which transports

moisture strongly differs between winter and summer). Trenberth et al. (2011) separated January and July in their analysis of the representation of water and energy budget in ERA-I and MERRA and showed the importance to study the seasons separately. Compared to this study, we added the analysis of the GPS dataset and use the new version of MERRA.

This paper is organized as follows. Section 2 details the datasets and methods used. Section 3 reports on the means and variability found in the GPS and reanalyses data, for the 1995-2010 period. Section 4 focuses on the monthly and seasonal

trends in GPS and ERA-Interim, for 1995-2010. In Section 5 we confront results of ERA-Interim and GPS to MERRA-2. In this section, the comparison between ERA-Interim and MERRA-2 was also extended to the 1980-2016 period and focused on two regions of intense trends: western Australia and north Africa/eastern Sahel. Section 6 summarizes and concludes the paper.

## 2 Datasets and methods

### 2.1 Reanalysis data

Reanalysis data from the ECMWF, ERA-Interim (Dee et al., 2011), and NASA, MERRA-2 (Gelaro et al., 2017), were extracted for the 1980-2016 period, on regular latitude-longitude grids, at their highest horizontal resolution (0.75° x 0.75° for ERA-Interim and 0.625° longitude x 0.5° latitude for MERRA-2). In this work, the two-dimensional (2D) distribution of IWV is investigated with reanalysis fields and with point observations from 104 GPS stations of the International GNSS (Global Navigation Satellite System) Service (IGS) network (Fig. 1). Because GPS heights and surface heights in the reanalyses are

not perfectly matched (see the GPS coordinates and ERA-Interim heights in the supplement Table S1), the IWV estimates were adjusted for the height differences using two different methods. In the 2D maps (e.g. Fig. 2), the monthly mean GPS IWV estimates were height corrected to match the nearest reanalysis grid point, while for the computation of IWV differences

(e.g. Fig. 4), a more elaborate interpolation method was used (described below). For the monthly mean IWV correction, specific humidity from the ERA-Interim pressure level data were integrated over the layer of atmosphere bounded by the model's surface height and the height of the GPS station. The ERA-Interim pressure level data contain a total of 37 levels between 1000 and 1 hPa, among which 27 levels lie between 1000 and 100 hPa. This ensures a good vertical sampling of the

5    troposphere where most of the water vapour is located. Note that for the sake of consistency, the same pressure-level data (i.e. ERA-Interim) are used for correcting the MERRA-2 monthly mean IWV data shown in the maps.

In the case of ERA-Interim, the height differences between GPS stations and nearest model grid points range from -1457 m (at the SANT (Santiago, Chile) station) to +3167 m (at the MKEA (Mauna Kea, Hawaii) station). The negative height difference means GPS height is below the model surface. The mean IWV corrections for these two stations amount to -3.4

10    kg.m$^{-2}$ and 21.7 kg.m$^{-2}$, respectively. Globally, inter-quartile range of the corrections is [-1.40, 0.39] kg.m$^{-2}$.

A more rigorous approach is adopted for the quantitative evaluation of the reanalysis IWV data with respect to GPS IWV data, in order to minimize temporal and spatial sampling issues. In this case, we time-matched the 6-hourly data and performed a spatial interpolation of the reanalysis IWV estimates to the latitude and longitude of the GPS site. A bilinear spatial interpolation is computed from the model IWV estimates at the 4 grid points surrounding each GPS station. The IWV model

estimates are then recomputed from the pressure level data by vertically integrating the specific humidity between the height of the GPS station and the top of the atmosphere. Most GPS station heights fall between two pressure levels and the specific humidity data can be interpolated. However, for stations located below 1000 hPa (the lowest pressure level) the reanalysis data are extrapolated. Interpolation and extrapolation are done linearly for specific humidity and temperature, and exponentially for pressure. This procedure minimizes differences between the reanalysis IWV data and the GPS estimates with better results

than previous correction methods (e.g. Bock et al., 2014). However, a perfect match between observations and model data is hindered by representativeness errors (Lorenc, 1986), especially in mountainous and coastal regions.

## 2.2 GPS data

The reprocessed GPS data set used in this work was produced by the NASA Jet Propulsion Laboratory (JPL) in 2010-2011. Basic details on the operational GPS data processing procedure are described by Byun and Bar-Server (2009). Compared to

the operational version at the time, the reprocessed data set was produced with more recent observation models (e.g. mapping functions, absolute antenna models) and consistently reprocessed satellite orbits and clocks (IGSMAIL-6298). Inspection of file headers revealed that the processing options were not updated for a small number of stations for a period of nearly one year between March 2008 and March 2009. The comparison of solutions with old and new processing options (available for year 2007) showed that this inconsistency in the processing has negligible impact at most stations, except for stations at high

southern latitudes (e.g. in Antarctica). The data set covers the period from January 1995 to December 2010 for 456 stations. Among these, 120 stations have time series with only small gaps over the 15-year period. However, the geographical distribution is quite unequal between hemispheres and even within a given hemisphere, with namely a cluster of 20 stations in the western USA with inter-station distance smaller than 0.75°. In order to avoid over-representation of this region, 16 out of

these 20 stations have been discarded (the selection retained those with the longer time series). The final GPS IWV dataset used in this study is thus limited to the selected 104 stations.

The basic GPS observables in this study are the Zenith Tropospheric Delay (ZTD) estimates available at a 5 minute temporal resolution. The ZTD data were screened using an adaptation of the methods described by Bock et al. (2014) and Bock et al. (2016). First, we applied a range check on the ZTD and formal error values using fixed thresholds representing the spatial and temporal range of expected values: 1 – 3 m for ZTD and 0 – 6 mm for formal errors. Second, we applied an outlier check based on site-specific thresholds. For ZTD, values outside the median ± 0.5 m are rejected and, for formal errors, values larger than 2.5 times the median are rejected. The median ZTD and formal error values are updated yearly. Using these thresholds, we detected no ZTD values outside the limits. This is because the limits were sufficiently large to accommodate for the natural variability of ZTD values (Bock et al., 2014). On the other hand, the formal error check rejected $8.8 \times 10^{-4}$ (i.e. less than 0.1 %) of the data overall. After screening, the 5-minute GPS ZTD data were averaged in 1-hourly bins.

The conversion of GPS ZTD to IWV was done using the following formula: *IWV = ZWD × κ(Tm)*. Where $\kappa$(Tm) is a function of weighted mean temperature Tm, and ZWD is the zenith wet delay, obtained from: *ZWD = ZTD – ZHD* and ZHD is the hydrostatic zenith delay (see Wang et al. (2005) or Bock et al. (2007) for further details). In this work, the surface pressure used to compute ZHD and the temperature and humidity profiles necessary to obtain Tm were obtained from ERA-Interim pressure level data. The profile variables are first interpolated or extrapolated to the height of the GPS stations at the 4 surrounding grid points and then interpolated bi-linearly to the latitude and longitude of the GPS stations. At this stage, the GPS and ERA-Interim data were time-matched (within ±1 hour) for both the ZTD to IWV conversion and IWV intercomparison.

Afterwards, monthly means of the 6-hourly IWV estimates are computed and those months which have less than 60 values (i.e. at least half of the expected monthly values) are rejected. Seasonal means are computed from the monthly values when at least 2 out of 3 months are available. These selection criteria ensure that the computed values are representative of the monthly and seasonal means.

In this work, inhomogeneities in the GPS IWV time series due to equipment changes were not corrected a priori, as the existing metadata may not be complete, but were rather detected and discussed during the course of the intercomparison with ERA-Interim. This work is a preliminary contribution to a more extensive detection and correction effort of the GPS IWV data.

## 2.3 Computation of trends

The linear trends were computed using the Theil-Sen method (Theil, 1950 and Sen, 1968), a non-parametric statistic that computes the median slope of all pairwise combinations of points. This method is described in more detail in the Appendix to this paper, where it is also compared with another commonly-used method for trend estimation, the Least Squares method.

The Theil-Sen method was applied to the anomalies obtained by removing the monthly climatology from the monthly data. In the case of seasonal trends, the mean anomalies for the months of December, January and February (DJF); and June, July and August (JJA) were used (when there are at least two months of data available per season, per year). The statistical significance

of the monthly and seasonal trends was assessed using a modified Mann-Kendall trend test (Hamed and Rao, 1998), which is suitable for autocorrelated data, at a 10 % significance level.

## 3 Means and variability in GPS and reanalyses IWV (1995-2010)

The reanalyses and GPS data have been used to investigate the mean seasonal IWV distribution and its interannual variability for December-January-February (DJF) and June-July-August (JJA).

In the maps of the means (Figs. 2a, b), we can see that ERA-Interim reproduces the spatial variability well compared to GPS, including the sharper gradients in IWV, for instance, on the northern and southern flanks of the Intertropical Convergence Zone (ITCZ) in both seasons, and in the regions of steep orography (for example, along the Andes region, in South America). For the analysis of the interannual variability we computed the relative standard deviation of the seasonal IWV time series (i.e. standard deviation of seasonal time series divided by its mean value). The relative variability emphasizes both regions where the variability is high compared to the mean IWV (e.g. the tropics) and regions where the mean IWV contents are small (e.g. cold dry polar and/or mountainous regions and warm dry desert areas). In DJF (Fig. 2c), strong interannual variability (> 15 %) is found for northern high-latitude regions (north-eastern Canada and eastern Greenland, polar Artic area, and a large part of Russia and north-eastern Asia) and for the tropical arid regions (Sahara, Arabic peninsula, central Australia). Some correlation is found between the seasonal IWV anomalies and the North Atlantic Oscillation (NAO) index (Barnston and Livezey, 1987) (not shown) over Siberia (r = 0.5) and Greenland (r = -0.5). Noticeable variability is also seen in the central tropical Pacific in DJF but this is due to the extremely large variability in absolute IWV contents (up to 6 kg.m$^{-2}$) associated with the ENSO. Linear correlation coefficients between the seasonal IWV anomalies and the Multivariate ENSO Index (MEI; Wolter and Timlin, 1993, 1998) in this region reach r = 0.80 (not shown). In JJA, large interannual variability is observed mainly over Antarctica and Australia (Fig. 2d). Locally enhanced variability is also seen over the Andes cordillera, but this is mainly due to the very low IWV values at high altitudes.

Similar mean patterns are observed in MERRA-2 for both seasons, although maximum values over the ITCZ have different intensities. In order to better gauge the differences, mean difference fields between MERRA-2 and ERA-Interim are shown (Figs. 3a and b). It is observed that ERA-Interim is generally drier than MERRA-2 over the ocean and in the regions of maximum IWV, and moister in the southern part of South America, north and south of the ITCZ over Africa, and in southern (DJF) and central (JJA) Asia.

In terms of interannual variability, for DJF, there are similar maxima of variability in ERA-Interim and MERRA-2 over the Arctic, Tropical Pacific, and Siberia and West Africa (to a lesser extent). Over Australia and Antarctica, the variability is lower in MERRA-2 than in ERA-Interim, while over Canada the variability is higher. The difference fields between MERRA-2 and ERA-Interim highlight these differences (Fig. 3c). Although some of the areas of interest are not covered by the long-term GPS observations, in general, there is better agreement between GPS and ERA-Interim than MERRA-2, especially at higher latitudes. For JJA (Fig. 3d), the intense variability in Australia and the Andes seen in ERA-Interim (and GPS) is not as intense in MERRA-2, while a stronger variability is observed for MERRA-2 over Antarctica.

The mean IWV differences are shown in Figs. 4a and b for all 104 GPS sites. It can be noticed that the negative differences (ERA-Interim IWV lower than GPS IWV) are almost all within the ITCZ, with few exceptions in north-western North America and in Antarctica. This result is confirmed when ERA-Interim is compared to MERRA-2 and consistent with differences between ERA-Interim and MERRA reported by Trenberth et al. 2011. A paired two-sample t-test detected 20 stations with

5 significant differences in the mean IWV values at 0.01 confidence level in DJF and 17 in JJA (the values for all stations can be found in Supplement Table S2). The sites with most notable differences, either absolute (in kg.m$^{-2}$) or relative (in %) are: CFAG in the Andes cordillera with a bias of 6.5 kg.m$^{-2}$ (26 %) in DJF and 3.9 kg.m$^{-2}$ (43 %) in JJA and, SANT in Chile with -2.4 kg.m$^{-2}$ (-15 %) in DJF, and TSKB (in Japan) with 1.9 kg.m$^{-2}$ (24 %) in DJF. In JJA, four other sites have large biases: KIT3 in Uzbekistan with a value of 6.2 kg.m$^{-2}$ (35 %), POL2 in Kirghizstan with 3.1 kg.m$^{-2}$ (20 %), SYOG in Antarctica with

10 0.6 kg.m$^{-2}$ (32 %), and MAW1 in Antarctica with 0.4 kg.m$^{-2}$ (31 %). The inspection of the time series shows that at some of these stations the biases are not constant in time but contain large seasonal variations, such as e.g. at CFAG (Fig. 5a) or KIT3 (Fig. 5b). These sites are located in coastal regions and/or regions with complex topography. Although we used an elaborate spatial and temporal matching of reanalysis and GPS data, representativeness errors are suspected to be the cause of these biases. To investigate this point, we compared the (vertically adjusted) IWV values from the 4 grid points surrounding each

15 GPS station to the interpolated IWV values (see Section 2). We found that at CFAG, KIT3, POL2, SYOG, and MAW1, the interpolated values did not minimize the IWV biases between the reanalysis and GPS. This is explained by large variations in the altitude of the grid points at these sites (between 500m and 1000m) and the difficulty for the vertical interpolation method to properly predict the IWV variations over such large altitude ranges. In the case of SANT, although the interpolated value matches the GPS value better than any of the four surrounding grid point values, there is still a large bias explained by a

20 variation in the altitude of the grid points of over 1500m. Statistics computed over all stations are given in Table 1. They indicate that ERA-Interim is slightly moister on average than GPS. The median bias is 0.51 kg m$^{-2}$ (6.2 %) in DJF and 0.52 kg.m$^{-2}$ (2.7 %) in JJA, and the standard deviation of the bias across the network amounts to 0.83 kg.m$^{-2}$ (6.9 %) in DJF and 0.95 kg.m$^{-2}$ (7.8 %) in JJA. As noticed above, there is some spatial variation in the mean difference, namely a negative mean difference in the tropics (ERA-Interim < GPS) which is compensated in the global median by the larger number of stations in

the extra-tropics which have a positive difference (ERA-Interim > GPS).

Most of the marked regional features of interannual variability are also confirmed by GPS observations (Figs. 2c, d). One can especially notice the good representation of the relative variability over Australia or South America, both in DJF and JJA, and in the northern high latitudes, where the gradients are strong and well captured. However, a few stations show different values compared to the ERA-Interim background. Figures 4c and d show the differences of relative standard deviations. The overall

statistics are given in the second part of Table 1. They indicate a median difference close to zero for both DJF and JJA with a standard deviation across the stations of 1.7 % in DJF and 4.1 % in JJA. We used a two-sample F-test to detect the stations where the variances differ significantly. However, this test detected only one result with a p-value < 0.01 (station MCM4 in JJA) and two with a p-value < 0.10 (MCM4 and CFAG in JJA). This statistical test is probably not very efficient in the case of our short time series (≤16 points). In JJA, the four stations with the largest differences (ERAI – GPS) are located in

Antarctica: MCM4, SYOG, MAW1, and DAV1 with differences of -39 % (p=0), -7.7 % (p=0.63), -4.8 % (p=0.81), and +3.9 % (p=0.27), respectively. In DJF, the largest differences are found for MKEA (Hawaii) and SYOG, where they amount to -11.4 % (p=0.52) and -4.8 % (p=0.30), respectively. Table S2 in the supplement provides the results for all stations. In the case of SYOG, MAW1, and DAV1, representativeness errors are suspected again because of the large variability in the IWV values of the surrounding grid points connected with large variations in the altitudes (> 500m) of these grid points. In the case of MKEA, the variation in the altitude of the surrounding grid points is quite small because of the limited imprint of Mauna Kea Island on the 0.75° resolution grid of ERA-Interim. However, the difference in altitude between the GPS station and all four grid points is larger than 3000 m which is far beyond the prediction capability of the interpolation method described in Section 2. In the case of MCM4 and SYOG, the inspection of the time series of monthly mean IWV and IWV differences (shown in Figs. 5c and d) reveals variations in the means which coincide with GPS equipment changes and processing changes and unexplained variations in the amplitude of the seasonal cycle resulting in a marked oscillation in the monthly mean differences (ERAI – GPS). Variations in the means introduce a spurious component of variability in the GPS IWV series (e.g. at MCM4 the standard deviation of GPS IWV is 0.78 kg m$^{-2}$ compared to 0.21 kg m$^{-2}$ for ERAI).

Three possible causes for the differences in the IWV means and variability between GPS and ERAI exist. As already discussed above, representativeness differences are expected in regions of complex terrain where the environmental conditions can differ. Strong horizontal gradients in IWV are a limitation for the bi-linear horizontal interpolation that we used. This kind of situation is generally encountered when the altitudes of the grid points surrounding the stations are very different (e.g. AREQ, SANT, KIT3, MAW1, SYOG, POL2). This problem is enhanced when the altitude of the GPS station is below the model surface (e.g. SANT, AREQ, KIT3, MAW1, SYOG), because the model profile data are extrapolated below the ground, and/or the model and GPS surface altitudes are very different (e.g. MKEA). Representativeness errors due to large spatial variations in IWV and altitude are expected at 20 stations among which are those cited just above. However, they don't explain all the significant biases and differences in variability actually observed. The second aspect is connected with errors in the GPS data, e.g. due to instrumental malfunctioning or measurement interferences, or changes in equipment resulting in variations in the mean IWV estimates. Such problems can be detected by comparison with IWV measurements from nearby GPS receivers or from other collocated instruments such as DORIS or VLBI (Bock et al., 2014; Ning et al., 2016). The third cause stems from errors in the reanalysis IWV data which are expected in data-sparse regions and regions where the performance of model physics and dynamics are poor. These can be diagnosed by comparing several reanalyses based on different models and different observational data or hypothesized by eliminating the other causes.

## 4 Trends in GPS and ERA-Interim IWV (1995-2010)

Trends from ERA-Interim based on the time series of monthly means (hereafter, referred to as monthly trends) are shown in Fig. 6. Significant positive trends (moistening) are observed over most of the tropical oceans and over the Arctic and significant negative (drying) trends are observed in south-tropical eastern Pacific region, west of the United States and generally south of 60 °S. The dipole structure in the south-eastern tropical Pacific area is consistent with the results of Mieruch et al (2014) and

is due to the different ENSO phases for this time period, as explained by Trenberth et al (2005). Over land, significant positive trends are observed in the equatorial region along the ITCZ, especially in northern South America, Central Africa, and Indonesia, and in the northern hemisphere, especially over northern North America, Greenland, most of Europe and Siberia. Significant negative trends over land are observed over North Africa, Australia, Antarctica, central Asia, south of South America, and most of the USA. In general, there is continuity between oceanic and continental trends (e.g. North and South America, Central Africa), suggesting a trend in air mass advections. However, the magnitudes of the larger trends (e.g. -3.5 kg m$^{-2}$ per decade or -17 % per decade over northern Africa) are questionable. To be physically explained such trends would imply a significant change in the regional and global water cycle. Alternatively, they might be due to inhomogenities in the observations assimilated in the reanalysis system. Comparison to GPS observations, when they are available, helps to address this question.

In general, the monthly trends computed at the GPS stations are consistent in sign and magnitude with ERA-Interim (Fig. 6). Many stations are operated in Europe and North America. Most of them show fairly consistent trends with ERA-Interim even in areas of marked gradients (e.g. between western Canada and the USA, or from central to western Europe). Australia is also well sampled with several stations, in the centre and along the coasts, and both the sign and spatial variations of trends are consistent (e.g. the eastern Australia's moistening trend). Many isolated stations in other regions confirm the ERA-Interim trends. Moistening trends are observed by stations (indicated in Fig.1) KOUR and BRAZ (northern part of South America), HRAO (South Africa), IISC (India), KELY (Greenland), DGAR (in the centre of the Indian ocean), FALE (in the Pacific Ocean), CRO1 (Puerto Rico), MAS1 (Canary Islands) and REYK (Iceland). In terms of drying trends, ERA-Interim and GPS trends are largely in agreement over the west coast of the United States, the southern half of South America (including the Andes region, which has steep IWV gradients) and the western half of Australia. It is also noteworthy that BRMU (in Bermuda) has a drying trend that is also captured in the ERA-Interim data. The results for all stations are given in the Supplement Table S3.

Inspection of Fig. 6 shows that there are a number of GPS stations where the trend estimates are large and of opposite sign compared to ERA-Interim: CCJM (south of the Japanese home islands), DARW (northern Australia), WUHN (eastern China), IRKT (central Russia), ANKR (Turkey), KOKB and MKEA (Hawaii), and MCM4 (Antarctica). Some of them (DARW, ANKR, KOKB, MKEA) are located in areas where the ERA-Interim trends change sign and a perfect spatial coincidence between the reanalysis and observations might not be expected. On the other hand, stations CCJM, WUHN, IRKT, and MCM4 are located within regions where the ERA-Interim trends are strong and significant, and extend over large areas. For some of these stations, the discrepancy is due to gaps and/or inhomogeneities in the GPS time series which corrupt the trend estimates. To mitigate the impact of gaps, time-matched series are also compared (Figs. 7a and b show the trend differences). The agreement is improved at DARW, ANKR and IRKT, and at many other sites (e.g. KELY in Greenland, SANT, MAW1). However, there are still many sites with large differences. Table 2 lists the stations with the largest differences. Inspection of time series reveals the presence of large inhomogeneities at CCJM, MCM4 (already discussed in the previous section, see Fig. 5c, WUHN, SHAO, and CRO1. At CCJM (see Fig. 7a), the GPS minus ERA-Interim IWV difference time series has a large

offset in 2001 which coincides with a GPS equipment change (receiver and antenna). This offset is responsible for a large negative trend estimate in the GPS series (-1.40 kg m$^{-2}$ per decade) whereas the time-matched ERA-Interim series gives a positive trend (+0.98 kg m$^{-2}$ per decade) consistent with the large-scale trend in the reanalysis seen in Fig. 6a. At WUHN (Fig. 8b), the GPS trend estimate is positive (0.34 kg m$^{-2}$ per decade) while the ERA-Interim estimate is negative (-1.45 kg m$^{-2}$ per

decade). The IWV difference time series shows several breaks (in 1999, 2005 and at the end of 2006) though none of them coincides with known GPS equipment changes. In fact, the break at the end of 2006 is associated with the change in radiosonde from the Shang-M to Shang-E, which is assimilated by ERA-Interim (Wang and Zhang, 2008). Zhao et al (2012) found that prior to this change there was a 2 kg.m$^{-2}$ wet bias in the radiosonde data at the Wuhan station, in comparison with GPS. This moist bias is also observed in ERA-Interim prior to the end of 2006. This case will be further discussed below when

trends from MERRA-2 reanalysis are analysed (Section 5). At SHAO and CRO1, and a few other sites (e.g. SYOG, DARW, ANKR) inhomogeneity in the IWV difference series coincide with documented GPS equipment changes (not shown). Representativeness differences are suspected at some mountainous and coastal sites (e.g. AREQ, CFAG, KIT3, MAW1, SANT, SYOG and the other sites discussed in the previous section). Some sites show also more gradual drifts in the times series which don't seem connected with known GPS equipment changes (e.g. MAW1, Antarctica). At such sites, drifts in the

reanalysis are plausible and will be further discussed below when MERRA-2 results are analysed.

Wang et al. (2016) studied nearly the same period (1995-2011) using radiosonde and GPS data over land and microwave (MWR) satellite data over oceans. Over the oceans, results that are significant in ERA-Interim are consistent (i.e. same sign) with those obtained by Wang et al. (2016), despite the fact that they are not always significant over land in the latter study. No results are obtained over most of Africa and the north-western part of South America due to a lack of data. The drying for

north-eastern Africa and moistening over central Africa and north-western South America are therefore not confirmed by the Wang et al. (2016) study. For the other continental areas with weaker trends, results are not always in agreement, for instance over central Asia, where a moistening trend is generally observed in Wang et al. (2016). The western part of USA presents a strong spatial variability in both studies but results are generally not consistent locally. Greenland trends also present opposite signs. When comparing with the GPS results obtained by Wang et al. (2016), there is a general sign agreement, with some

different sign trends in central Australia (ALIC station) and Iceland (REYK). These differences may be due to the extra year in their analysis (as differences in the beginning and ending of time series have an impact on the trend estimation, especially when trends are of low intensity and not significant, and the period at study is relatively short).

Although the study does not concern the same period, Trenberth et al (2005) reported similar trend signs to ERA-interim over Africa and South America in the NVAP data (1988-2001) and positive trends over western Pacific, the Indian and Atlantic

oceans with SSM /I data (1988-2003). As discussed above, differences are observed over Eastern Pacific where El Niño events strongly affect the trend estimates. Note also a difference in the sign of the trend over Australia (an area which will be discussed later). Wagner et al. (2006) studied the IWV trends in satellite observations from the Global Ozone Monitoring Experiment for the 1996-2002 period. Although their study period is short, they also found positive IWV trends over the western tropical Pacific Ocean and large parts of the southern oceans, and negative trends over North Africa. Over northern Australia, they

found a negative trend, which is in agreement with what we obtain but not with Trenberth et al. (2005). This area is thus likely sensitive to the period at stake. The western part of the USA is also an area where differences between the studies are present, but it seems that spatial variability is strong and thus results strongly depend on the resolution of the datasets, and not only on the period. Thus, despite the different periods and the use of different observing systems, some areas show consistent trend signs with ERA-Interim which indicates that the results are likely robust. However, the trends obtained in our study can differ from those presented by other authors for other periods, as the trend estimation is dependent on the time period at study.

To better understand the trends, we separated them by seasons (DJF or JJA), which are presented in Figs. 9a and b, respectively. A striking feature of the seasonal trends is their relatively larger magnitude compared to the monthly trends. Large changes in magnitude and/or sign are also noticeable in most regions. These features emphasize that atmospheric circulation (which is largely changing between seasons) plays an important role in IWV trends. Trends of opposite signs between winter and summer can be observed in western Antarctica, central South America, south Africa, eastern Europe and off the West coast of the USA. A strong drying occurs over Antarctica in JJA and over central Asia during JJA and DJF (though not exactly at the same location). Western Europe shows a drying in winter (DJF) and a moistening in summer, which explains the weak trend when considering the whole year. Over Australia, according to ERA-Interim, the drying is stronger in DJF, i.e. when associated with a decrease of the intensity of the moist flow during the monsoon period. The differences between our study and the one of Trenberth et al. (2005) are consistent with the theory that precipitation over western and northern Australia (the part of Australia mostly influenced by the monsoon flow in DJF) are strongly sensitive to the SST over the western central Pacific Ocean (10 °S-10 °N; 150 °E-200 °E) (Brown et al., 2016). In ERA-Interim and satellite observations by Wang et al. (2016), during 1995-2010, the SST over this part of ocean has increased (indicated by a moistening, according to C-C law), and is associated with a drying over Australia, while during 1988-2001, a strong drying is observed over the central western Pacific Ocean, associated with a moistening over Australia. Another area likely sensitive to the intensity of the monsoon flow is northern Africa, where the drying is occurring in JJA over eastern Sahel, in a band covering Chad, Sudan and Eritrea.

Overall, the seasonal trends estimated from the GPS data confirm the features discussed above for ERA-Interim. The sites with largest differences in the seasonal estimates are also listed in Table 2. In addition to the sites where issues were noticed in the monthly trends, the list includes a few more sites which also show up in Figs. 9a and b (most notably KIRU in Sweden, COCO in the Indian Ocean, IRKT in Russia, and ANKR in Turkey). Trend estimates at some of these sites might be inaccurate due to the enhanced impact of time gaps for the short seasonal time series (based on 16 years at best).

## 5 Trends in ERA-Interim and MERRA-2

### 5.1 Global analysis

In this section, MERRA-2 is inspected to complement ERA-Interim and GPS namely in regions of high uncertainty in these datasets (e.g. Antarctica) or in regions where few or no GPS data are available (e.g. Africa, Asia, the global oceans). The monthly IWV trends computed for MERRA-2 (Figs. 10a, b) show many of the same drying and moistening regions as ERA-

Interim (Figs. 6a, b). They describe consistent global moistening/drying dipoles along the inter-tropical Pacific Ocean, across Australia, South America and between eastern and western USA, and general moistening over the Arctic and Europe. However, there appears to be also significant differences over several parts of the globe, in particular over Indonesia and the Indian Ocean, central Africa, western (coastal) and northern Africa, central Asia and Antarctica.

Over Antarctica, the monthly trends in MERRA-2 (Figs. 10a, b) are significantly positive, in opposition to what is seen in ERA-Interim (Figs. 6a, b) where the trends are mainly negative, especially in the interior of the continent. However, one can notice that ERA-Interim shows spotted areas of positive trends in the vicinity of the GPS stations which are in reasonable agreement with MERRA-2 and GPS. These locally positive trends in ERA-Interim might be explained by the influence of surface and/or upper air observations collected from these sites that are assimilated in this reanalysis. Comparing ERA-Interim

and MERRA-2 IWV time series in the interior of the continent reveals that the reanalyses diverge mainly before year 2000, with a positive trend in MERRA-2 between 1995 and 2000 (not shown). This divergence might be explained by a combination of differences in the observations actually assimilated and differences in the assimilation systems. Observations in the interior of the continent are most likely from satellites only. General documentation indicates that both reanalyses use the same types of satellite observations globally (Dee et al., 2011; Gelaro et al., 2017). However, it is not said whether or not these data are

actually assimilated over Antarctica. This kind of information can only be checked from assimilation feedback statistics.

Over Indonesia and the Maritime Continent ERA-Interim trends are positive while MERRA-2 trends are negative. Comparing IWV time series in the central part of this region reveals that though they are well correlated (r = 0.89), MERRA-2 shows a larger seasonal cycle than ERA-Interim (not shown). GPS observations are only available at the outer bound of the domain where both reanalyses are in better agreement with each other and also with GPS except for station CCJM where GPS has a

large discontinuity. Another exception is station GUAM (14 °N, 145 °E), where GPS is in agreement with ERA-Interim while MERRA-2 has opposite trends. ERA-Interim and MERRA-2 trends also disagree in several places in central Asia, namely in China where GPS data are in better agreement with MERRA-2 (stations WUHN and SHAO), and in northern and central Africa (but no GPS data are available there). The moistening trend over northern South America is seen in both reanalyses but is less intense in MERRA-2. However, GPS trends at stations KOUR and BRAZ are in better agreement with ERA-Interim.

The seasonal trends computed from MERRA-2 (Figs. 10c, d) are in better agreement, in terms of moistening/ drying patterns, with ERA-Interim and GPS in DJF than in JJA. In DJF, the east-west dipole in the trends over Antarctica seen in ERA-Interim is confirmed by MERRA-2 (though the intensities are different), as well as the strong drying trend over Siberia, the Arabic peninsula, western Australia, western Europe, and most of the USA; and the strong moistening over the Arctic. In JJA, on the other hand, we can find many differences between the two reanalyses which were already noticed in the monthly trends.

Opposite trends are seen in Indonesia and in most of south Asia, north and central Africa, Antarctica, but also in the eastern Arctic region.

Over Antarctica, the seasonal trends at the GPS stations are more consistent with MERRA-2 than with ERA-Interim in both seasons (Figs. 10c, d), though the GPS trends have a positive bias because of a processing inhomogeneity already mentioned. Interestingly, in JJA the contrasted trends with different signs seen in the GPS data (positive around SYOG, MAW1 and

MCM4, and negative around DAV1 and CAS1) are well reproduced by MERRA-2, whereas the ERA-Interim trends are negative throughout.

Over Europe and Middle East, the contrasted trends between seasons seen in ERA-Interim (Figs. 6c, d) are confirmed with MERRA-2 (Figs. 10c, d) and are in accordance with GPS data at many sites, except at a few sites where GPS trends are opposite to both reanalyses, mainly: at KIRU (Sweden), IRKT (Russia), ANKR (Turkey). Opposite trends are also observed at other GPS sites over Scandinavia, namely at METS and SVTL (Finland) and ONSA (Sweden).

Over northern Africa, the two reanalyses show similar trend patterns in DJF, though not perfectly collocated (e.g. a tongue of negative trend is extending across western Sahel in ERA-Interim whereas in MERRA-2 it is more limited to the western countries: Senegal, Mauritania, and western Mali). In JJA, on the other hand, the strong drying already highlighted in ERA-Interim which extends over most of north Africa is almost absent in MERRA-2 where most of northern Africa is seen as moistening. This striking difference emphasizes the uncertainty of reanalyses in this data sparse region as also noticed by Bauer, 2009, and Karbou et al., 2010.

Interpretation of IWV trends must be tempered by the fact that the time series used here are relatively short. Indeed, Trenberth et al. (2005) argued that the dominance of the 1997-98 El Niño event suggests that a longer time series may be required to obtain fully stable patterns of linear trends. The number of years needed to obtain a statistically significant trend in IWV in some regions, given its high variability, may never be achieved. In order to assess how consistent our trends obtained for the 1995-2010 period (when GPS data are available) are with longer-term trends, we computed them for the full length common to ERA-Interim and MERRA-2 (1980-2016).

On the monthly trends, most structures in ERA-Interim are similar for the short (Fig. 6a) and the long (Fig. 11a) period, although the intensities are weaker for the longer period (note that the colour bars are different for Figs. 6 and 10), but most of them are significant. Over land, the drying and moistening trends over Africa and South America show similar patterns, as well as the moistening trends over eastern and northern regions of Europe and the drying trends over Antarctica. The main differences appear over the Arabic Peninsula, Western Australia, Mexico, and a small part of Antarctica. The drying trend over Australia observed for the shorter period is not observed in the long term. For this longer period, trends are mostly not statistically significant, which suggests that there might have been a moistening trend before the drying trend. Over the oceans, an overall moistening trend (except a strong drying off the coast of Antarctica) is observed, especially in the northern hemisphere, but several areas show different patterns for both periods. For the Atlantic Ocean, a different sign is observed along the eastern coast of North America, with a significant moistening for the longer period, while a drying is confirmed by GPS around Bermuda for the shorter period. In the south, the drying trend is spatially more extended and statistically more significant for the longer period. Over the Indian ocean, for the short period, the western part moistens and the eastern part dries, and opposite trends are obtained over the longer period. Over the Pacific Ocean, even though the patterns look similar, the spatial variability is stronger for the shorter period, with a more intense moistening along the equator, and west of Patagonia and a weaker moistening around Alaska.

Comparing the seasonal trends in ERA-Interim for both periods, the JJA patterns are mostly consistent over land and ocean between the two periods (Figs. 9b and 12g). Slight differences appear over India (where the moistening trend is more spatially extended in the longer period), Australia (where the trend is no longer significant), and Antarctica, where the drying trend is shifted eastward. For DJF (Figs. 9a and 12c), stronger differences exist. While the moistening trend of the short period over

northern South America, southern part of Africa, Central and northern Europe, western Canada and Alaska and Artic are consistent with the longer period, the ones over Patagonia, part of China and Afghanistan, part of Antarctica and western Africa are no longer visible. The drying trends over Antarctica are extended to the entire continent for the longer period. The eastern USA that dries between 1995 and 2010 presents a moistening trend when considering the longer period. The strong drying obtained over Australia in DJF is mostly cancelled over the long period. Over the oceans, differences exist over the

Indian Ocean, western Atlantic (along the east coast of USA), part of the south Atlantic and Pacific and mostly around Antarctica.

According to the Clausius-Clapeyron (C-C) equation, it is expected that an increasing temperature trend corresponds to an increasing IWV trend, especially over the oceans where the source of humidity is infinite. In order to assess the link between temperature and IWV trends, the trends in the 2-meter temperature were computed (even if the use of 2-m temperature may

not be the best proxy of temperature in C-C equation). Monthly and globally (not shown), over the oceans, the temperature and water vapour trends have the same sign, despite some small-scale differences. Over land, all areas show an increase in T2m, except the high latitudes of the southern hemisphere. This means that, except over Antarctica, the drying observed in the afore-mentioned areas does not follow Clausius-Clapeyron relation. However, when we consider each season separately, some areas indicate a cooling (Figs. 12a, e) consistent with a drying (Figs. 12c, g). This is observed over Antarctica and to a lesser

extent over Central Asia in DJF. Over eastern Australia, and South Africa, however, a weak cooling is observed while a significant moistening has been computed. For JJA, all continental areas show a significant warming, with the exception of parts of Antarctica, and a small area over northern Australia, where a cooling is also displayed, albeit not significant. Thus the C-C scaling ratio is not a good proxy for humidity when considering seasonal and regional variabilities and trends due to the important role of dynamics which allow the advection of dry or wet air masses (e.g. over USA, South America, eastern Sahel,

and South Africa in JJA).

Trends in MERRA-2 over the long period (Figs. 12d, h) present different trends from ERA-Interim (Figs. 12c, g) over some areas as seen previously for the short period. They result from both the uncertainties that exist when computing trends, and from the differences in the physics and dynamics of the two reanalyses.

It is evident from Figs. 11 and 12 that MERRA-2 presents a more general moistening trend than ERA-Interim, especially in

the southern hemisphere in DJF (Figs. 12c, d), and in both hemispheres in JJA (Figs. 12g, h). The main differences in the trends over oceans appear all around Antarctica, and those over continental areas are observed over Africa (where trends are positive in the North and negative in central Africa in MERRA-2 and the opposite in ERA-Interim) and USA in JJA, over Australia in DJF and over Antarctica in both JJA and DJF. Over Africa and Antarctica, the important differences which exist between ERA-Interim and MERRA2 for both long and short term periods suggest that the physical processes are not well

represented. These areas correspond to areas with very few observations available for data assimilation, reducing the constraint on the models. A more detailed investigation of the dynamics over Africa and Australia is presented in the following subsection.

Other regions, such as the Indo-Pacific region have different trends over the shorter period, but are in better agreement over the longer period. This is more obvious during JJA (although there are also differences in DJF) and can be explained by the strong variability that requires longer time series in order to obtain meaningful trends. The good agreement between reanalyses over this area is an important result regarding the fact that CMIP5 models have large biases over this region in present day Sea Surface Temperature, which has direct consequences on the future projection of precipitation over Australia (Brown et al., 2016; Grose et al., 2014). However, the link between IWV trends over these oceans and Australia is not that strong here, since over Australia, while reanalyses were in good agreement over the shorter period, the western part presents a significant moistening in MERRA-2 over the long period in DJF, and a weak and not statistically significant drying in ERA-Interim. This area is thus investigated in more details in the next subsection. This may suggest discontinuities in the reanalysis data (due to changes in the data assimilated) or an uncertainty in the computation of long-term trends (due to the presence of different sign shorter term trends during the longer period).

## 5.2 Analysis over Western Australia

Figure 13 displays the time series of IWV and temperature anomalies for a box over Western Australia (15-30 °S, 115-135 °E, as shown in Fig. 14) for both the short and long periods, for both the full time series and the DJF seasons.

For the 1995-2010 period, both reanalyses show drying and warming monthly trends (Fig. 14b), with significant IWV trend in both while the temperature trend is only significant in ERA-Interim. For the longer time period (Fig. 14a), on the contrary, the monthly IWV trends are positive (moistening) for both reanalyses on average over the box, but not significant for ERA-Interim, and the temperature trends are again positive but smaller, and again significant for ERA-Interim. For DJF, the trends are generally consistent with the monthly trends, but of larger magnitude (e.g. IWV trends about -2.4 kg m$^{-2}$ decade$^{-1}$ for the short period) though not statistically significant. It is noticeable that the difference in the IWV trends between reanalyses comes from the fact that ERA-Interim IWV starts with higher anomalies than MERRA-2 until 1990, but ends with lower anomalies after the late 2000s, so that the resulting trend is close to zero and not significant.

What is striking when looking at the full time series (Fig. 13a) is the existence of extreme cold and humid periods in both reanalyses after 1992, with a strong occurrence around the 2000s, which impact the linear trend estimate over the short period more strongly than over the long period. These periods correspond to DJF seasons 1997, 1999, 2000, 2001, 2006, and 2011 (Fig. 13c). Power et al. (1998) and Hendon et al. (2007) have shown that during DJF the correlation between wetter years and colder years is strong at interannual time scales. For the longer period, the correlation between T and IWV is around r = -0.55 for both reanalyses, while for the shorter period it is higher at r = -0.78 for ERA-Interim and r = -0.73 for MERRA-2. However, there is also a more complex interaction between temperature, IWV and precipitation in this region, with studies over Australia concluding that dynamics mostly explain the variability and trend of temperature and precipitation.

Consequently, we consider the wind at 925 hPa to assess the role of dynamics in these trends and variability. Figure 14 presents the mean zonal ($u_{925}$) and meridional ($v_{925}$) components of the wind, superposed by the trends of each component in contours. The mean states in $u_{925}$ and $v_{925}$ are similar in both reanalyses. The zonal components show mainly an easterly wind in the latitude band between 30 °S and 5 °S throughout the year (Figs. 14a, b), slightly reduced at its northern border during DJF where the wind turns westerly (Figs. 14 e, f). The mean meridional component is southerly over Australia (Figs. 14c, d) with a changing direction at its northern border during DJF (Figs. 14 g, h) leading to a convergence within the box in DJF. The convergence is roughly from north-east in the northern part of the box and from south-east in the southern part.

The trends show a reinforcement of the mean easterly component in the box in both reanalyses (Figs. 14a, b). In DJF, the easterly flow in ERA-Interim is increasing at the eastern border of the box and over central Australia while it is decreasing at the western border and over eastern Indian Ocean, hence explaining at least partly the drying and warming trend over the box (Fig. 13d). The trends in the mean meridional component are positive though quite weak in both reanalyses (Figs. 14c, d), thus indicating a strengthening of the mean southerly flow. During DJF, the meridional trend is very small in ERA-Interim while it is slightly positive again in MERRA-2.

Figure 15 displays the time series of wind vectors over the same box as Fig. 14. Figure 15a shows that the mean wind direction does not change much over the year, though the strength is slightly larger in the summer season (from January to March) due to an increase in the easterly component. The interannual variability is quite marked, both in DJF (Fig. 15b) and JJA (Fig. 15c). It is clear that the anomalously moister summers seen in Fig. 13 are associated with a dynamical anomaly, with a weaker wind, and a direction switching from south-easterly to easterly. The amplitude of wind direction difference is stronger in MERRA-2 than in ERA-Interim but both reanalyses are consistent.

## 5.3 Analysis over North Africa/eastern Sahel

Here we focus on a box over the eastern Sahel (10-20 °N, 10-40 °E). The monthly trend in IWV is negative (drying) and significant in both reanalyses, though it is twice as intense in ERA-Interim than in MERRA-2 (Fig. 16b). Similarly, the temperature trends are positive (warming) and significant in both reanalyses. Over the long period, the IWV trend in MERRA-2 is close to zero and not significant while that of ERA-Interim is still significantly negative, while the temperature trends are again positive (Fig. 16a). Though the monthly anomalies show many similarities, their agreement is not as good as seen for the box over Australia. The general strong negative IWV trend in ERA-Interim implies that IWV anomalies are higher in ERA-Interim at the beginning of the period and lower at the end of the period. However, both reanalyses present four different periods in the time IWV series: a drying trend at the very beginning (1980-1985) followed by a moistening trend until 1995, then followed by a new drying period lasting until around 2008 when the trend seems to stop. As a consequence, over the shorter period, both reanalyses show a significant monthly drying, even though for ERA-Interim the IWV trend is twice as intense (Fig. 16b).

As observed for IWV anomalies, the trend in T anomalies stops at around 2008 (Fig. 16a). Before that period, the temperature anomaly is increasing significantly, despite strong month-to-month variability. However, low correlation appears between

IWV and T anomalies when considering full time series (Fig. 16a). For the longer period, the correlation between T and IWV is close to zero for MERRA-2 and about r = -0.31 for ERA-Interim, while for the shorter period it is higher at r = -0.41 for ERA-Interim and r = -0.29 for MERRA-2. In JJA, the trends are strong and go on after 2008 (Fig. 16c). The correlation of anomalies for JJA between both reanalyses is quite good, both for IWV (around r = 0.67 for the short period and r = 0.63 for the longer period) and T (around r = 0.69 for both periods), although their amplitudes and trends are quite different. MERRA-2 presents an overall moistening trend in JJA over the long period, while ERA-Interim shows a drying (Fig. 16c). IWV trends in both reanalyses are significant but of opposite signs, while the temperature trends are both positive and significant. Over the short period (Fig. 16d), the IWV trend in MERRA-2 becomes close to zero while the other trends remain consistent with the longer period. Over the short period, the IWV trend in JJA in MERRA-2 is close to zero while it is still strongly negative in ERA-Interim (Fig. 16d). The relation between IWV and temperature trends in this arid region is not expected to follow C-C relationship and IWV especially is expected to be more related to changes in the atmospheric circulation.

The zonal and meridional wind components at 925hPa over the short period are shown in Fig. 17. The mean states are plotted in colours over which the contours of the trends are superposed. The mean states in u925 and v925 are similar in both reanalyses, with a mean monthly north-easterly wind over the box (Figs. 17a, b, c, d) which is almost completely replaced with a south-westerly wind in JJA (Figs. 17e, f, g, h). This wind is slightly stronger in ERA-Interim than in MERRA-2. For both reanalyses, the trends in the mean flow indicate an increase in the zonal component (Figs. 17a, b). The trends in the meridional wind component show a dominant increase in the northerly from the Sahara. This trend may explain the general warming and drying in the eastern Sahel. The trends differ, however, with MERRA-2 showing a decrease in the northerly flow in upper-left angle of the box (Fig. 17d) while ERA-Interim shows an increase there and an increasing southerly inflow at the southern border of the box (Fig. 17c). This difference can explain the difference of intensity in these trends. In JJA, the trends in MERRA-2 are very weak (Figs. 17f, h) while in ERA-Interim there is a strong increasing of the southerly flow from the Central Africa and of the north-easterly flow from the Sahara, explaining the net drying and warming (Fig. 16d).

The monthly mean time series of the wind in the box (Fig. 18a) clearly indicates the time of the monsoon onset (May), when the wind shifts from north-easterly to south-westerly and retreat (September). This monsoon flow appears stronger in ERA-Interim than in MERRA-2, while the flow in the dry season is stronger in MERRA-2 (see also Figs. 18b and c). From the time series of JJA wind vectors (Fig. 18b) it is clear that ERA-Interim has a stronger southerly flow with large interannual and decadal variability. The time series of wind in JJA in MERRA-2 clearly indicates the same four periods than for the IWV trends identified above, with a weakening of the south-westerly wind between 1980 and 1985, followed by an intensification of the monsoon flow arriving in this box between 1985 and 1995, and a wind decreasing and turning to the west until 2005 or 2006 and then becoming more stable on average. In ERA-Interim, we only observe two main periods: a weaker south/south-westerly wind at the beginning of the period followed by an intensification after 1990. The wind intensity is maximum between 1995 and 2000 but stays quite intense and with a south/south-westerly direction until the end of the period, being stronger and more southerly than in MERRA-2 after 2000. The different dynamics of the two reanalyses observed in this box partly explains the increasing deviation between both reanalyses at the end of the period.

## 6 Summary and conclusions

Atmospheric reanalyses play an important role in the global climate change assessment and their accuracy has significantly improved in recent years. In this study we investigated the means, variability, and trends in two modern reanalyses (ERA-Interim and MERRA-2). The means and variability in IWV in the reanalyses were inter-compared and compared to ground-based GPS data for the period 1995-2010. The global distribution of IWV in ERA-Interim and GPS is remarkably consistent, even in regions of strong gradients, where IWV varies strongly. However, ERA-Interim was shown to exhibit a slight moist bias in the extra-tropics (~ 0.5 kg m$^{-2}$) and a slight dry bias in the tropics (also found in comparison with MERRA-2), which is consistent with other studies (Trenberth et al., 2011). Inter-annual variability in ERA-Interim is highly consistent with GPS and is dominated by ENSO with variations as large as 20% IWV in DJF in the tropics and in the mid to high northern latitudes. Differences were pointed out between GPS and ERA-Interim at only a few stations, mostly located in coastal regions and regions of complex topography, where representativeness errors put a limit to the comparison of gridded reanalysis data and point observations.

Previous studies have concluded that during recent decades IWV has increased with time both over land and ocean regardless of the time period and dataset analysed, except for some of the older reanalyses and/or some inhomogeneous observational datasets (Trenberth et al., 2005; Dessler and Davis, 2010; Bock et al., 2014; Schröder et al., 2016; Wang et al. 2016). Nevertheless, most global atmospheric reanalyses still have substantial limitations in representing decadal variability and trends in the water cycle components because of assimilation increments and observing system changes (Trenberth et al., 2011). In this study we found that trends in IWV and surface temperature in ERA-Interim and MERRA-2 are fairly consistent, with positive IWV trends generally correlated with surface warming over most of the tropical oceans, as well as the Arctic, part of North America, Europe, and the Amazon. However, significant differences are found as well over several parts of the globe, with MERRA-2 presenting a more general global moistening trend compared to ERA-Interim. The most striking uncertainties are seen over Antarctica and most of the southern hemisphere, especially during JJA, where IWV trends are often of opposite signs, but also over most of central and northern Africa, as well as Indonesia, the Indian Ocean, and central Asia. The discrepancies are observed for both the extended common time record (1980-2016) and for the shorter time period (1995-2010) when GPS data are available. Over the latter period, the GPS IWV data point to a large erroneous negative (drying) trend in ERA-Interim over Antarctica. Few in-situ observations are available for assimilation in this region and the spurious trends in ERA-Interim might be due to model biases and changes and in the assimilated satellite data (Dee et al., 2011). Further investigation using assimilation feedback statistics and satellite IWV observations would help to better understand the origin of biases and spurious trends in this region. In most other regions, the trends in ERA-Interim have the same sign but different magnitudes than GPS, with positive biases in the tropics and negative biases in the higher northern latitudes.

Another distinct feature in ERA-Interim is the unphysically strong summer (JJA) negative IWV trend over north-eastern Africa. The absence of long term GPS records over this region prevents us from its direct assessment. The comparison with MERRA-2 indicates that both reanalyses have problems over Sahel and Sahara. This is actually not surprising as very few in-

situ observations are available in this region, and models are known to have large biases over this region. These biases are associated with problems in representing some of the governing continental physical and dynamical processes, namely the dry convection in the Saharan heat low and moisture advections from the ocean (Meynadier et al., 2010). Variations in IWV and atmospheric circulation are strongly correlated in this arid region. This co-variability provides a reasonable explanation for the observed variability and decadal trends in each of the reanalyses and of the differences between them (e.g. stronger increase of the dry northerly flow in ERA-Interim). Here as well, assimilation feedback statistics might help to understand the origin of biases, and their link with atmospheric dynamics in this region.

A more detailed investigation of IWV, surface temperature, and atmospheric circulation was also presented for western Australia, which is in many aspects governed by similar atmospheric processes as northern Africa (dry continental convection associated with a heat low and summer monsoon). However, this region benefits from more direct in-situ observations, as well as moisture and surface wind observations over the ocean from space, which have a strong impact on moisture transport from the surrounding oceans to the continent. Hence it is not surprising that both reanalyses are in better agreement and closer to the observed GPS IWV trends there. Interestingly, the region is marked by positive trends in surface temperature and IWV for the longer period, consistently with the global warming, but with an opposite IWV trend for the shorter period. The time series of IWV anomalies and surface temperature show that strong interannual to decadal variability in IWV is again correlated with anomalies in atmospheric circulation with colder years being wetter.

Compared to past studies using older reanalyses, we find that modern reanalysis made significant improvements in the representation of IWV means and the strong interannual variability over the oceans and most continental areas. However, the weaker decadal variability and trends still suffer from large uncertainties in data-sparse regions such as Africa and Antarctica. More generally, model biases and changes in the observing system are still suspected, which prevent reanalyses produced with different models and assimilation systems from being consistent to better than about ± 10 % IWV per decade. It will be of special interest as future work to investigate ERA5, the new reanalysis from ECMWF, which benefits from many improvements compared to ERA-Interim (https://software.ecmwf.int/wiki/pages/viewpage.action?pageId=74764925).

An absolute assessment of the reanalyses could be made in this study using independent IWV data from the ground-based GPS network for the period 1995-2010. Even though the GPS data were produced using homogeneous reprocessing and quality check, inhomogeneities due to equipment changes were evidenced for a small number of sites. Homogenisation of the GPS dataset is currently being undertaken using different processing and modelling strategies as well as statistical homogenisation techniques that should help detection and correcting the biases and offsets. An extension of the dataset is also planned as a few more years of observations are now available for reprocessing.

**Appendix**

There are many methods used in the literature to estimate linear trends from geophysical data. Two of the most widely used are compared in this Appendix: Theil-Sen method (after Theil, 1950, and Sen, 1968) which we used in the manuscript, and the

Least Squares method (e.g.: Weatherhead et al., 1998). Both methods assume that the data time series, $y_i$, can be modelled by a linear function of time $t_i$ of the form:

$$y_i = at_i + b + N_i \qquad\qquad i = 1, ..., n \quad \text{(A1)}$$

Where a and b are the unknown slope and intercept parameters to be estimated, and $N_i$ is the random noise or error. The ordinary Least Squares Method (LSM) determines the parameters $\hat{a}$ and $\hat{b}$ that minimizes the sum of squared residuals given by:

$$\min_{(\hat{a},\hat{b})} \sum_{i=1..n} \left[ y_i - (\hat{a}t_i + \hat{b}) \right]^2 \qquad\qquad \text{(A2)}$$

More sophisticated variants are described in Rousseeuw and Leroy (2003) which include non-uniform weighting and models that take autocorrelation in the noise term into account whereas the ordinary LSM assume identically independent noise samples.

The Theil-Sen estimator as defined by Theil (1950) is a non-parametric method that determines the trend by computing the median of slopes of lines through all pairs of points in the time series:

$$\hat{a} = \operatorname*{med}_{1 \le i < j \le n} \frac{y_j - y_i}{t_j - t_i} \qquad\qquad \text{(A3)}$$

Once the slope has been determined, the intercept is derived from $\hat{b} = \operatorname*{med}_{1 \le i \le n} y_i - \hat{a}t_i$. Sen (1968) extended the method to handle ties among the times (i.e. the case when two data points have the same time).

Both methods were applied to the global ERA-Interim and GPS monthly mean IWV anomalies, obtained after the removal of the seasonal cycle as described in Section 2 of the paper, for the 1995-2010 period. The results are shown in Fig. A1.

The differences between trends obtained using the two methods for the ERAI data are shown in Fig. A1a. These differences are under 0.5 kg.m$^{-2}$.decade$^{-1}$ for most of the globe, except around the Equator, where the ordinary LSM overestimates the trends in the eastern Pacific Ocean and underestimates the trends in the western Pacific Ocean. Consistent results are observed from the GPS IWV anomalies including gaps in the times series as shown in Fig. A1b.

The time series and trends at two points over the regions with large opposite differences are shown in Figs. A1c (eastern Pacific Ocean) and d (Coco Island in the western Pacific Ocean). It is observed that the Theil-Sen method is less affected by the strong positive anomalies observed in 1997/1998 in the tropical Pacific (due to a strong El Niño event), and at the end of the time series, in 2010, for the Coco Island GPS station.

The Theil-Sen estimator is known to be generally more robust than the Least Squares method (Rousseeuw and Leroy, 2003) and less sensible to the beginning and ending of the time-series (Wang et al., 2016), so this was the method chosen to estimate the trends analysed throughout the paper.

**Acknowledgements.** This work was developed in the framework of the VEGA project and supported by the CNRS program LEFE/INSU and ANR REMEMBER project (grant ANR-12-SENV-001). This work is a contribution to the European COST Action ES1206 GNSS4SWEC (GNSS for Severe Weather and Climate monitoring; http://www.cost.eu/COST_Actions/essem/ES1206) aiming at the development of the global GPS network for atmospheric
research and climate change monitoring.

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

**Table 1: Statistics (median ± one standard deviation over 104 stations) of differences (ERAI minus GPS) of mean IWV values and of relative standard deviations.**

| | Diff. of mean IWV (ERAI – GPS) | Diff. of rel. std. (ERAI – GPS) |
|---|---|---|
| **DJF** | +0.51 kg.m$^{-2}$ ± 0.83 kg.m$^{-2}$ | -0.05 % ± 1.69 % |
| | +6.2 % ± 6.9 % | |
| **JJA** | +0.52 kg.m$^{-2}$ ± 0.95 kg.m$^{-2}$ | -0.15 % ± 4.07 % |
| | +2.7 % ± 7.8 % | |

5   **Table 2: Stations with most intense trend differences (ERAI – GPS) computed from time-matched GPS and ERA-Interim IWV series.**

| | Full time series | DJF | JJA |
|---|---|---|---|
| Tr.diff < -1 kg.m$^{-2}$.decade$^{-1}$ | WUHN, CRO1, CFAG, SHAO | WUHN, PIN1 | WUHN, SHAO, CRO1, CFAG, KOUR, WSLR |
| Tr.diff > 1 kg.m$^{-2}$.decade$^{-1}$ | CCJM | CCJM, DARW, GUAM, LPGS, COCO, SANT | CCJM, POL2 |
| Tr.diff < -7 %.decade$^{-1}$ (*) | MCM4, MAW1, PIN1 | IRKT, POL2, PIN1, WUHN, YELL, WSLR | MCM4, MAW1, SYOG |
| Tr.diff > 7 %.decade$^{-1}$ (*) | CCJM | CCJM, KIRU | AREQ |

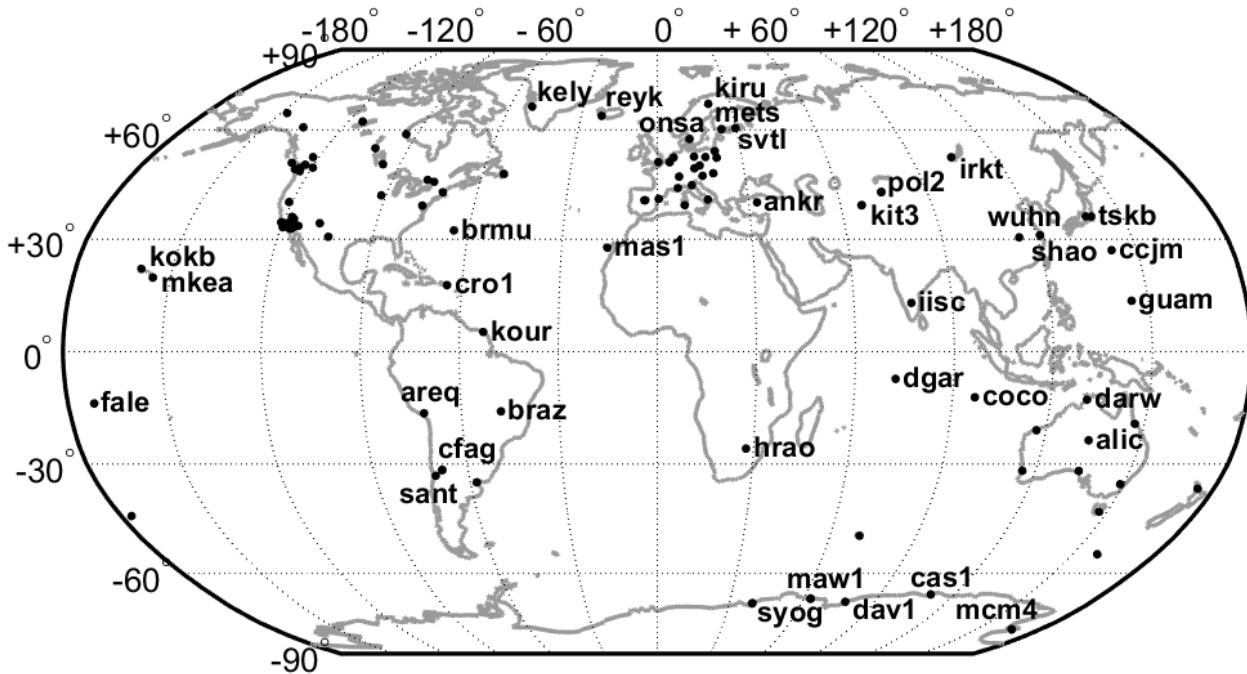

**Figure 1: Map showing the 104 GPS stations used in this study. The stations discussed in the text are identified by their 4-character ID.**

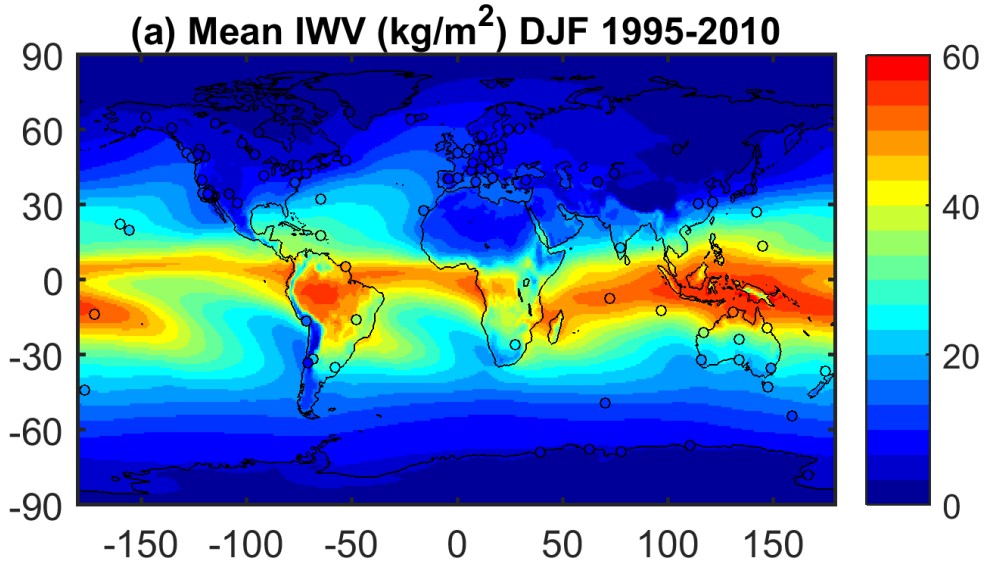

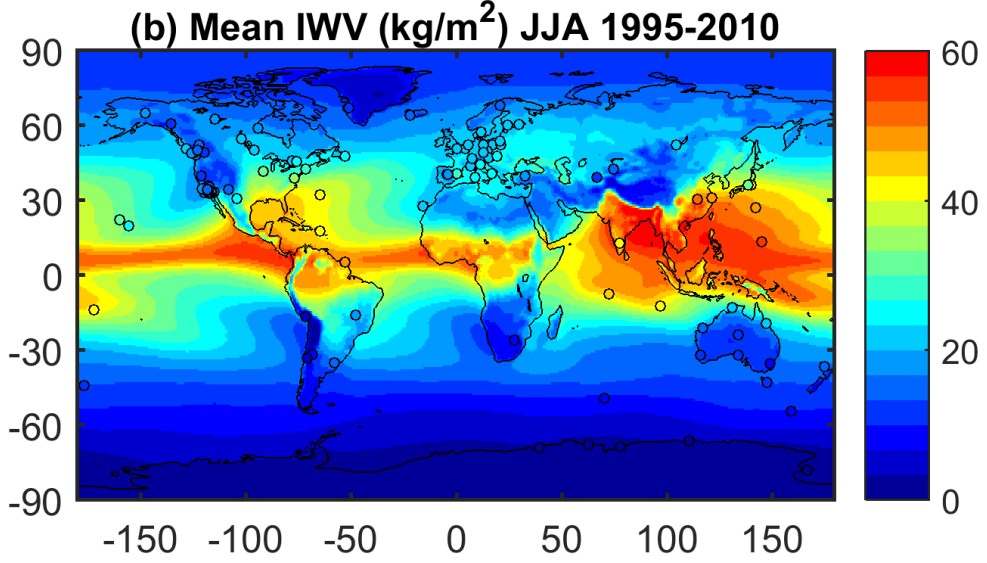

**Figure 2: (a) Mean IWV for DJF 1995-2010 from ERA-Interim (shading) and GPS (filled circles), (b) same as (a) for JJA.**

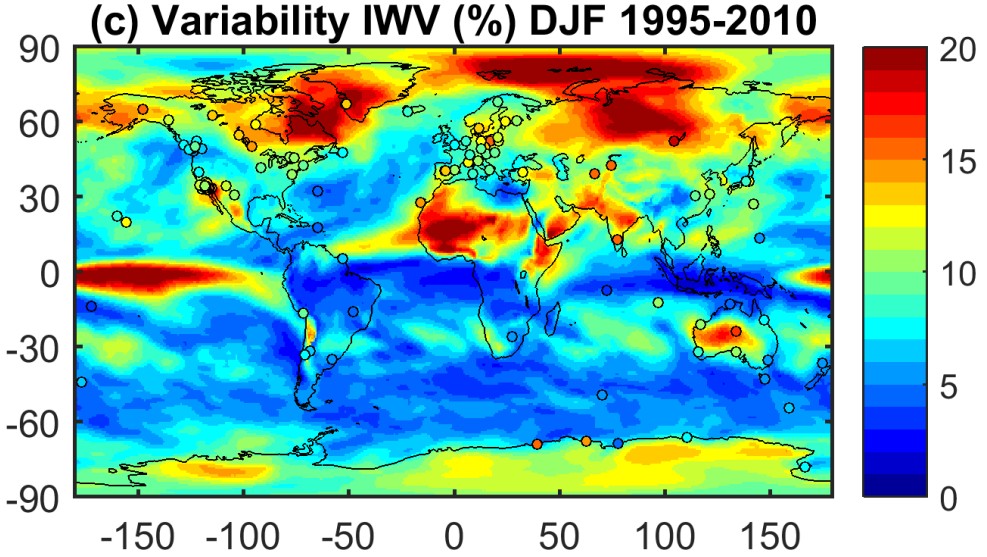

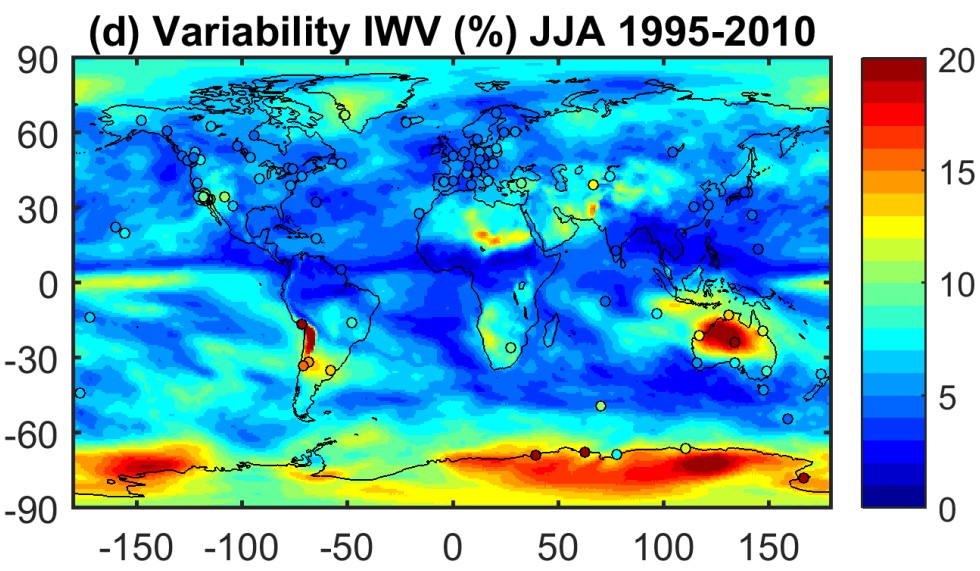

**Figure 2 (continued): (c) Relative variability in % (standard deviation of the IWV series divided by its mean) for DJF 1995-2010, (d) Same as (c) for JJA.**

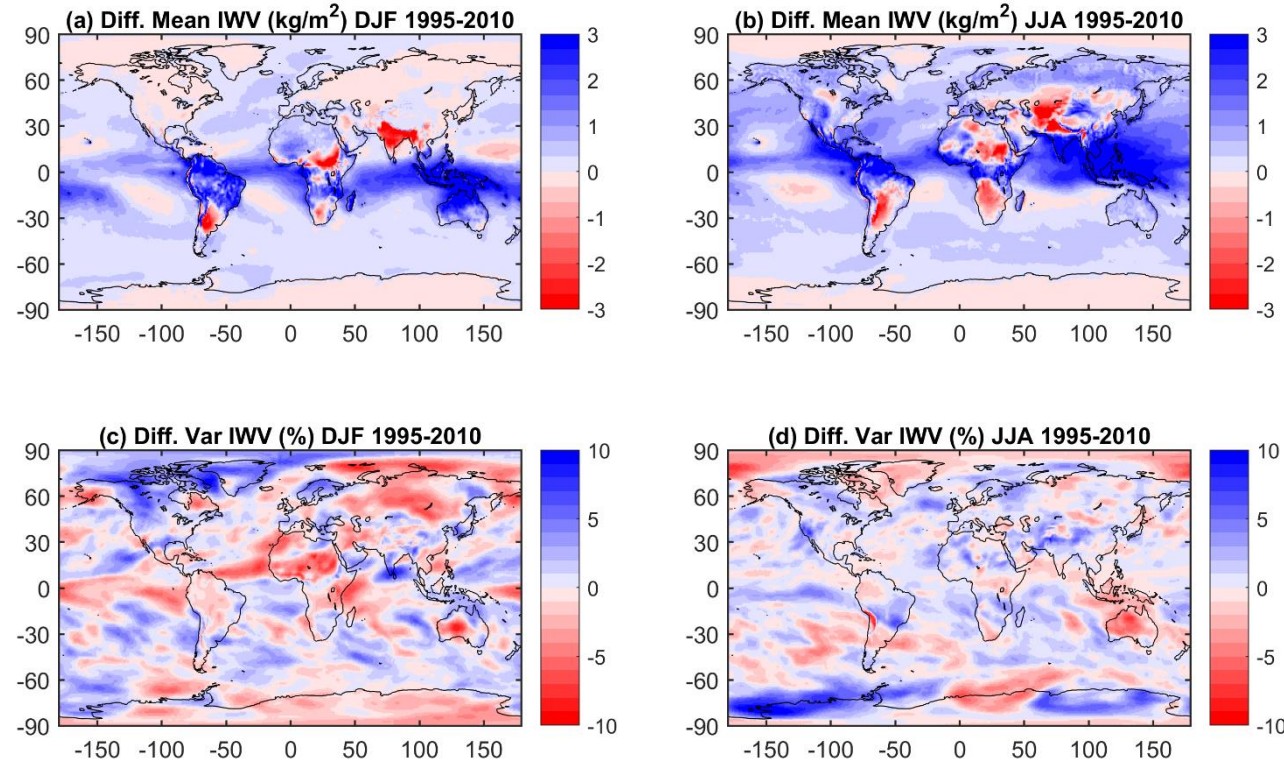

**Figure 3: (a)** Difference of mean IWV estimates (MERRA-2 minus ERA-Interim) for DJF 1995-2010. The global mean difference is 0.35 kg.m-2 (0.94 %) and the standard deviation of the difference is 0.32 kg.m-2 (1.56 %). **(b)** Same as (a) for JJA. The global mean difference is 0.56 kg.m-2 (1.66 %) and the standard deviation of the difference is 0.37 kg.m-2 (1.67 %). **(c)** Difference of relative variability estimates (MERRA-2 variability minus ERA-Interim variability) for DJF 1995-2010. The global mean difference is -0.01 % and the standard deviation of the difference is 0.26 %. **(d)** same as (c) for JJA. The global mean difference is 0.28 % and the standard deviation of the difference is 0.42 %.

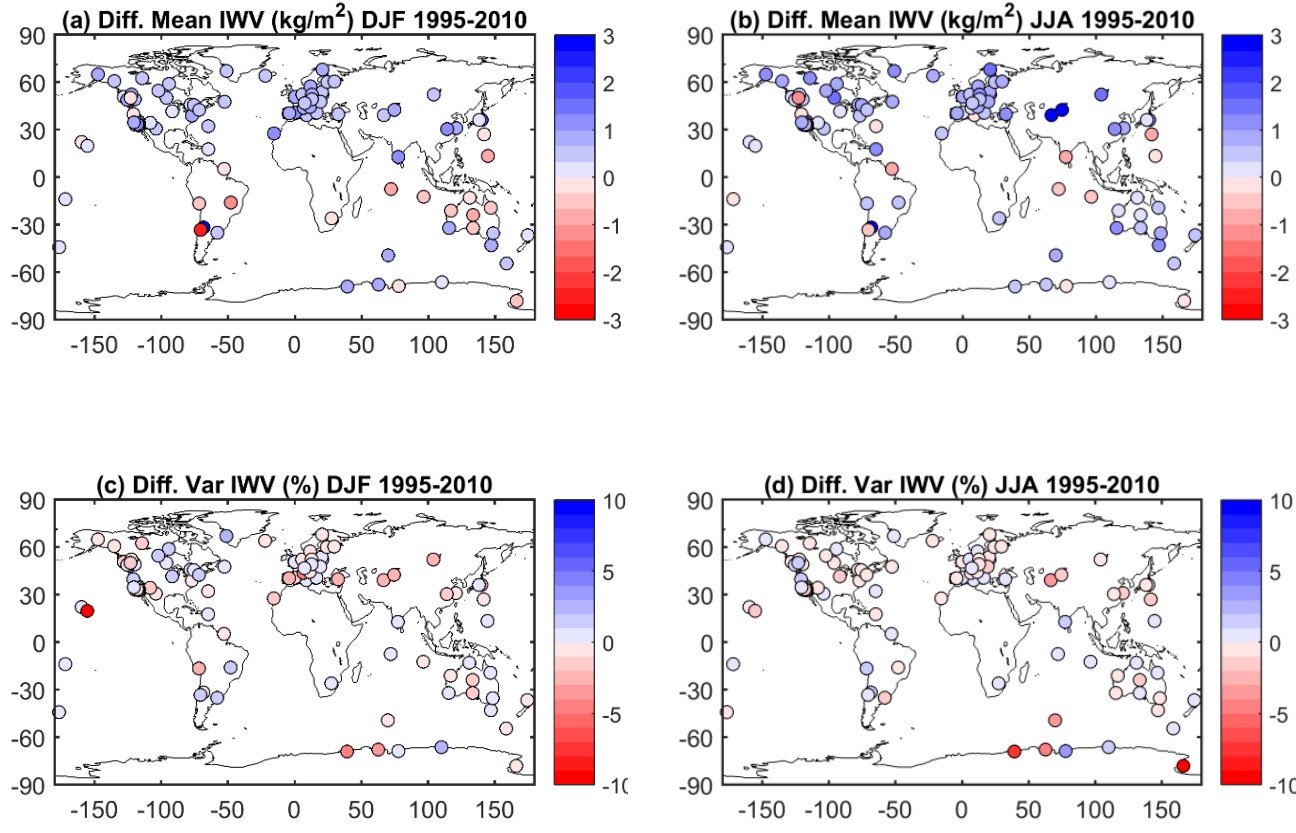

**Figure 4: (a) Difference of mean IWV estimates (ERA-Interim minus GPS) for DJF 1995-2010 from time-matched IWV series, (b) same as (a) for JJA, (c) difference of relative variability estimates (ERA-Interim variability minus GPS variability) for DJF 1995-2010 from time-matched IWV series, (d) same as (c) for JJA.**

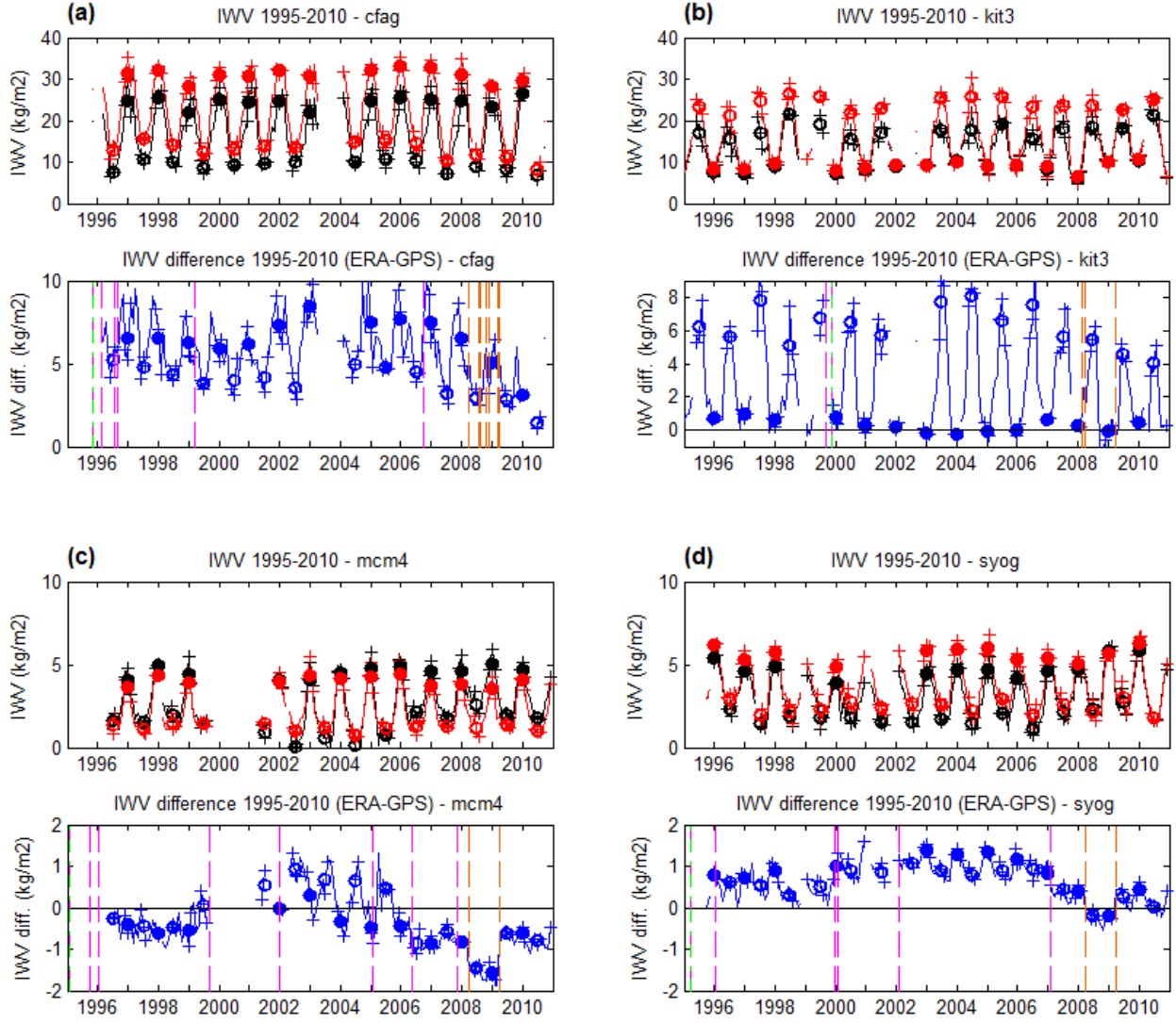

**Figure 5: Time series of IWV from GPS (black) and ERAI (red), and IWV difference (blue) at stations (a) CFAG, (b) KIT3, (c) MCM4 and (d) SYOG. Filled circles show the DJF values and open cicles the JJA values. Crosses show the individual months used in both seasons. Vertical dashed lines indicate GPS equipement changes (receiver in magenta, antenna in green) and GPS processing changes (in orange). Note the change in vertical scales between figures.**

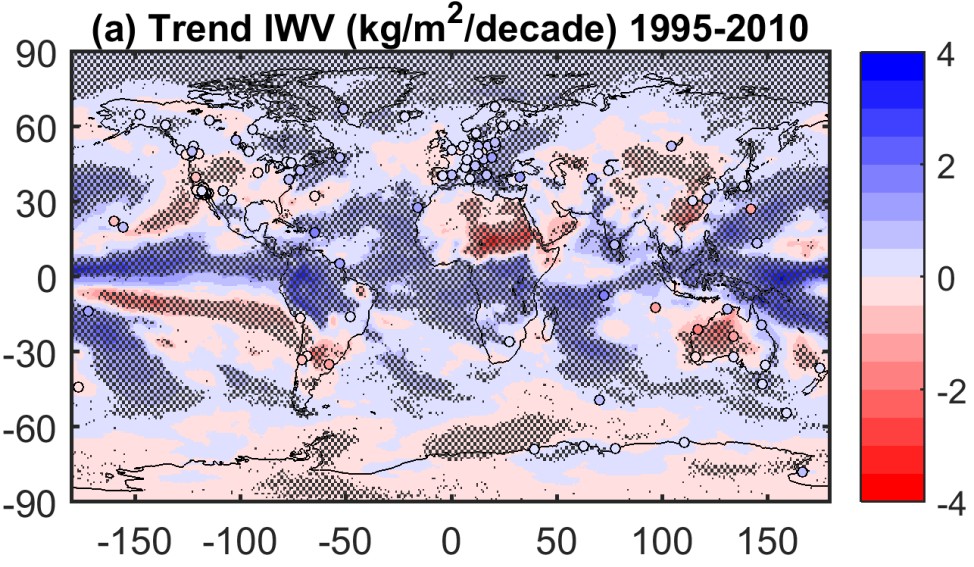

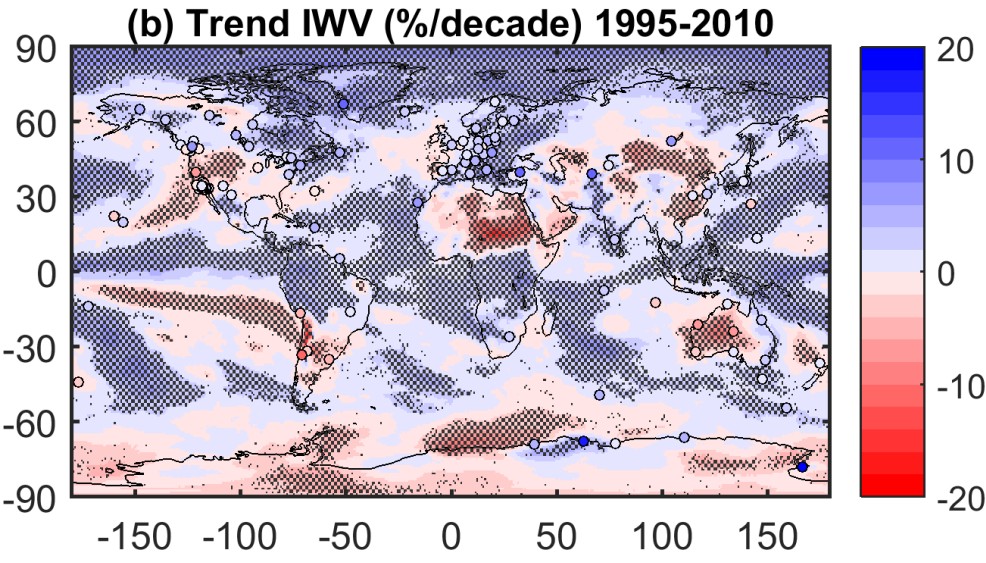

**Figure 6: Absolute (a) and relative (b) IWV trends for the 1995-2010 period from ERA-Interim and GPS (stations marked as circles). The statistically significant trends from ERA-Interim are highlighted by stippling. Absolute trends are in kg.m$^{-2}$ per decade and relative trends in % per decade.**

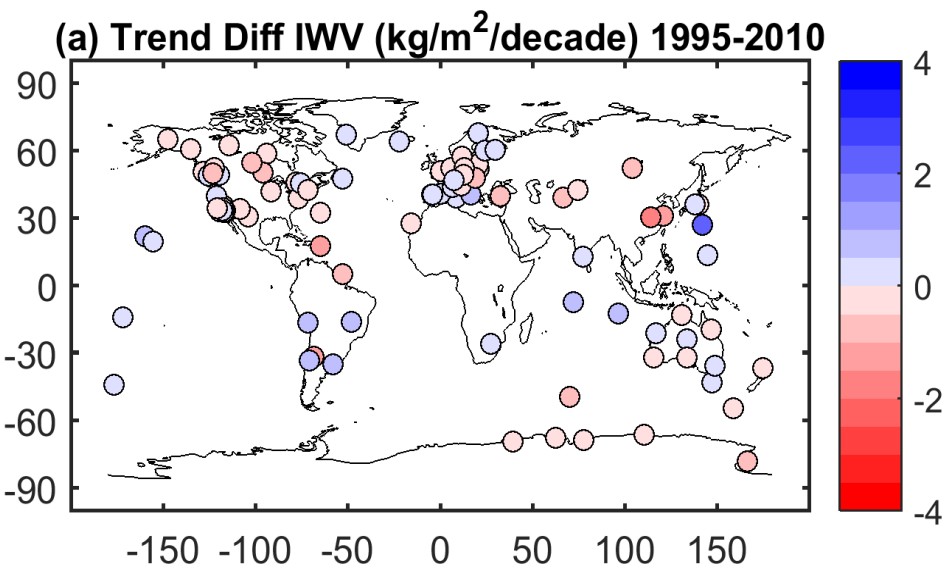

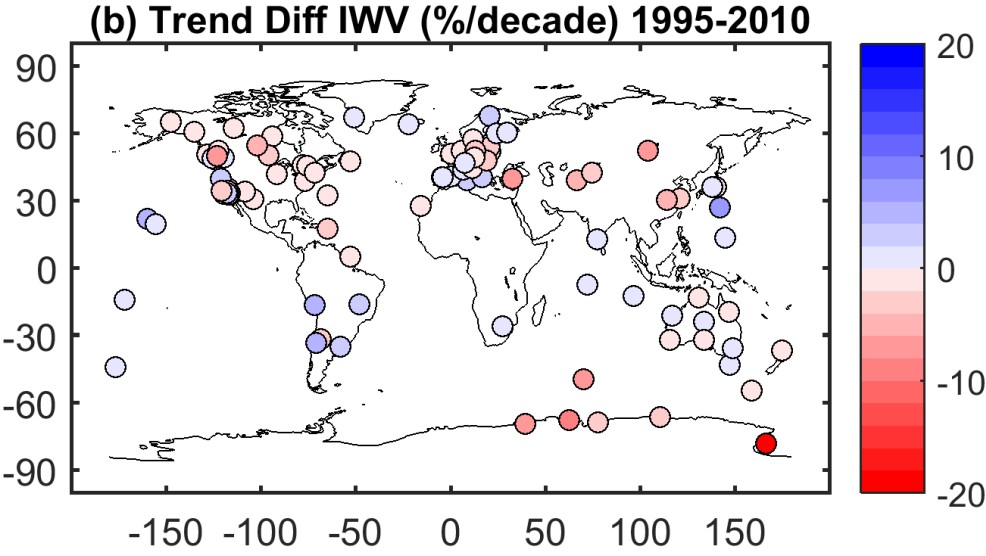

**Figure 7: Difference of IWV trends (ERA-Interim minus GPS) for the 1995 to 2010 period, for time-matched series: (a) trends in kg.m-2 per decade and (b) relative trends in % per decade.**

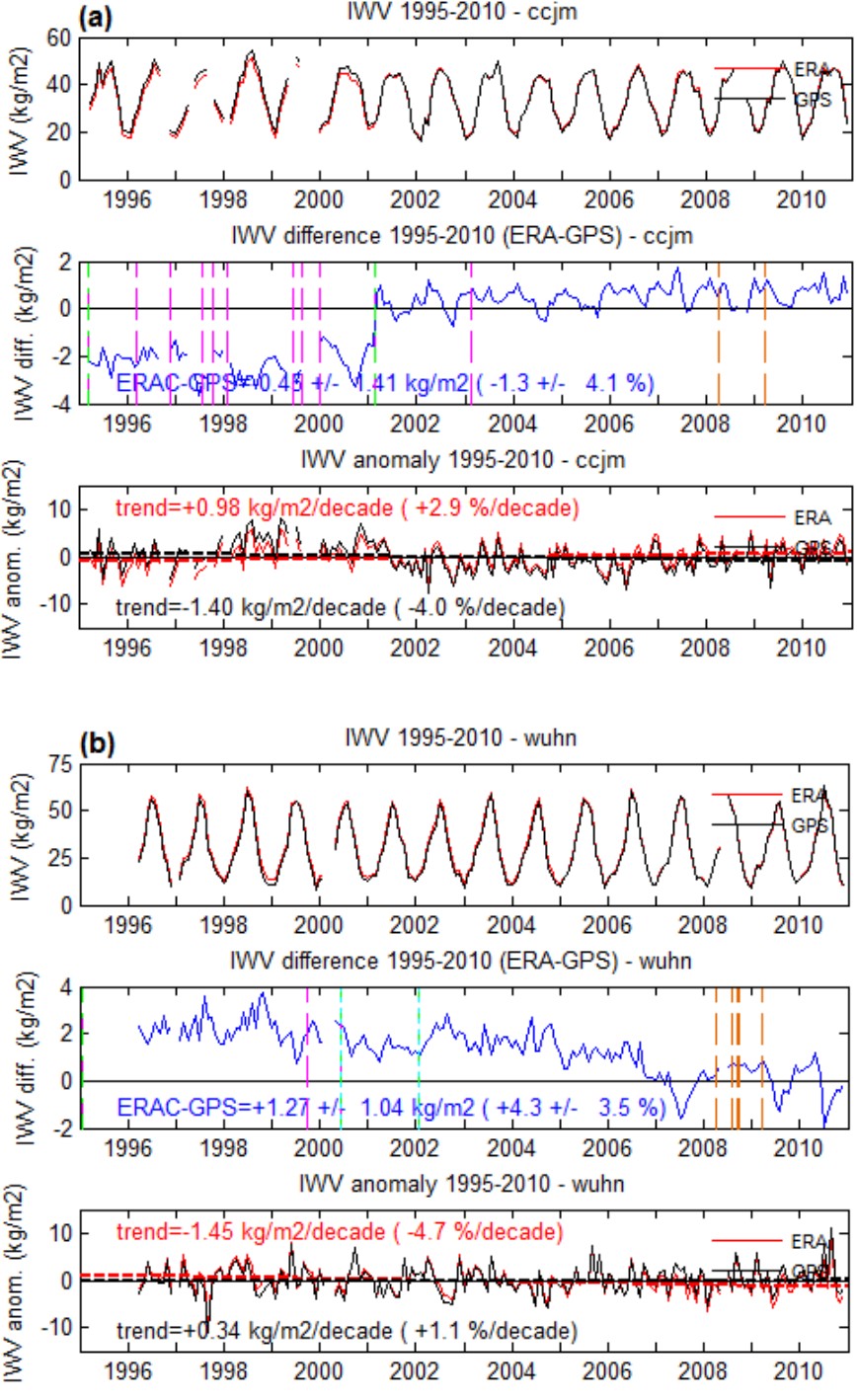

**Figure 8: Examples of GPS time series with inhomogeneities: (a) station CCJM and (b) WUHN. Upper plots show IWV for GPS (black) and ERA-Interim (red), middle plots show IWV difference (ERA-Interim minus GPS) and lower plots show monthly anomalies with their linear trend estimates (dashed lines).**

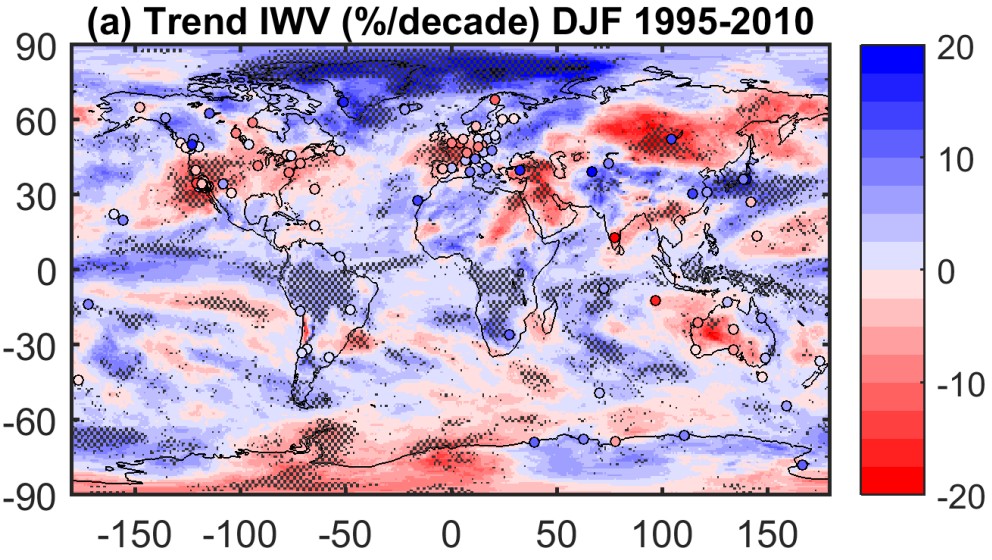

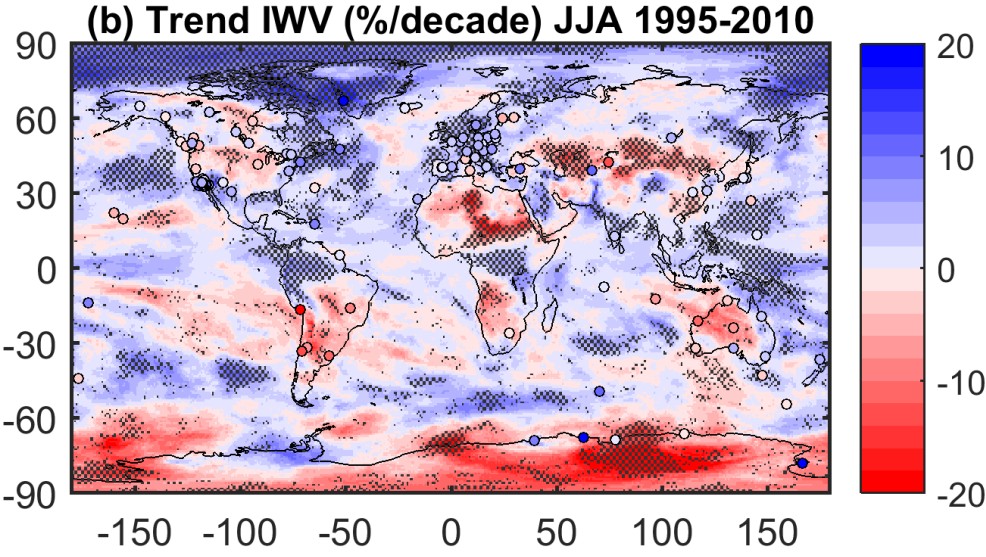

**Figure 9: Seasonal IWV trends for the 1995-2010 period from ERA-Interim (shading) and GPS (filled circles) for DJF (a) and JJA (b). The statistically significant trends from ERA-Interim are highlighted by stippling.**

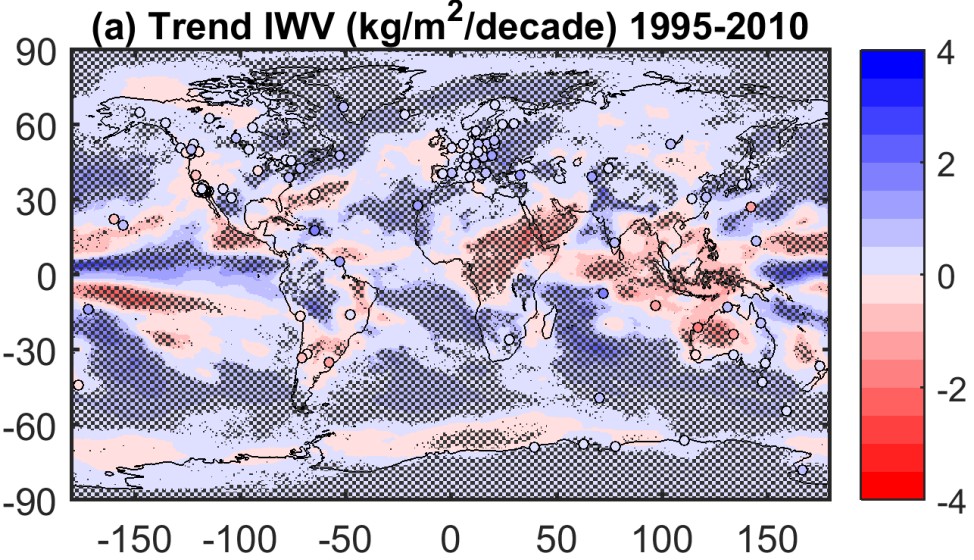

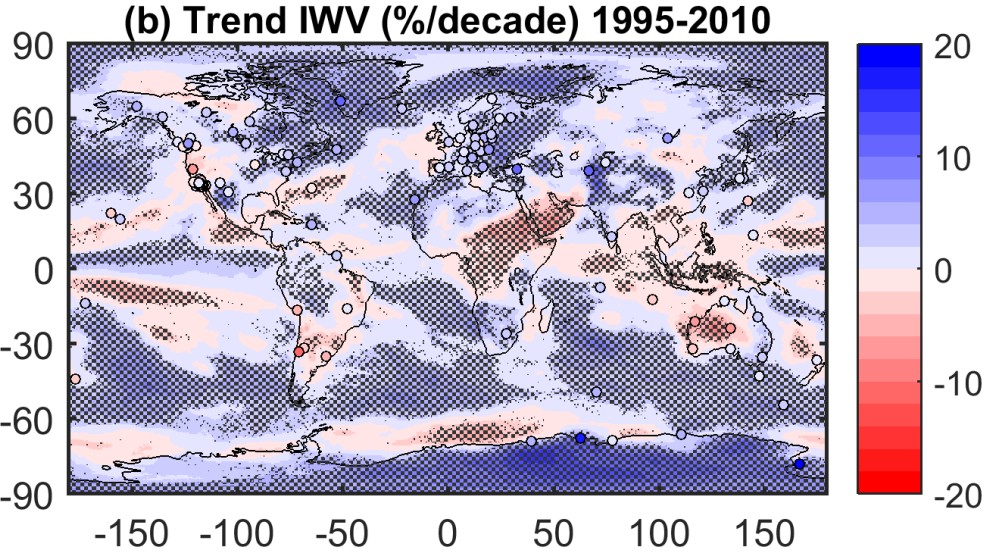

**Figure 10: Absolute (a) and relative (b) trends in IWV in the MERRA-2 reanalysis for the 1995-2010 period. The statistically significant trends are highlighted by stippling.**

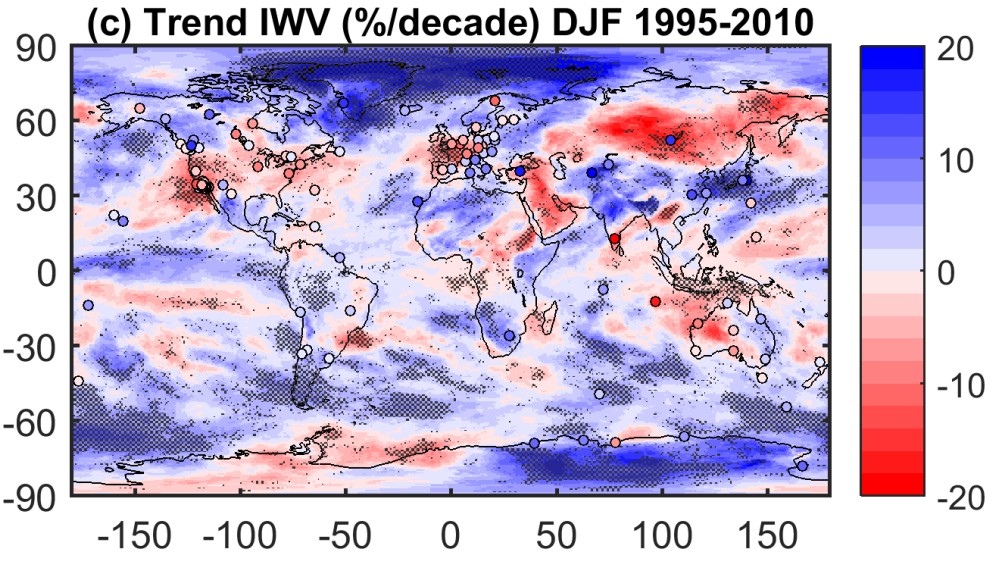

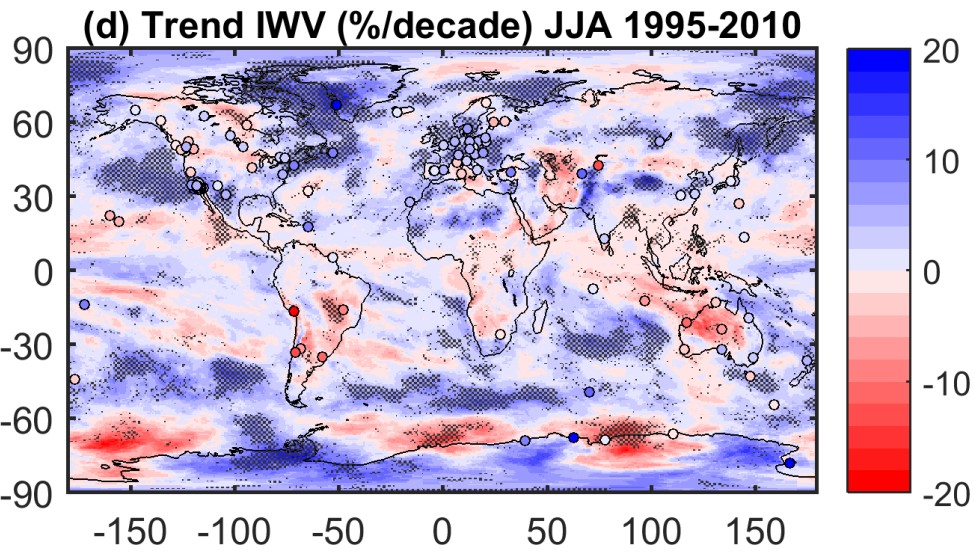

**Figure 10 (continued): Relative trends in IWV in the MERRA-2 reanalysis for the 1995-2010 period for DJF (c) and JJA (d). The statistically significant trends are highlighted by stippling.**

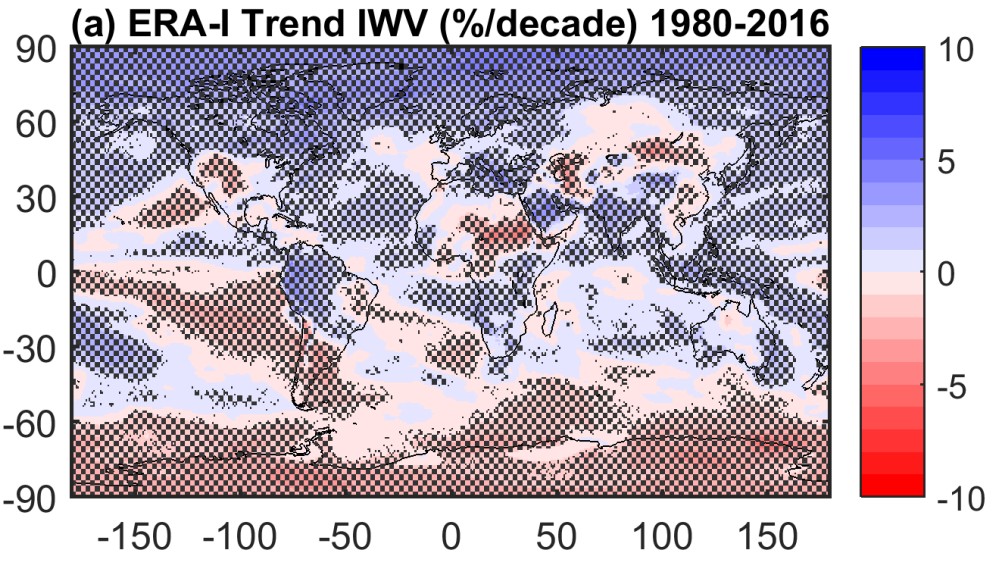

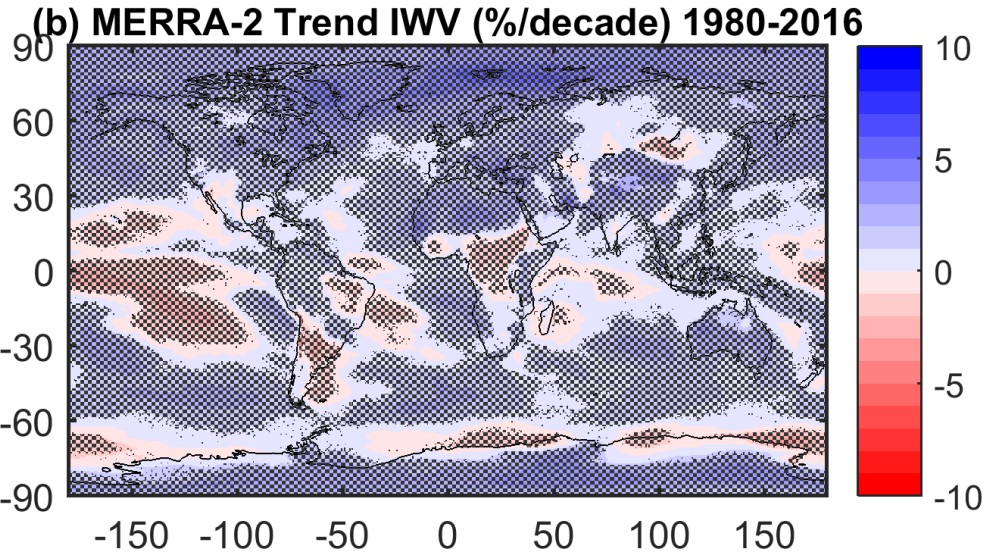

**Figure 11: Monthly trends in IWV for the 1980-2016 period for: (a) ERA-Interim, (b) MERRA-2. The statistically significant trends are highlighted by stippling.**

**ERA-Interim**              **MERRA-2**

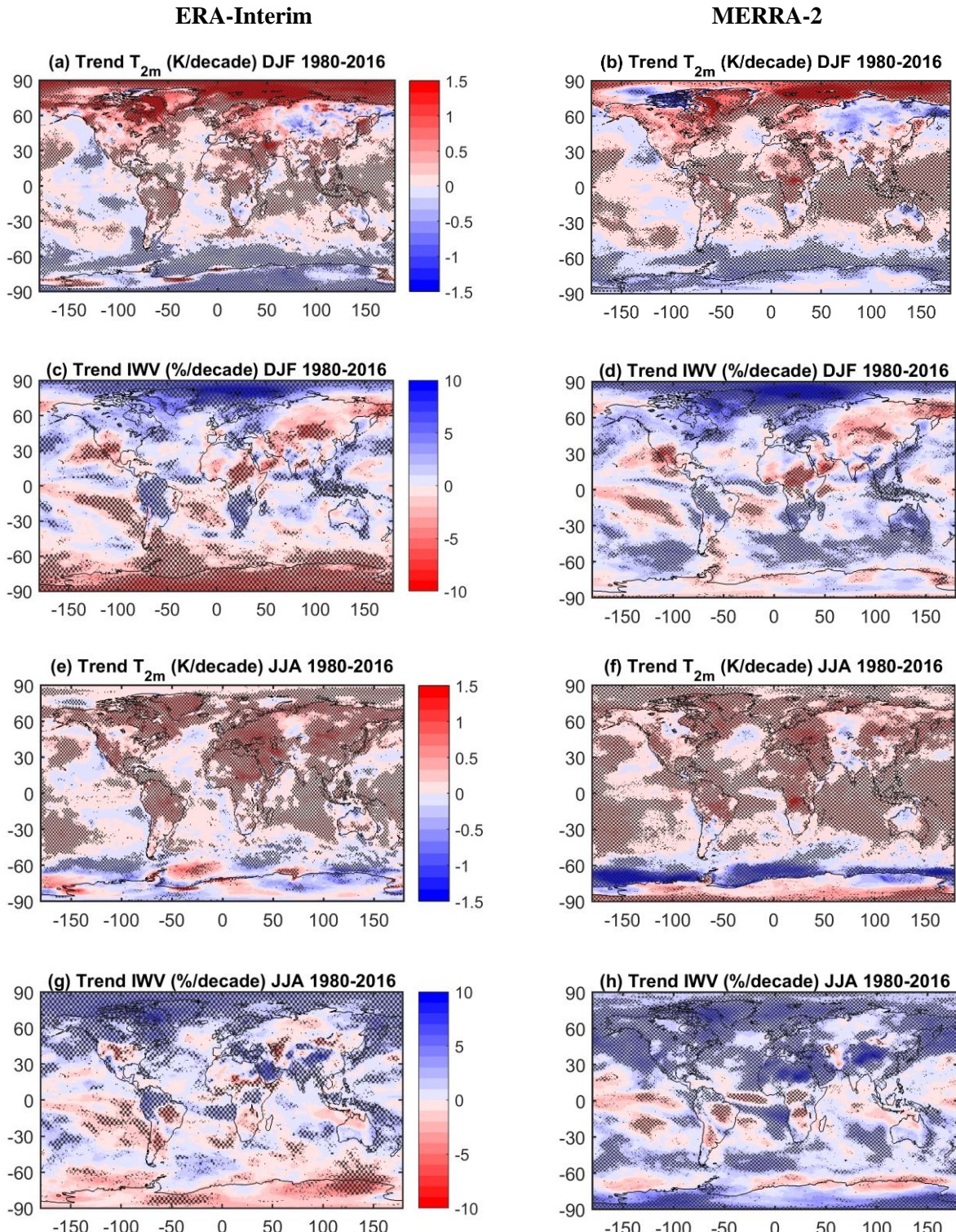

Figure 12: Seasonal trends in T2m and IWV for the 1980 to 2016 period for: (left) ERA-Interim and (right) MERRA-2.

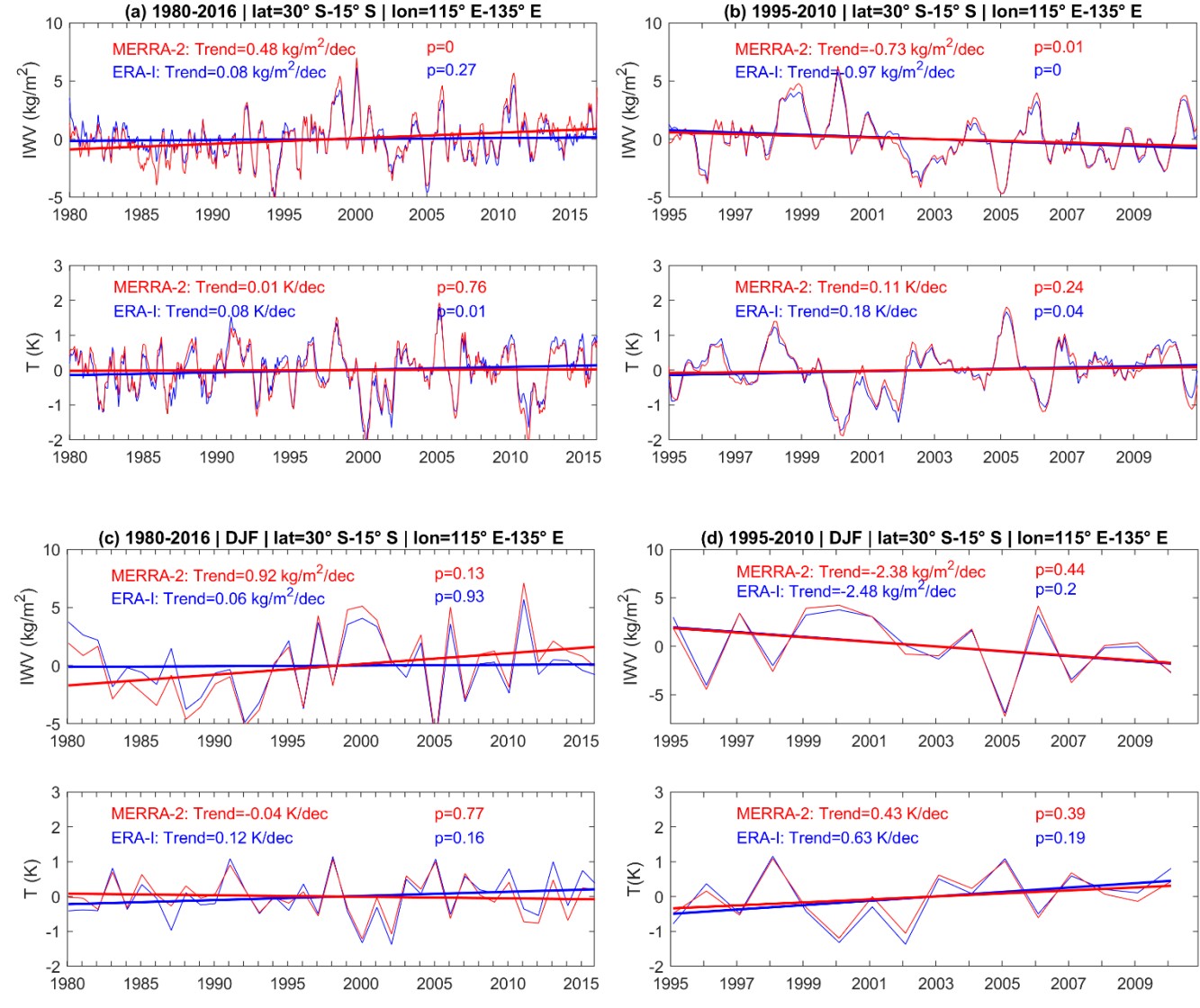

**Figure 13: Temperature and IWV anomalies time series for a box over Western Australia (see Fig. 14), using ERA-Interim (blue) and MERRA-2 (red) data, for: (a, c) the 1980 to 2016 period, (b, d) the 1995 to 2010 period, and (a, b) the monthly time series and (c, d) the DJF season.**

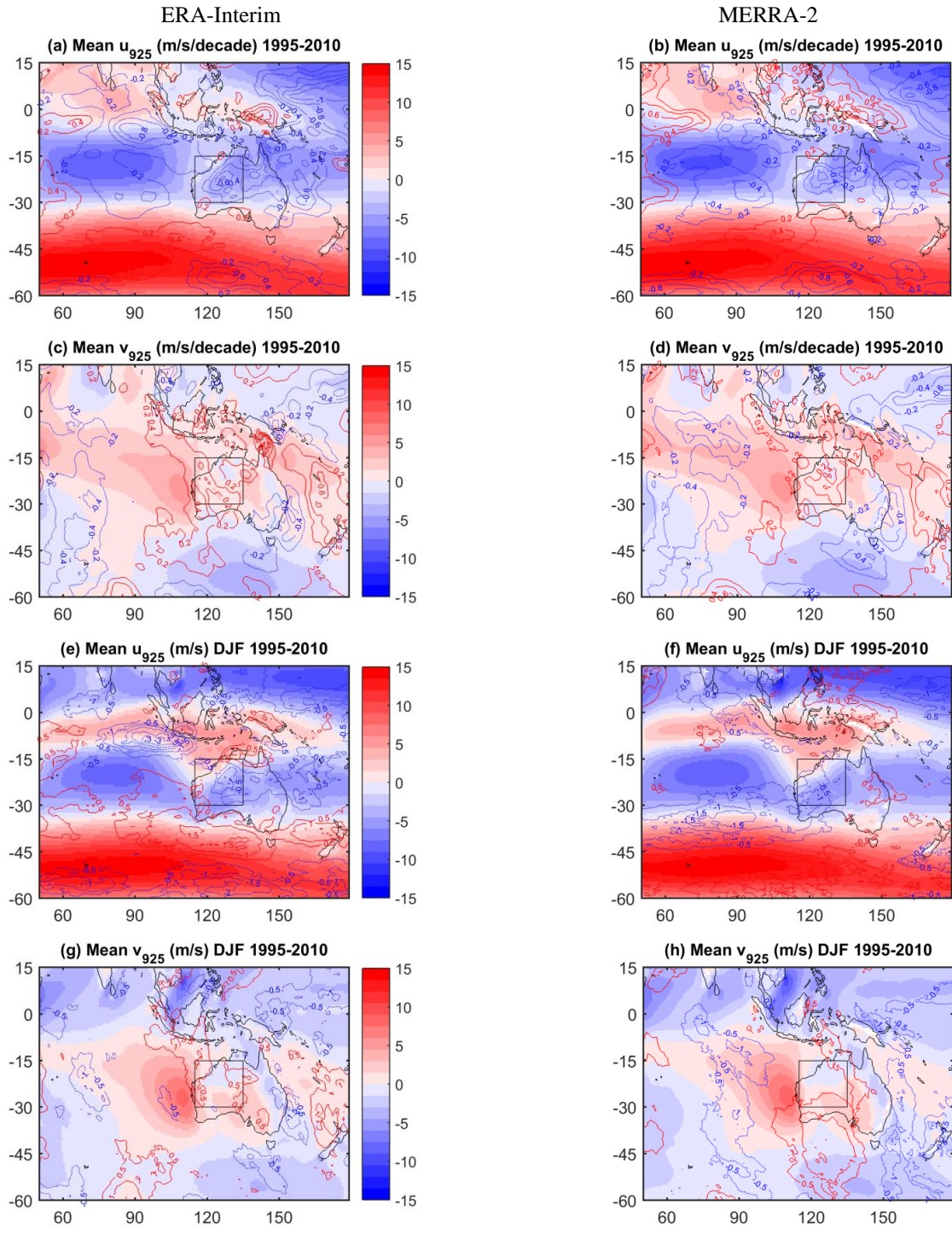

**Figure 14: Zoom over Western Australia of the mean monthly and DJF fields and trends of the u and v wind components at 925 hPa (shaded) and their trends (contours). The area of focus (where IWV trends are most intense in ERA-Interim) is marked by a box.**

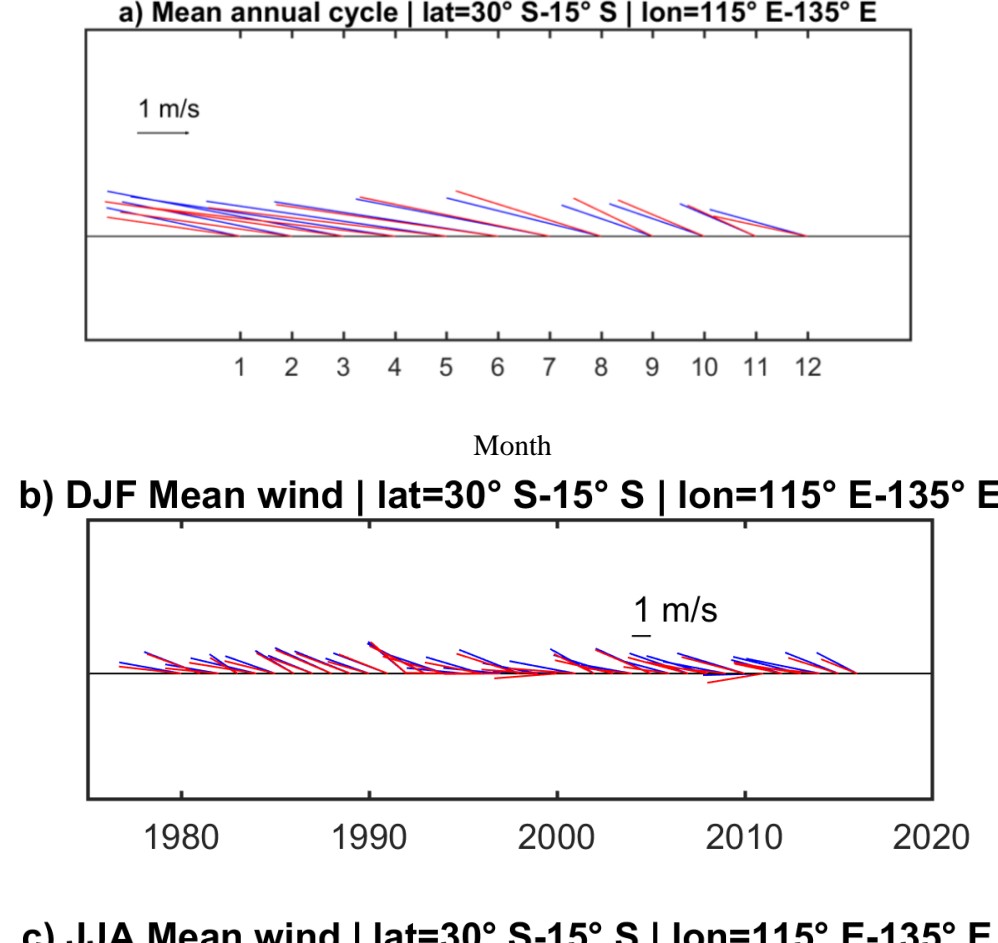

Month

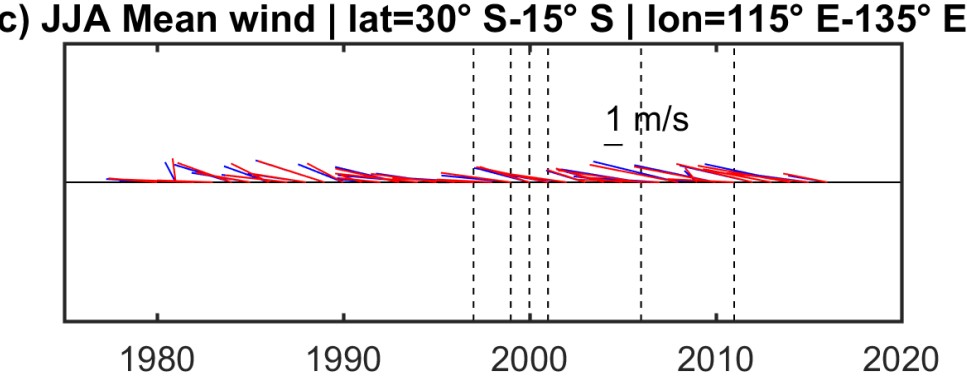

**Figure 15: Time series of mean wind vectors for a box over Western Australia (see Fig. 14), using ERA-Interim (blue) and MERRA-2 (red) data, for the 1980 2016 period: mean annual cycle (a), and the monthly time series for the DJF and JJA seasons (b, c).**

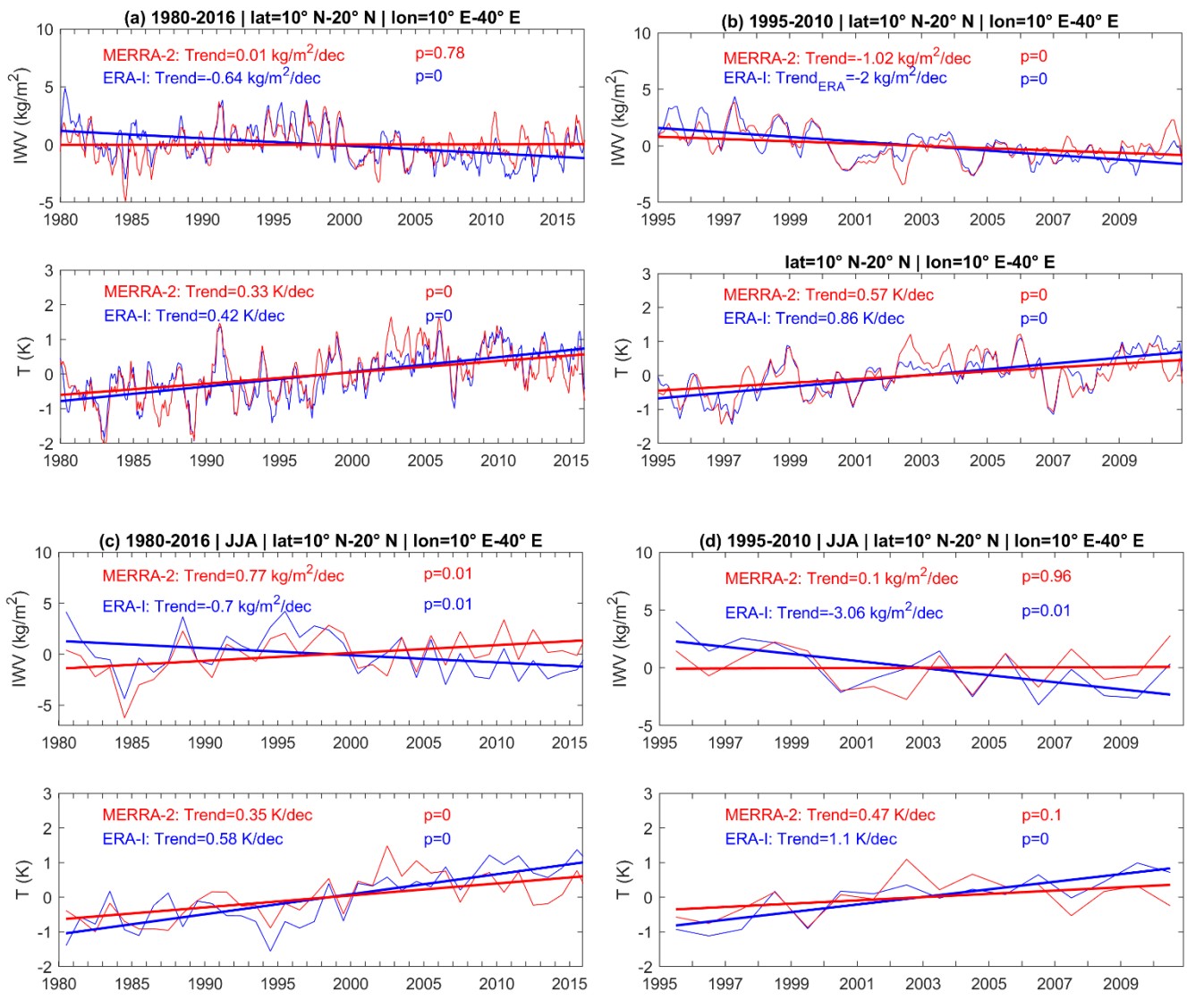

**Figure 16: Temperature and IWV anomalies time series for a box over eastern Sahel (see Fig. 17), using ERA-Interim (blue) and MERRA-2 (red) data, for: (a, c) the 1980 to 2016 period, (b, d) the 1995 to 2010 period, and (a, b) the monthly time series and (c, d) the JJA season.**

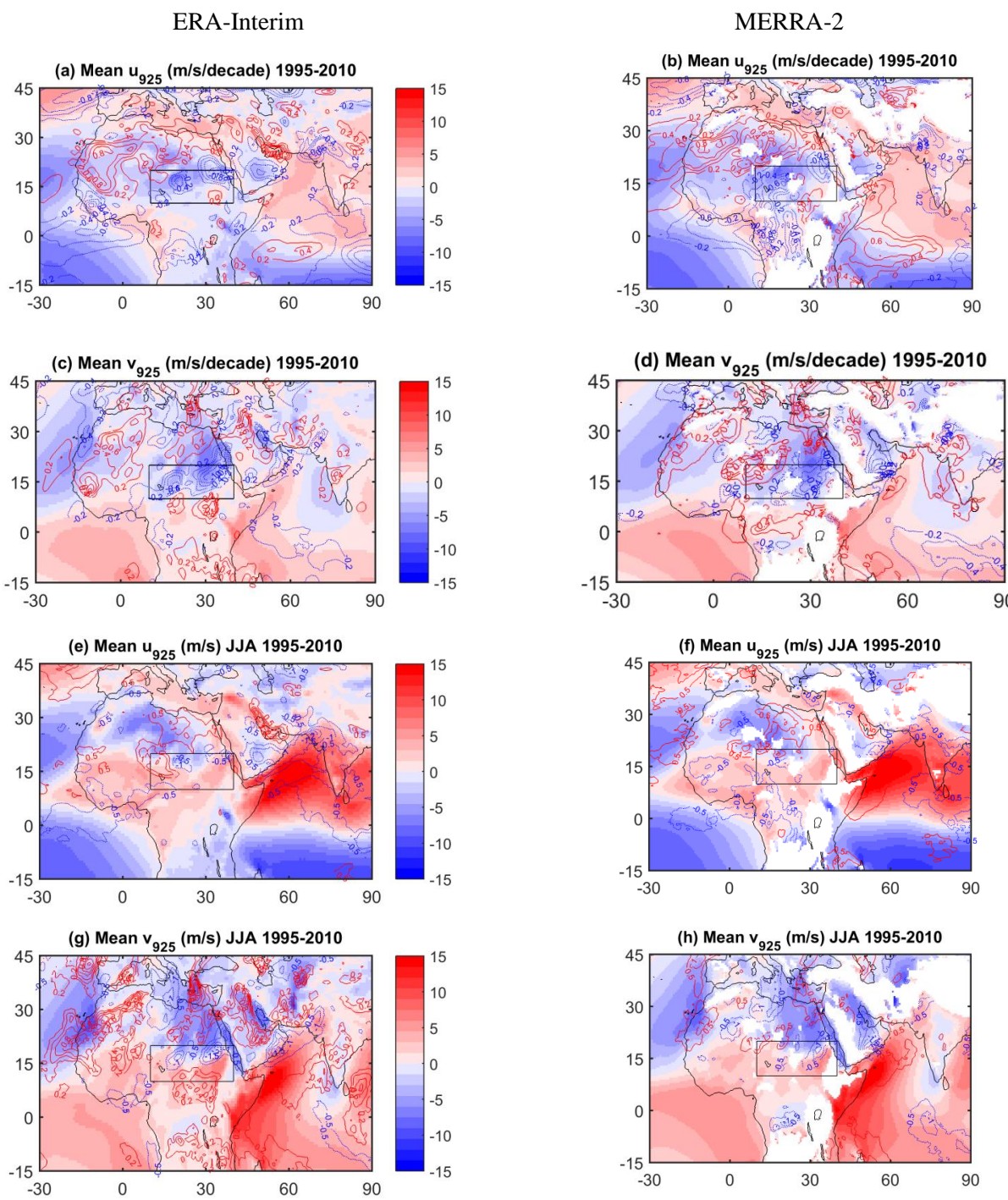

**Figure 17: Zoom over North Africa of the mean monthly and JJA fields and trends of the u and v wind components at 925 hPa (shaded) and their trends (contours). The area of focus (where IWV trends are most intense in ERA-Interim) is marked by a box.**

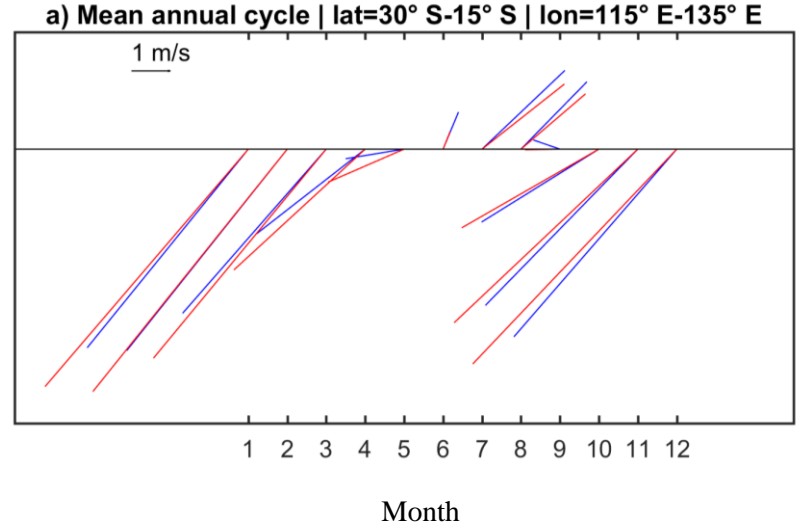

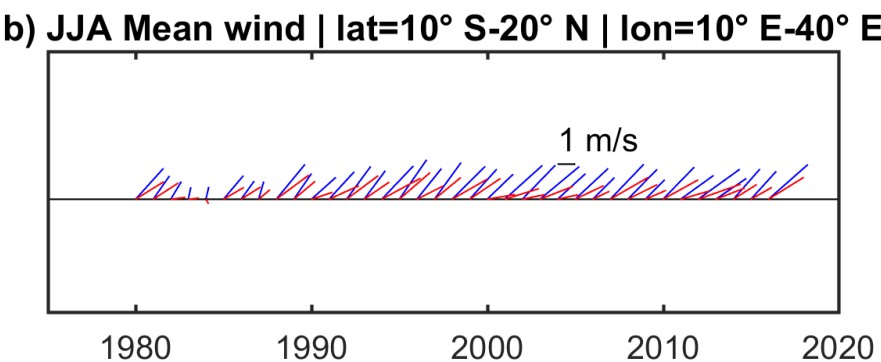

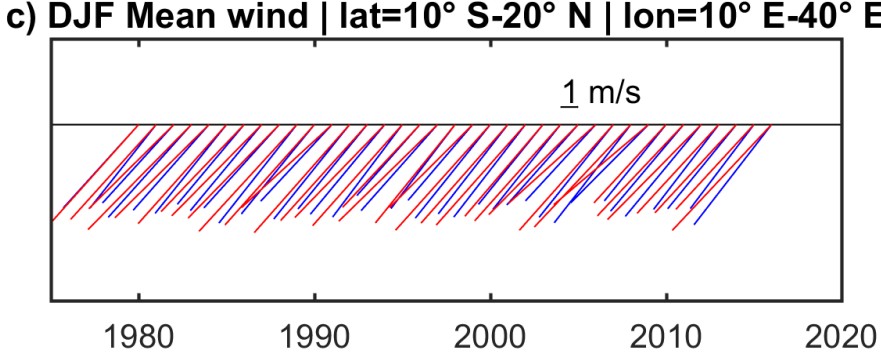

**Figure 18: Time series of mean wind vectors for a box over Eastern Sahel (see Fig. 17), using ERA-Interim (blue) and MERRA-2 (red) data, for the 1980 2016 period: mean annual cycle (a), and the monthly time series for the JJA and DJF seasons (b, c).**

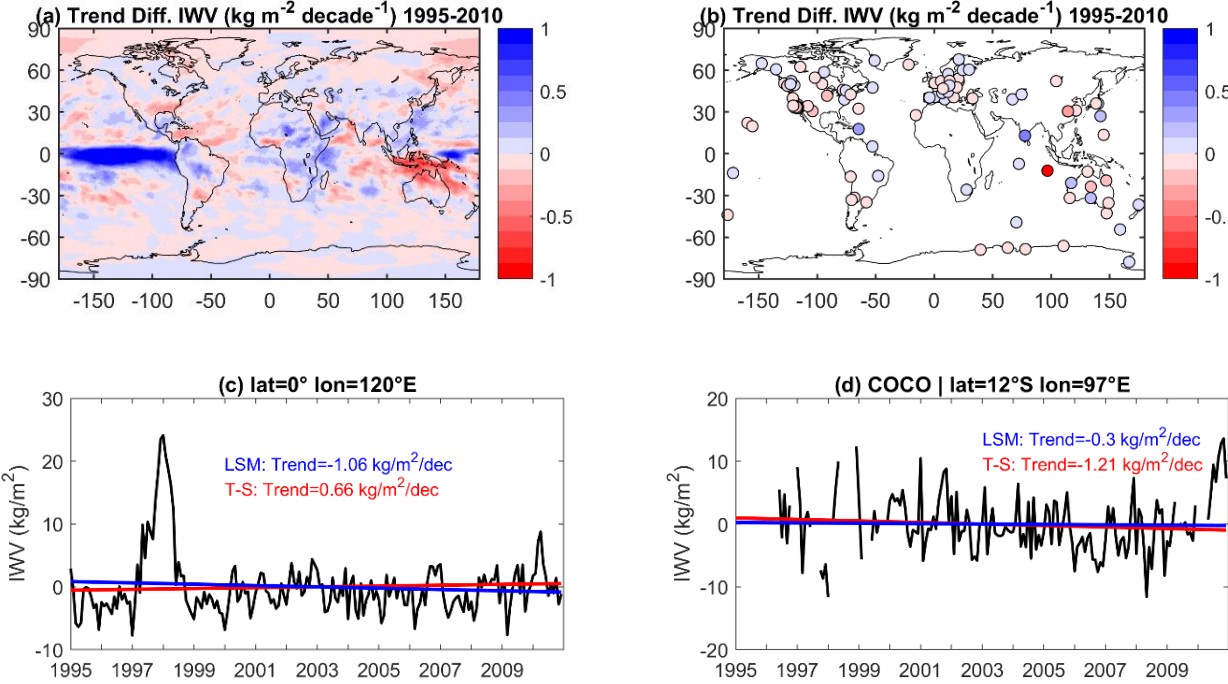

**Figure A3: (a)Trend differences between two methods of computing trends (Theil Sen minus the Least Squares method) for the ERA-Interim data used in the paper. The root mean square of the difference is 0,20 kg.m[-2].decade[-1]. (b) Same as (a), but for the GPS data at 104 stations. The root mean square of the difference is 0,14 kg.m[-2].decade[-1].(c) Time series of IWV anomaly in ERA-Interim at a point of large difference between methods in the tropical Pacific ocean with superposed linear trends. (d) Time series of IWV anomaly at the COCO GPS site with superposed linear trends.**