# Peer review of "Global IWV trends and variability in atmospheric reanalyses and GPS observations"

_Atmospheric Chemistry and Physics, 2018_

## Referee Comment (RC1) · Anonymous Referee #1 · 30 Mar 2018

General Comments

The manuscript presents results for IWV mean values and linear trends over time periods of 15-20 years in a global perspective. As far as I know this paper is unique and it offers additional knowledge to the paper by Wang et al. (2016) which in some aspects is similar, such as the use of the GPS dataset from the IGS.

At places I think the manuscript reminds of a text book. For example, I think there is no need to explain the general distribution of the IWV, as in the beginning of Section 3, with mean values of the IWV for different areas. In general I find the text to use too many words. I will mention a couple of additional examples below. If the text is more focused on what is new it will be easier for the reader to grasp the important results, i.e. these that are new or are contradicting previously published results.

[Figure]

It is a bit disturbing that the time period with GPS data ends in 2010, covering a 15 year period, when we are now in 2018. I wonder if the additional 5-7 years of data would influence the overall results. Trends are small and a longer time period would be beneficial. Perhaps also the GPS stations have been more stable in terms of less changes of equipment during the recent years? Perhaps there is a another manuscript in preparation?

I note that the last section is called "Summary" rather than "Conclusions". Perhaps this is intentional? I miss conclusions in terms of what have you found that affects other studies? Do you have recommendations concerning the use of the different data sets? What are the future problems to address, et cetera? Given that the summary section to large extent repeats results already presented, it can be significantly shorter.

Specific comments

pages 2-4: The introduction could be much more focused. Many of the issues here are not addressed by the results presented later.

page 5, line 20: I do not see a problem of using all the 20 stations in the western USA, because you never present any global averages or trends. Please explain why overrepresentation would be an issue.

page 6, lines 8-11: I would like to see more strict requirements on the data availability in order to have representative values for a month and for a season. It is stated that GPS offers continuous data coverage back to 1995 so from that point of view it does not make sense to be so "generous" when accepting data.

page 6, subsection 2.3: This method used for the calculation of the trends has become popular recently. Here I miss one ore more quantitative example(s). All IWV results are presented as absolute (kg/mˆ2/decade) or relative (%/decade) trends for the IWV. Temperature trends are presented e.g. in Subsection 5.3. What are the corresponding resulting uncertainties obtained with this method compared to the classical least

squares fit? Furthermore, it would indeed be interesting (for at least a few sites) to present the differences between trends obtained by these two methods.

page 7, line 23: The words "good agreement" needs to be defined. Actually I think there is no information in this sentence. The quality of the agreement is discussed in detail thereafter and "good" means different things to different readers.

page 10, lines 3 and 6: again unclear, what is good agreement?

page 10, line s 14-16: This statement is not needed. It is just statistics (that are expected) and not used later.

page 11, lines 3-4: It is a bit unclear if all coastal and mountainous sites have a problem with representativeness? Otherwise is not this too much hand waving?

page 11, line 9: again the term "good agreement" is used. Try instead to describe quantitatively how they agree (and perhaps not agree? For example if a difference between two trends is 0.2 kg/m^2/decade, is that a good agreement or is it a disagreement?

page 11 line 11: "... none of the values computed by Wang et al (2016) are significant but the drying over western Australia is also observed." How is it observed when the values are not significant?

page 11, line 15: Here is an example where you refer to "low values". Does that mean that they are comparable to the uncertainties (in kg/m^2/decade, this connects to the comment on page 6)?

page 12, line 20: Here you define that "good agreement" means that some features are confirmed. However, you can just state that the features are confirmed and ignore to add the subjective wording "good agreement"

page 19, line 3: define "high IWV gradients"?

Technical Corrections

page 1, line 18: "Monthly IWV trends" is unclear, trends over a month or trends for all the January months et cetera, or (total) trends based on all monthly means?

page 1, line 25: "found to not" –> "found not to"?

page 1, line 30; and page 2, lines 3 and 4 (as well as many additional places in the manuscript): Leave a space between a value and its unit, according to SI also for the units percent (%), degree north and east (°N and °E), and degree centigrade (°C).

page 2, line 6: IPCC report is not in the reference list, specify which year and give an URL address?

page 2, line 11: Tropics –> tropics (not a name)

page 3, line 5: annual and –> annual, and

page 3, line 9: GPS data has –> GPS data have

page 3, line 11: data is –> data are

page 3, line 14: This data is –> These data are

page 3, line 15: ERA-Interim , –> ERA-Interim,

page 4, line 1: e.g. dynamical –> e.g. the dynamical

page 4, line 8: In section 5 – > In Section 5

page 5, line 23: rate –> temporal resolution (rate is not measured in minutes)

page 6, line 7: such a statement requires a reference, otherwise it should not be stated.

page 6, line 23: no need to repeat the requirement from above

page 8, line 6: section 2 –> Section 2

page 10, lines 5-6: well documented –> well sampled ?

page 11, line 1: section 5 –> Section 5

page 12, lines 18-19: "More details will be given in the discussion section." There is no section with the title "Discussion"?

page 12, line 28: In this section MERRA-2 is –> MERRA-2 is now?

page 13, line 31: (Fig. 9c and d) –> (Figs. 9c and d)

page 14, line 1: (Fig. 5c and d) –> (Figs. 5c and d)

page 14, line 2: (Fig. 9d and d) –> (Figs. 9c and d) ?

page 15, line 23: (Fig. 11d, h) –> (Figs. 11d and h)

page 15, line 23: (Fig. 11b, f) –> (Figs. 11b and f) and again two times on line 27, same page, and many times on pages 17 and 18 ...

page 18, line 22: the dry season flow –> the flow in the dry season

page 19, line 2: In this paper we –> We

page 19, lines 10-11: again, define "monthly trends"?

page 19, line 20: seasonal trends –> long term trends for winters and summers? (this is the same language issue as "monthly trends")

page 25: font size of station names is too small to be readable

pages 38 and 41: font size of the text within the figure frames is too small to be readable

page 41, fig. 16 –> Fig. 16

page 41, caption: siries –> series

— End of Comments

---

## Referee Comment (RC2) · Anonymous Referee #2 · 20 Apr 2018

The manuscript includes a lot of information. It (1) compares PW monthly and seasonal means, interannual variability, and linear trends between reanalyses and GPS data for 1995-2010, (2) studies PW trends for 1980-2016 using two RA products, (3) looks at the relationship between PW and surface temperature trends, and then (4) tries to link the dynamics with PW trends and variability. The authors have done a lot of work, but it is hard to figure out what the focus of this study is and what original results are achieved. In the major comments below, I raised several major concerns. Based on that, I think that the manuscript needs major revision or is resubmitted later. Major comments: 1. Scientific originality: The scientific originality first starts from the review of prior studies and the motivation of this study. As I mentioned below, some important references are missing in the introduction. Then the authors have to provide rationale

on why they want to study those four things (listed above). Have they never done before? Are your data better than that previous studies used? Are you going into more depth on those topics? I didn't see the strong motivation explained in the introduction. All those topics have been studies extensively before. What new and significant results does this study provide? The authors touched so many things, but didn't emphasize their originality. The authors try to describe all things they have done in a tedious way, so the manuscript looks more like a work summary, rather than synthesized scientific paper. I think that previous studies have done a lot for #1, #2, #3. Maybe the focus should be on briefly summarizing your results to establish the bases on using reanalysis data, and then on linking the dynamics with PW variability. For the first three, your results should be compared with previous studies, and then emphasize new results you found. 2. Technical quality: Again too much information is provided including too many topics, tedious descriptions of all results and too many plots. After you decide the focus, the manuscript should be reorganized and be shortened. 3. References: The manuscript didn't include some of relevant references. I mentioned this in several places in Specific comments. You can find a lot of references from the review paper by Guerrova et al (2016, Review of the state of the art and future prospects of the ground-based GNSS meteorology in Europe, AMT).

Specific comments: 1. GPS vs. GNSS: I would suggest that GNSS is used instead of GPS. 2. P1, L13: Do gaps affect monthly mean if they last longer? 3. P1, L24: What temperature? Surface or upper air? 4. Abstract: I didn't learn a lot anything new from this. 5. P2, L5-8: Lots of papers have discussed this. Please list some references. 6. P3, L10, Fig. 1: This map only shows 104 stations, much less than thousands of available stations. It is not convincing about "growing". I understand that those are the stations used in this study. But it is not convincing to use this to make your point here. 7. P3, L28: There have been quite a few studies using GNSS PW to evaluate reanalysis products. First of all, you should summarize the prior studies on this, and then describe what is different (unique) about this study. 8. P5, L1: Should you just simply average the surrounding grid points or have more complicated regression? It

depends on the location and topography. See Fig. 1 in Mears, C., J. Wang, D. Smith, and F. J. Wentz, 2015: Intercomparison of total precipitable water measurements made by satelliteborne microwave radiometers and ground-based GPS instruments. J. Geophys. Res. Atmos., 120, 2492–2504, doi:10.1002/2014JD022694. 9. P7, L10, the standard deviation is calculated from seasonal mean values. Is it right? In other words, only 16 data points are used to calculate standard deviation? Is the standard deviation statistically significant given only 16 data points? 10. Fig. 5: The trends are calculated using monthly PW anomaly, not monthly mean, correct? Fig. 5 is a similar plot as Fig. 4 in Wang et al. (2016). You need to discuss how your results compare with Wang et al. (2016) here. 11. P10, L33-34: For WUHN, the big change in Oct 2016 is due to the radiosonde type change from Shang-M to Shang-E. Radiosonde data over land are the main source of upper air humidity for reanalyses, so any inhomogeneity in radiosonde data would be reflected in the reanalysis data.

---

## Author Comment (AC1) · 31 May 2018

The referee's comments are presented followed by our responses in blue.

Anonymous Referee #1

General Comments

The manuscript presents results for IWV mean values and linear trends over time periods of 15-20 years in a global perspective. As far as I know this paper is unique and it offers additional knowledge to the paper by Wang et al. (2016) which in some aspects is similar, such as the use of the GPS dataset from the IGS. At places I think the manuscript reminds of a text book. For example, I think there is no need to explain the general distribution of the IWV, as in the beginning of Section 3, with mean values of the IWV for different areas. In general, I find the text to use too many words. I will mention a couple of additional examples below. If the text is more focused on what is new it will be easier for the reader to grasp the important results, i.e. these that are new or are contradicting previously published results. It is a bit disturbing that the time period with GPS data ends in 2010, covering a 15 year period, when we are now in 2018. I wonder if the additional 5-7 years of data would influence the overall results. Trends are small and a longer time period would be beneficial. Perhaps also the GPS stations have been more stable in terms of less changes of equipment during the recent years? Perhaps there is a another manuscript in preparation?

I note that the last section is called "Summary" rather than "Conclusions". Perhaps this is intentional? I miss conclusions in terms of what have you found that affects other studies? Do you have recommendations concerning the use of the different data sets? What are the future problems to address, et cetera? Given that the summary section to large extent repeats results already presented, it can be significantly shorter.

>> We thank the referee for the constructive comments that helped improving the manuscript. The referee is right in pointing that this work is in some aspects similar to Wang et al., 2016, but also that it is unique and offers a great deal of additional knowledge, such as separating the analysis in seasons (DJF and JJA), which is important to help understanding the underlying climate processes, and also evaluating modern reanalysis, which are widely used by the climate community.

We took the general comments into account and tried to make the manuscript shorter overall. Specifically, the general description in Section 3 was almost completely removed and the Introduction and Conclusion sections were largely rewritten (according the specific comments given below). The introduction focuses more directly on the topics addressed later in the paper, and the conclusions state more clearly what is new and what are future problems to address.

Regarding the comment about the period used in this study, the reason why it stops in 2010 is that it uses the IGS repro1 dataset (the same Wang et al. 2016 used, with slight differences in the post-processing as explained in Section 2). This is the only official reprocessed IGS tropospheric dataset available to date. Tropospheric data from a 2nd more recent reprocessing which goes until end of 2013/2014 are only available from individual analysis centers and are not qualified so far. They might be evaluated in a future study and namely require to check the homogeneity of the more recent years. However, in order to solve the new homogeneity issues highlighted in our study, a complete reprocessing of all stations would be preferable.

If the extra 5-7 years were included the results might change slightly for two reasons. First, additional inhomogeneities are expected both in the GPS data (e.g. due to recent equipment changes) and in the reanalyses (due to recent observing system changes). Second, we know that the trends are not exactly linear, hence the period of study has an impact on the trends computed. That is why we are careful to state the trends are for 1995-2010.

The GPS stations have undergone systematic equipment changes in recent years to adapt to the new GNSS signals from Galileo, Beidou, etc. These changes are not expected to have as dramatic impact as

observed with some older equipment as more accurate processing models are available but the exact impact on IWV trends from these data has not yet been evaluated.

To be noticed also is the new figure 3 presenting differences in the means and interannual variability between ERA-Interim and MERRA-2. In the first submission only trend results were shown for MERRA-2. We added this figure for the sake of completeness and higher consistency in the discussion of both reanalyses. It allows us to draw more general conclusions on the performance of modern reanalyses.

Specific comments

pages 2-4: The introduction could be much more focused. Many of the issues here are not addressed by the results presented later.

>> The introduction has been in large part re-written.

page 5, line 20: I do not see a problem of using all the 20 stations in the western USA, because you never present any global averages or trends. Please explain why overrepresentation would be an issue.

>> We do present global statistics (i.e. including all stations) for the means and standard deviations of IWV differences in Table 1.

In addition, it would be impossible to distinguish between points when all 20 stations are included in the maps, so it is not useful to retain all stations.

page 6, lines 8-11: I would like to see stricter requirements on the data availability in order to have representative values for a month and for a season. It is stated that GPS offers continuous data coverage back to 1995 so from that point of view it does not make sense to be so "generous" when accepting data.

>> The two reasons for adopting these criteria were the following:

1) GPS is not really continuous (it contains gaps in the time series), so we need to apply a selection.

2) However we do not want to eliminate too many months, because when we compare with ERA-Interim, in the maps, they are not time-matched, so the elimination of too many months would impede their comparison.

To avoid overstatement, we removed the word "continuous" from the description of the GPS data.

page 6, subsection 2.3: This method used for the calculation of the trends has become popular recently. Here I miss one or more quantitative example(s). All IWV results are presented as absolute (kg/m^2/decade) or relative (%/decade) trends for the IWV. Temperature trends are presented e.g. in Subsection 5.3. What are the corresponding resulting uncertainties obtained with this method compared to the classical least squares fit? Furthermore, it would indeed be interesting (for at least a few sites) to present the differences between trends obtained by these two methods.

>> Before deciding to use the Theil-Sen method to compute the IWV trends, we compared it to the Least Square method. Both methods were applied to the monthly mean ERA-Interim and GPS IWV data, and the differences between the trends were plotted and are presented in an Appendix that was added to the paper.

page 7, line 23: The words "good agreement" needs to be defined. Actually I think there is no information in this sentence. The quality of the agreement is discussed in detail thereafter and "good" means different things to different readers.

>> The entire sentence here was removed.

page 10, lines 3 and 6: again unclear, what is good agreement?

>> Same sign, the text has been corrected to make it clearer: *"In general, the monthly trends computed at the GPS stations are consistent in sign and magnitude with ERA-Interim (Fig. 6)."*

page 10, lines 14-16: This statement is not needed. It is just statistics (that are expected) and not used later.

>> The statement was removed.

page 11, lines 3-4: It is a bit unclear if all coastal and mountainous sites have a problem with representativeness? Otherwise is not this too much hand waving?

>> Not all, the sentence was corrected: *"Representativeness differences are suspected at some mountainous and coastal sites (e.g. AREQ, CFAG, KIT3, MAW1, SANT, SYOG and the other sites discussed in the previous section)."*

page 11, line 9: again the term "good agreement" is used. Try instead to describe quantitatively how they agree (and perhaps not agree? For example if a difference between two trends is 0.2 kg/m^2/decade, is that a good agreement or is it a disagreement?

>> The sentence was changed: *"Over the oceans, results that are significant in ERA-Interim are consistent (i.e. same sign) with those obtained by Wang et al. (2016), despite the fact that they are not always significant over land in the latter study."*

page 11 line 11: "... none of the values computed by Wang et al (2016) are significant but the drying over western Australia is also observed." How is it observed when the values are not significant?

>> This sentence was removed.

page 11, line 17: Here is an example where you refer to "low values". Does that mean that they are comparable to the uncertainties (in kg/m^2/decade, this connects to the comment on page 6)?

>> This part of the sentence was removed. In fact, only the absolute trends are low (because IWV has low values in this region), the relative trends are not small.

page 12, line 20: Here you define that "good agreement" means that some features are confirmed. However, you can just state that the features are confirmed and ignore to add the subjective wording "good agreement"

>> Sentence has been re-written to reflect this: *"Overall, the seasonal trends estimated from the GPS data confirm the features discussed above for ERA-Interim."*

page 19, line 3: define "high IWV gradients"?

>> High IWV gradients are regions where IWV varies strongly (e.g. around the ITCZ, mountain regions). This definition was added to the sentence.

Technical Corrections

>> All the corrections suggested by the referee have been implemented.

page 1, line 18: "Monthly IWV trends" is unclear, trends over a month or trends for all the January months et cetera, or (total) trends based on all monthly means?

>> The abstract has been rewritten, but later in Section 4, it is specified that trends computed from the time series of monthly means are referred to as monthly trends.

page 1, line 25: "found to not" –> "found not to"?

page 1, line 30; and page 2, lines 3 and 4 (as well as many additional places in the manuscript): Leave a space between a value and its unit, according to SI also for the units percent (%), degree north and east (∘N and∘E), and degree centigrade (∘C).

page 2, line 6: IPCC report is not in the reference list, specify which year and give an URL address? => no longer referenced

page 2, line 11: Tropics –> tropics (not a name)

page 3, line 5: annual and –> annual, and

page 3, line 9: GPS data has –> GPS data have

page 3, line 11: data is –> data are

page 3, line 14: This data is –> These data are

page 3, line 15: ERA-Interim , –> ERA-Interim,

page 4, line 1: e.g. dynamical –> e.g. the dynamical

page 4, line 8: In section 5 – > In Section 5

page 5, line 23: rate –> temporal resolution (rate is not measured in minutes)

page 6, line 7: such a statement requires a reference, otherwise it should not be stated.

page 6, line 23: no need to repeat the requirement from above

page 8, line 6: section 2 –> Section 2

page 10, lines 5-6: well documented –> well sampled ?

page 11, line 1: section 5 –> Section 5

page 12, lines 18-19: "More details will be given in the discussion section." There is no section with the title "Discussion"?

>> Summary section has been replaced with a "Summary and conclusions" section and the text has been corrected.

page 12, line 28: In this section MERRA-2 is –> MERRA-2 is now?

page 13, line 31: (Fig. 9c and d) –> (Figs. 9c and d)

page 14, line 1: (Fig. 5c and d) –> (Figs. 5c and d)

page 14, line 2: (Fig. 9d and d) –> (Figs. 9c and d) ?

page 15, line 23: (Fig. 11d, h) –> (Figs. 11d and h)

page 15, line 23: (Fig. 11b, f) –> (Figs. 11b and f) and again two times on line 27, same page, and many times on pages 17 and 18 ...

page 18, line 22: the dry season flow –> the flow in the dry season

page 19, line 2: In this paper we –> We

page 19, lines 10-11: again, define "monthly trends"?

page 19, line 20: seasonal trends –> long term trends for winters and summers? (this is the same language issue as "monthly trends")

page 25: font size of station names is too small to be readable

pages 38 and 41: font size of the text within the figure frames is too small to be readable

page 41, fig. 16 –> Fig. 16

page 41, caption: siries –> series

Anonymous Referee #2

The manuscript includes a lot of information. It (1) compares PW monthly and seasonal means, interannual variability, and linear trends between reanalyses and GPS data for 1995-2010, (2) studies PW trends for 1980-2016 using two RA products, (3) looks at the relationship between PW and surface temperature trends, and then (4) tries to link the dynamics with PW trends and variability. The authors have done a lot of work, but it is hard to figure out what the focus of this study is and what original results are achieved. In the major comments below, I raised several major concerns. Based on that, I think that the manuscript needs major revision or is resubmitted later.

>> We thank the reviewer for the comments which we tried to implement in the revised manuscript. The manuscript has been revised throughout to highlight what is new and original in our results and the conclusion section has been rewritten. The introduction has also been re-written and refocused on the main questions addressed in the manuscript, including additional references.

Major comments:

1. Scientific originality: The scientific originality first starts from the review of prior studies and the motivation of this study. As I mentioned below, some important references are missing in the introduction. Then the authors have to provide rationale on why they want to study those four things (listed above). Have they never done before? Are your data better than that previous studies used? Are you going into more depth on those topics? I didn't see the strong motivation explained in the introduction. All those topics have been studies extensively before. What new and significant results does this study provide? The authors touched so many things, but didn't emphasize their originality. The authors try to describe all things they have done in a tedious way, so the manuscript looks more like a work summary, rather than synthesized scientific paper. I think that previous studies have done a lot for #1, #2, #3. Maybe the focus should be on briefly summarizing your results to establish the bases on using reanalysis data, and then on linking the dynamics with PW variability. For the first three, your results should be compared with previous studies, and then emphasize new results you found.

>> The introduction has been re-written and additional references have been included. We have emphasized our motivation and explained why we use these IWV datasets, why we focus on means, variability, and trends, and how our approach and data differ or complement previous studies. We have also tried to shorten our descriptions of the results (e.g. the description of the mean IWV has been squeezed, as also required by the first reviewer), and compared our results to previous studies when possible. Finally, we have rewritten the last section to include not only a summary of results but also a discussion of the conclusions, in terms of what is new and what future work should be done on this topic.

To answer more specifically the questions posed by the reviewer, only few studies investigated the means, variability, and trends in IWV and their link with surface air temperature. Major deficiencies in older reanalyses and observations-based datasets have been evidenced in past studies. Compared to those, we used modern reanalyses (ERA-Interim and, especially the very recent MERRA-2) and a new independent reprocessed 16-yr long ground-based GPS dataset to validate them. To our knowledge it is the first time reprocessed GPS IWV data are used to validate global reanalyses over such extended periods. We showed that the modern reanalyses still suffer from uncertainties in data sparse regions such

as Africa and Antarctica, and that IWV trends and variability at regional scale are dominated by changes in atmospheric circulation rather than surface temperature variations predicted by Clausius-Clapeyron.

2. Technical quality: Again too much information is provided including too many topics, tedious descriptions of all results and too many plots. After you decide the focus, the manuscript should be reorganized and be shortened.

>> The rationale has been sharpened and text shortened. More specific comments would help.

3. References:

The manuscript didn't include some of relevant references. I mentioned this in several places in Specific comments. You can find a lot of references from the review paper by Guerrova et al (2016, Review of the state of the art and future prospects of the ground-based GNSS meteorology in Europe, AMT).

>> New references have been added to the paper:

Dee, D. P. and Uppala, S.: Variational bias correction of satellite radiance data in the ERA-Interim reanalysis. Q.J.R. Meteorol. Soc., 135: 1830-1841. doi:10.1002/qj.493, 2009.

Mears, C. A., Wang, J., Smith, D., and Wentz, F. J.: Intercomparison of total precipitable water measurements made by satellite-borne microwave radiometers and ground-based GPS instruments. J. Geophys. Res. Atmos., 120, 2492–2504. doi: 10.1002/2014JD022694, 2015.

Meynadier, R., Bock, O., Gervois, S., Guichard, F., Redelsperger, J.-L., Agustí-Panareda, A., and Beljaars, A.: West African Monsoon water cycle: 2. Assessment of numerical weather prediction water budgets, J. Geophys. Res., 115, D19107, doi:10.1029/2010JD013919, 2010.

O'Gorman, P. A. and Muller, C. J.: How closely do changes in surface and column water vapor follow Clausius–Clapeyron scaling in climate-change simulations? Environ. Res. Lett. 5, 025207, 2010.

Robertson, F.R., Bosilovich, M.G., Roberts, J.B., Reichle, R.H., Adler, R., Ricciardulli, L., Berg, W., and Huffman, G.J.: Consistency of Estimated Global Water Cycle Variations over the Satellite Era. J. Climate, 27, 6135–6154, https://doi.org/10.1175/JCLI-D-13-00384.1, 2014.

Schneider, T., O'Gorman, P. A., and Levine, X. J.: Water vapor and the dynamics of climate changes, Rev. Geophys., 48, RG3001, doi:10.1029/2009RG000302, 2010.

Sherwood, S. C., Roca, R., Weckwerth, T. M., and Andronova, N. G.: Tropospheric water vapor, convection, and climate, Rev. Geophys., 48, RG2001, doi:10.1029/2009RG000301, 2010.

Thorne, P.W. and Vose, R.S.: Reanalyses Suitable for Characterizing Long-Term Trends. Bull. Amer. Meteor. Soc., 91, 353–362, https://doi.org/10.1175/2009BAMS2858.1, 2010.

Trenberth, K.E., J.T. Fasullo, and J. Mackaro, 2011: Atmospheric Moisture Transports from Ocean to Land and Global Energy Flows in Reanalyses. J. Climate, 24, 4907–4924, https://doi.org/10.1175/2011JCLI4171.1

Wang, J., and Zhang, L.: Systematic errors in global radiosonde precipitable water data from comparisons with ground-based GPS measurements. J. Climate, 21(10), 2218-2238, https://doi.org/10.1175/2007JCLI1944.1, 2008.

Zhao, T., Dai, A., and Wang, J.: Trends in tropospheric humidity from 1970 to 2008 over China from a homogenized radiosonde dataset J. Climate, 25(13), 4549-4567, https://doi.org/10.1175/JCLI-D-11-00557.1, 2012.

Specific comments:

1. GPS vs. GNSS: I would suggest that GNSS is used instead of GPS.

>> In this study we use a GPS-only solution, so it is correct to refer to it as GPS.

2. P1, L13: Do gaps affect monthly mean if they last longer?

>> If the reviewer means if gaps in the 6-hourly GPS IWV time series affect the monthly means: yes, and the impact is larger if the gaps last longer, that is why we reject months which have less than 60 values (half of the expected values). This ensures that we obtain representative monthly means, while not eliminating too many data (necessary for the comparison with ERA-Interim monthly mean fields like in Fig. 2, which do not have gaps).

3. P1, L24: What temperature? Surface or upper air?

>> The surface air temperature (2 m) was used. This information has been added to the abstract.

4. Abstract: I didn't learn a lot anything new from this.

>> Thanks for the notice. We hope that the revised abstract will be more instructive.

5. P2, L5-8: Lots of papers have discussed this. Please list some references.

>> Several references have been added.

6. P3, L10, Fig. 1: This map only shows 104 stations, much less than thousands of available stations. It is not convincing about "growing". I understand that those are the stations used in this study. But it is not convincing to use this to make your point here.

>> The term "growing" has been removed from the text. Note that even if thousands of stations are operated routinely, not all are publicly available and/or of scientific quality. Here we consider the IGS network which is a freely available global scientific network counting nominally about 400 stations.

7. P3, L28: There have been quite a few studies using GNSS PW to evaluate reanalysis products. First of all, you should summarize the prior studies on this, and then describe what is different (unique) about this study.

>> We don't think a summary of all prior studies is useful here because there is quite poor agreement between them due to the study of different periods and regions, and use of different datasets with various issues (e.g. older reanalyses had time-varying biases due to observing system changes and incorrect assimilation of recent satellite data, and most observational datasets did not use reprocessed and homogenized data). Nevertheless, we cite a few studies dealing with the same scope as ours and discussing these issues. This leads us to the main objectives and novelties of this study which are the assessment of modern reanalyses and the use reprocessed GPS data. These points have been made more clear in the revised introduction.

8. P5, L1: Should you just simply average the surrounding grid points or have more complicated regression? It depends on the location and topography. See Fig. 1 in Mears, C., J. Wang, D. Smith, and F. J. Wentz, 2015: Intercomparison of total precipitable water measurements made by satelliteborne microwave radiometers and ground-based GPS instruments. J. Geo-phys. Res. Atmos., 120, 2492–2504, doi:10.1002/2014JD022694.

>> The method we used is not simply an average of the surrounding grid points. As stated:

*"A bilinear spatial interpolation is computed from the model IWV estimates at the 4 grid points surrounding each GPS station. The IWV model estimates are then recomputed from the pressure level data by vertically integrating the specific humidity between the height of the GPS station and the top of the atmosphere. Most GPS station heights fall between two pressure levels and the specific humidity data can be interpolated. However, for stations located below 1000 hPa (the lowest pressure level) the reanalysis data are extrapolated. Interpolation and extrapolation are done linearly for specific humidity and temperature, and exponentially for pressure."*

9. P7, L10, the standard deviation is calculated from seasonal mean values. Is it right? In other words, only 16 data points are used to calculate standard deviation? Is the standard deviation statistically significant given only 16 data points?

>> Yes. The two-sample F-test was used to determine whether the difference in variances between GPS and ERA-Interim were statistically significant.

10. Fig. 5: The trends are calculated using monthly PW anomaly, not monthly mean, correct? Fig. 5 is a similar plot as Fig. 4 in Wang et al. (2016). You need to discuss how your results compare with Wang et al. (2016) here.

>> Yes, the trends are calculated using monthly PW anomaly, as described in Section 2.3: *"The Theil-Sen method was applied to the anomalies obtained by removing the monthly climatology from the monthly data."*

The results of Fig. 5 were compared with Wang et al. (2016) a little further ahead in the text, so as not to break the discussion of the agreement between GPS and ERA-Interim. We decided to keep the organization this way in the revised paper.

11. P10, L33-34: For WUHN, the big change in Oct 2006 is due to the radiosonde type change from Shang-M to Shang-E. Radiosonde data over land are the main source of upper air humidity for reanalyses, so any inhomogeneity in radiosonde data would be reflected in the reanalysis data.

>> Thank you for this information. It has been added to the manuscript: *"In fact, the break at the end of 2006 is associated with the change in radiosonde from the Shang-M to Shang-E, which is assimilated by ERA-Interim (Wang and Zhang, 2008). Zhao et al (2012) found that prior to this change there was a 2 kg.m-2 wet bias in the radiosonde data at the Wuhan station, in comparison with GPS. This moist bias is also observed in ERA-Interim prior to the end of 2006."*

---

## Author Response (AR1)

The referee's comments are presented followed by our responses in blue.

Anonymous Referee #1

General Comments

The manuscript presents results for IWV mean values and linear trends over time periods of 15-20 years in a global perspective. As far as I know this paper is unique and it offers additional knowledge to the paper by Wang et al. (2016) which in some aspects is similar, such as the use of the GPS dataset from the IGS. At places I think the manuscript reminds of a text book. For example, I think there is no need to explain the general distribution of the IWV, as in the beginning of Section 3, with mean values of the IWV for different areas. In general, I find the text to use too many words. I will mention a couple of additional examples below. If the text is more focused on what is new it will be easier for the reader to grasp the important results, i.e. these that are new or are contradicting previously published results. It is a bit disturbing that the time period with GPS data ends in 2010, covering a 15 year period, when we are now in 2018. I wonder if the additional 5-7 years of data would influence the overall results. Trends are small and a longer time period would be beneficial. Perhaps also the GPS stations have been more stable in terms of less changes of equipment during the recent years? Perhaps there is a another manuscript in preparation?

I note that the last section is called "Summary" rather than "Conclusions". Perhaps this is intentional? I miss conclusions in terms of what have you found that affects other studies? Do you have recommendations concerning the use of the different data sets? What are the future problems to address, et cetera? Given that the summary section to large extent repeats results already presented, it can be significantly shorter.

>> We thank the referee for the constructive comments that helped improving the manuscript. The referee is right in pointing that this work is in some aspects similar to Wang et al., 2016, but also that it is unique and offers a great deal of additional knowledge, such as separating the analysis in seasons (DJF and JJA), which is important to help understanding the underlying climate processes, and also evaluating modern reanalysis, which are widely used by the climate community.

We took the general comments into account and tried to make the manuscript shorter overall. Specifically, the general description in Section 3 was almost completely removed and the Introduction and Conclusion sections were largely rewritten (according the specific comments given below). The introduction focuses more directly on the topics addressed later in the paper, and the conclusions state more clearly what is new and what are future problems to address.

Regarding the comment about the period used in this study, the reason why it stops in 2010 is that it uses the IGS repro1 dataset (the same Wang et al. 2016 used, with slight differences in the post-processing as explained in Section 2). This is the only official reprocessed IGS tropospheric dataset available to date. Tropospheric data from a 2nd more recent reprocessing which goes until end of 2013/2014 are only available from individual analysis centers and are not qualified so far. They might be evaluated in a future study and namely require to check the homogeneity of the more recent years. However, in order to solve the new homogeneity issues highlighted in our study, a complete reprocessing of all stations would be preferable.

If the extra 5-7 years were included the results might change slightly for two reasons. First, additional inhomogeneities are expected both in the GPS data (e.g. due to recent equipment changes) and in the reanalyses (due to recent observing system changes). Second, we know that the trends are not exactly linear, hence the period of study has an impact on the trends computed. That is why we are careful to state the trends are for 1995-2010.

The GPS stations have undergone systematic equipment changes in recent years to adapt to the new GNSS signals from Galileo, Beidou, etc. These changes are not expected to have as dramatic impact as

observed with some older equipment as more accurate processing models are available but the exact impact on IWV trends from these data has not yet been evaluated.

To be noticed also is the new figure 3 presenting differences in the means and interannual variability between ERA-Interim and MERRA-2. In the first submission only trend results were shown for MERRA-2. We added this figure for the sake of completeness and higher consistency in the discussion of both reanalyses. It allows us to draw more general conclusions on the performance of modern reanalyses.

Specific comments

pages 2-4: The introduction could be much more focused. Many of the issues here are not addressed by the results presented later.

>> The introduction has been in large part re-written.

page 5, line 20: I do not see a problem of using all the 20 stations in the western USA, because you never present any global averages or trends. Please explain why overrepresentation would be an issue.

>> We do present global statistics (i.e. including all stations) for the means and standard deviations of IWV differences in Table 1.

In addition, it would be impossible to distinguish between points when all 20 stations are included in the maps, so it is not useful to retain all stations.

page 6, lines 8-11: I would like to see stricter requirements on the data availability in order to have representative values for a month and for a season. It is stated that GPS offers continuous data coverage back to 1995 so from that point of view it does not make sense to be so "generous" when accepting data.

>> The two reasons for adopting these criteria were the following:

1) GPS is not really continuous (it contains gaps in the time series), so we need to apply a selection.

2) However we do not want to eliminate too many months, because when we compare with ERA-Interim, in the maps, they are not time-matched, so the elimination of too many months would impede their comparison.

To avoid overstatement, we removed the word "continuous" from the description of the GPS data.

page 6, subsection 2.3: This method used for the calculation of the trends has become popular recently. Here I miss one or more quantitative example(s). All IWV results are presented as absolute (kg/m^2/decade) or relative (%/decade) trends for the IWV. Temperature trends are presented e.g. in Subsection 5.3. What are the corresponding resulting uncertainties obtained with this method compared to the classical least squares fit? Furthermore, it would indeed be interesting (for at least a few sites) to present the differences between trends obtained by these two methods.

>> Before deciding to use the Theil-Sen method to compute the IWV trends, we compared it to the Least Square method. Both methods were applied to the monthly mean ERA-Interim and GPS IWV data, and the differences between the trends were plotted and are presented in an Appendix that was added to the paper.

page 7, line 23: The words "good agreement" needs to be defined. Actually I think there is no information in this sentence. The quality of the agreement is discussed in detail thereafter and "good" means different things to different readers.

>> The entire sentence here was removed.

page 10, lines 3 and 6: again unclear, what is good agreement?

>> Same sign, the text has been corrected to make it clearer: *"In general, the monthly trends computed at the GPS stations are consistent in sign and magnitude with ERA-Interim (Fig. 6)."*

page 10, lines 14-16: This statement is not needed. It is just statistics (that are expected) and not used later.

>> The statement was removed.

page 11, lines 3-4: It is a bit unclear if all coastal and mountainous sites have a problem with representativeness? Otherwise is not this too much hand waving?

>> Not all, the sentence was corrected: *"Representativeness differences are suspected at some mountainous and coastal sites (e.g. AREQ, CFAG, KIT3, MAW1, SANT, SYOG and the other sites discussed in the previous section)."*

page 11, line 9: again the term "good agreement" is used. Try instead to describe quantitatively how they agree (and perhaps not agree? For example if a difference between two trends is 0.2 kg/m^2/decade, is that a good agreement or is it a disagreement?

>> The sentence was changed: *"Over the oceans, results that are significant in ERA-Interim are consistent (i.e. same sign) with those obtained by Wang et al. (2016), despite the fact that they are not always significant over land in the latter study."*

page 11 line 11: "... none of the values computed by Wang et al (2016) are significant but the drying over western Australia is also observed." How is it observed when the values are not significant?

>> This sentence was removed.

page 11, line 17: Here is an example where you refer to "low values". Does that mean that they are comparable to the uncertainties (in kg/m^2/decade, this connects to the comment on page 6)?

>> This part of the sentence was removed. In fact, only the absolute trends are low (because IWV has low values in this region), the relative trends are not small.

page 12, line 20: Here you define that "good agreement" means that some features are confirmed. However, you can just state that the features are confirmed and ignore to add the subjective wording "good agreement"

>> Sentence has been re-written to reflect this: *"Overall, the seasonal trends estimated from the GPS data confirm the features discussed above for ERA-Interim."*

page 19, line 3: define "high IWV gradients"?

>> High IWV gradients are regions where IWV varies strongly (e.g. around the ITCZ, mountain regions). This definition was added to the sentence.

Technical Corrections

>> All the corrections suggested by the referee have been implemented.

page 1, line 18: "Monthly IWV trends" is unclear, trends over a month or trends for all the January months et cetera, or (total) trends based on all monthly means?

>> The abstract has been rewritten, but later in Section 4, it is specified that trends computed from the time series of monthly means are referred to as monthly trends.

page 1, line 25: "found to not" –> "found not to"?

page 1, line 30; and page 2, lines 3 and 4 (as well as many additional places in the manuscript): Leave a space between a value and its unit, according to SI also for the units percent (%), degree north and east (∘N and∘E), and degree centigrade (∘C).

page 2, line 6: IPCC report is not in the reference list, specify which year and give an URL address? => no longer referenced

page 2, line 11: Tropics –> tropics (not a name)

page 3, line 5: annual and –> annual, and

page 3, line 9: GPS data has –> GPS data have

page 3, line 11: data is –> data are

page 3, line 14: This data is –> These data are

page 3, line 15: ERA-Interim , –> ERA-Interim,

page 4, line 1: e.g. dynamical –> e.g. the dynamical

page 4, line 8: In section 5 – > In Section 5

page 5, line 23: rate –> temporal resolution (rate is not measured in minutes)

page 6, line 7: such a statement requires a reference, otherwise it should not be stated.

page 6, line 23: no need to repeat the requirement from above

page 8, line 6: section 2 –> Section 2

page 10, lines 5-6: well documented –> well sampled ?

page 11, line 1: section 5 –> Section 5

page 12, lines 18-19: "More details will be given in the discussion section." There is no section with the title "Discussion"?

>> Summary section has been replaced with a "Summary and conclusions" section and the text has been corrected.

page 12, line 28: In this section MERRA-2 is –> MERRA-2 is now?

page 13, line 31: (Fig. 9c and d) –> (Figs. 9c and d)

page 14, line 1: (Fig. 5c and d) –> (Figs. 5c and d)

page 14, line 2: (Fig. 9d and d) –> (Figs. 9c and d) ?

page 15, line 23: (Fig. 11d, h) –> (Figs. 11d and h)

page 15, line 23: (Fig. 11b, f) –> (Figs. 11b and f) and again two times on line 27, same page, and many times on pages 17 and 18 ...

page 18, line 22: the dry season flow –> the flow in the dry season

page 19, line 2: In this paper we –> We

page 19, lines 10-11: again, define "monthly trends"?

page 19, line 20: seasonal trends –> long term trends for winters and summers? (this is the same language issue as "monthly trends")

page 25: font size of station names is too small to be readable

pages 38 and 41: font size of the text within the figure frames is too small to be readable

page 41, fig. 16 –> Fig. 16

page 41, caption: siries –> series

Anonymous Referee #2

The manuscript includes a lot of information. It (1) compares PW monthly and seasonal means, interannual variability, and linear trends between reanalyses and GPS data for 1995-2010, (2) studies PW trends for 1980-2016 using two RA products, (3) looks at the relationship between PW and surface temperature trends, and then (4) tries to link the dynamics with PW trends and variability. The authors have done a lot of work, but it is hard to figure out what the focus of this study is and what original results are achieved. In the major comments below, I raised several major concerns. Based on that, I think that the manuscript needs major revision or is resubmitted later.

>> We thank the reviewer for the comments which we tried to implement in the revised manuscript. The manuscript has been revised throughout to highlight what is new and original in our results and the conclusion section has been rewritten. The introduction has also been re-written and refocused on the main questions addressed in the manuscript, including additional references.

Major comments:

1. Scientific originality: The scientific originality first starts from the review of prior studies and the motivation of this study. As I mentioned below, some important references are missing in the introduction. Then the authors have to provide rationale on why they want to study those four things (listed above). Have they never done before? Are your data better than that previous studies used? Are you going into more depth on those topics? I didn't see the strong motivation explained in the introduction. All those topics have been studies extensively before. What new and significant results does this study provide? The authors touched so many things, but didn't emphasize their originality. The authors try to describe all things they have done in a tedious way, so the manuscript looks more like a work summary, rather than synthesized scientific paper. I think that previous studies have done a lot for #1, #2, #3. Maybe the focus should be on briefly summarizing your results to establish the bases on using reanalysis data, and then on linking the dynamics with PW variability. For the first three, your results should be compared with previous studies, and then emphasize new results you found.

>> The introduction has been re-written and additional references have been included. We have emphasized our motivation and explained why we use these IWV datasets, why we focus on means, variability, and trends, and how our approach and data differ or complement previous studies. We have also tried to shorten our descriptions of the results (e.g. the description of the mean IWV has been squeezed, as also required by the first reviewer), and compared our results to previous studies when possible. Finally, we have rewritten the last section to include not only a summary of results but also a discussion of the conclusions, in terms of what is new and what future work should be done on this topic.

To answer more specifically the questions posed by the reviewer, only few studies investigated the means, variability, and trends in IWV and their link with surface air temperature. Major deficiencies in older reanalyses and observations-based datasets have been evidenced in past studies. Compared to those, we used modern reanalyses (ERA-Interim and, especially the very recent MERRA-2) and a new independent reprocessed 16-yr long ground-based GPS dataset to validate them. To our knowledge it is the first time reprocessed GPS IWV data are used to validate global reanalyses over such extended periods. We showed that the modern reanalyses still suffer from uncertainties in data sparse regions such as Africa and Antarctica, and that IWV trends and variability at regional scale are dominated by changes in atmospheric circulation rather than surface temperature variations predicted by Clausius-Clapeyron.

2. Technical quality: Again too much information is provided including too many topics, tedious descriptions of all results and too many plots. After you decide the focus, the manuscript should be reorganized and be shortened.

>> The rationale has been sharpened and text shortened. More specific comments would help.

*Correspondence to*: Ana C. Parracho (ana.parracho@etu.upmc.fr)

**Abstract.**

10  ~~of water vapour data. In this work we use Integrated Water Vapour (IWV) estimated from GPS observations and atmospheric reanalyses. Annual and seasonal means, interannual variability, and linear trends are analysed and compared for the period between 1995 and 2010. A general good agreement is found but this study highlights issues in both GPS and reanalysis data sets. In GPS, gaps and inhomogeneities in the time series are evidenced, which affect mainly variability and trend estimation. In ERA Interim, too strong trends in certain regions (e.g. drying over northern Africa and Australia, and moistening over~~

15  ~~northern South America) were found. Representativeness differences in coastal areas and regions of complex topography (mountain ranges, islands) are also evidenced as limitations to the intercomparison of the point observations and reanalysis data. A general good agreement is found for the means and variabilities, with the exception of a few stations where representativeness issues are suspected. Annual IWV trends are also found to be in good sign agreement, with the exception of a handful of stations where, in addition to representativeness errors, there might be inhomogeneities in the GPS time series.~~

20  ~~Seasonal trends are found to be different and more intense than annual trends, which emphasizes the influence of atmospheric circulation on IWV trends. In order to assess strong trends over regions lacking GPS stations, a second reanalysis, MERRA 2, is introduced. The period of analysis is extended to 1980 2016 (the longest period the reanalyses have in common) and differences with the shorter period are found. This exemplifies how much IWV trends are dependent on the time period at study and must be interpreted carefully. Temperature trends are also computed for both reanalyses. The Clausius Clapeyron~~

25

This study investigates the means, variability, and trends in Integrated Water Vapour (IWV) from two modern reanalyses (ERA-Interim and MERRA-2) from 1980 to 2016 and ground-based GPS data from 1995 to 2010. It is found that the mean

30  distributions and inter-annual variability in IWV in the reanalyses and GPS are consistent, even in regions of strong gradients. Inter-annual variability is dominated by ENSO with variations as large as 20% IWV in the tropics and the mid to high northern

latitudes in winter. ERA-Interim is shown to exhibit a slight moist bias in the extra-tropics and a slight dry bias in the tropics (both in the order of 0.5 to 1 kg m$^{-2}$) compared to GPS. ERA-Interim is also generally drier than MERRA-2 over the ocean and within the tropics. Differences in variability and trends are pointed out at a few GPS sites, which might be due to representativeness errors, for sites located in coastal regions and regions of complex topography, gaps and inhomogeneities in the GPS series, due to equipment changes, and potential inhomogeneities in the reanalyses, due to observing system changes. Trends in IWV and surface temperature in ERA-Interim and MERRA-2 are shown to be consistent, with positive IWV trends generally correlated with surface warming, but MERRA-2 presents a more general global moistening trend compared to ERA-Interim. Inconsistent trends are found between the two reanalyses over Antarctica and most of the southern hemisphere, and over central and northern Africa. The uncertainty in current reanalyses remains quite high in these regions where few in-situ observations are available and the spread between models is generally important. Interannual and decadal variations in IWV are also shown to be strongly linked with variations in the atmospheric circulation, especially in arid regions, such as North Africa and Western Australia, which add uncertainty in the trend estimates over the shorter period. In these regions, the Clausius-Clapeyron scaling ratio is found not to be a good humidity proxy for interannual variability and decadal trends.

**1 Introduction**

Water vapour is a key component of the Earth's atmosphere and plays a key role in the planet's energy balance. It is the major greenhouse gas in the atmosphere and accounts for about 75 % of the total greenhouse effect globally (Kondratev, 1972). This is a global average, as the greenhouse effect of water vapour depends on the total amount of water vapour in the column which is spatially heterogeneous. At global scale, theThe total amount of water vapour is mainly controlled by temperature, closely following closely the Clausius-Clapeyron (C-C) equation (Held and Soden, 2006; Semenov and Bengtsson, 2002). According to C-C, a temperature increase in the lower troposphere of 1°C leads to an increase in the vertical profile of water vapour of 6 to 7% (globally). ItSchneider et al., 2010). Water vapour is thus an important part of the response of the climate system to external forcing, constituting a positive feedback in global warming (IPCC report).Held and Soden, 2006). However, at a regional scale, deviations from C-C law are observed and the strength of the feedback can vary, also because the radiative effect of absorption by water vapour is sensitive to the fractional change in water vapour, not to the absolute change. (O'Gorman and Muller, 2010).

Integrated water vapour (IWV) has also been shown to be an important parameter in precipitation onset. Neelin et al. (2009), Holloway and Neelin (2009) and Sahany et al. (2012) concluded that IWV is a better proxy than surface humidity, sea surface temperature or integrated column saturation for transition to deep convection in the Tropics because at higher temperatures, deep convection occurs at lower relative humidity rates. Entrainment processes actually play a substantial role in the onset of deep convection, which is thus sensitive to the lower tropospheric humidity. However, the relationship between IWV and precipitation is a two-way interaction since convection also moistens the free troposphere (the upper troposphere mainly). This

relationship is a key issue for models in a warming climate. Bastin et al. (submitted) used it to evaluate simulations performed in the framework of MED-CORDEX (Ruti et al. 2015) over the Mediterranean area and concluded that models with "too light, too often" precipitation could be better constrained by IWV-temperature relationship. Therefore, seasonal, interannual and temperature-IWV variability should be studied.

5   At these (seasonal and interannual) scales, climate variations also result from natural variability. The spatial structure of climateglobal IWV variability at seasonal and longer time scales evidences patterns that result from interactions between the atmospheric circulation and the land and ocean surfaces. These include the El Niñõ Southern Oscillation (ENSO) and the North Atlantic Oscillation (NAO). ENSO is a quasi-periodical oscillation in winds and sea surface temperature over the tropical eastern Pacific Ocean, which impacts the weather and climate worldwide. The NAO fluctuates at time scales that go from days

10  to decades, and has an impact over the regional climate variability in Europe, particularly in winter. and are dominated by El Niñõ Southern Oscillation (ENSO) (Trenberth et al., 2005).

Although El Niño events are associated with increasing temperatures in the eastern and central Pacific with impact on the global weather and climate, it is not well known if global warming will lead to more frequent or intense El Niño events (ColinsCollins et al., 2010). Conversely, although aAnother strong cause of variability in the northern hemisphere is the North

15  Atlantic Oscillation (NAO). A high positive NAO (when the gradient between the Icelandic Low and the Azores High is enhanced) is associated with warmer winters in the Eurasian landmass, due to the stronger westerly and south-westerly airflow that brings in warmer maritime air. However, it is not clear how the phase or intensity of NAO has been, or will be, affected by climate change (Visbeck et al., 2001).

All these parameters, and the fact that the time of residence of water vapour in the atmosphere is short, make IWV a highly

20  variable component and its study in terms of variability and trends areis rather challenging. Several studies have reported onSherwood et al., 2010, compared the long-term IWV trends obtained fromreported in several studies using different IWV datasets. Although there appears to be a global positive trend in the overall IWV data, which is consistent with a global warming trend, it is difficult to compare results from different studies, as they refer to different data sources, time periods and different sites and spatial coverage. Trenberth et al. (2005) found major problems in the means, variability, and trends from 1988 to

25  2001 for the National Centers for Environmental Prediction (NCEP) reanalyses 1 and 2, and for the 40-year European Centre for Medium-Range Weather Forecasts (ECMWF) reanalysis (ERA-40) over the oceans. The reanalyses showed reasonable results over land where they are constrained by radiosonde observations. Only the reprocessed IWV data from the special sensor microwave imager (SSM/I) appeared to be realistic in terms of means, variability, and trends over the oceans. Their work points to two important issues. First, the reanalyses generally lack assimilation of water vapour information and suffer

30  from model biases and, in the case of ERA40, problems in bias corrections with new satellites (namely after major volcano eruptions). Second, they highlight the need for the reprocessing of data, and point to the shortcomings in reanalyses due to the changing observing system. The bias correction of new satellite radiances in the ECMWF reanalysis system has recently been improved using a variational bias-correction scheme, including the detection of instrument calibration errors and long term drifts, as well as volcano eruptions (Dee and Uppala, 2009). Reduction of model biases and enhanced assimilation capabilities

of satellite data (e.g. rain-affected radiances) have generally improved the water cycle in modern reanalyses. Reanalyses data agree thus generally well at representing the short-term variability (e.g. ENSO) but their ability for detecting climate trends is still debated (Dessler and Davis, 2010; Thorne and Vose, 2010; Trenberth et al., 2011; Robertson et al., 2014; Schröder et al., 2016).

5 ~~There are several sources of IWV data, including different types of measurements (using instruments such as radiosondes, Global Positioning System (GPS), and satellites), atmospheric reanalyses, and climate models. For studies at the scale of climate change (seasonal, annual and interannual scales), the data must be available long term, must be consistent and preferably homogeneous over time so as to not include (or reduce) non-climatic influences such as drifts and abrupt changes. Indeed, differences in trend estimates exist between the existing IWV products, due to a lack of homogenized datasets (Ning~~
10

~~In this paper, GPS-derived IWV data is used. GPS has the advantage of having a growing global network of mostly land-based stations (see Fig. 1), which gather data under most weather conditions, at a high temporal resolution, and with a continuous temporal coverage that dates back to 1995. The GPS data has been consistently reprocessed to ensure a homogeneous retrieval of IWV. However, it can still be affected by inhomogeneities, due to (for instance) changes in GPS equipment and algorithm~~
15  In this study we will focus on analysing the mean distributions, inter-annual variability, and decadal trends from two recent reanalyses, ECMWF reanalysis ERA-Interim (Dee et al., 2011), referred to as ERAI, and the 2nd Modern-Era Retrospective analysis for Research and Applications, MERRA-2 (Gelaro et al., 2017). MERRA-2 benefits from recent developments in NASA's Goddard Earth Observing System (GEOS) model suite intended to address the impact of the changes in observing system (Gelaro et al., 2017). As a result, atmospheric water balance and variability in MERRA-2
20 are more realistic, though variations of IWV with temperature are weaker in the main satellite data reanalyses (namely ERA-Interim and MERRA-2) compared to microwave satellite observations over the oceans (Bosilovich et al., 2017). Here, the IWV contents of the two reanalyses are intercompared and compared to a global, homogeneously reprocessed Global Positioning System (GPS) dataset over ocean and land. The ground-based GPS observations are independent from the reanalyses as they are so far not assimilated and constitute a valuable validation data for atmospheric reanalyses (Bock et al.,
25 2007) and satellite data (Mears et al., 2015). A critical assessment of the homogeneity of GPS dataset itself is provided throughout the study as previous work detected small offsets associated to GPS equipment changes (Vey et al., 2009; Ning et al., 2016).

Furthermore, to add new insights in both the evaluation of ERA-interim and MERRA-2 reanalyses

30 ~~(ECMWF) reanalysis, ERA Interim , which provides a multivariate, spatially complete, and coherent record of the global atmospheric circulation (Dee et al., 2011), thus a priori a good complement of the more sparse GPS dataset. ERA-interim has been chosen because it is quite recent, is used to drive/force a lot of regional climate simulations, and is often used to assess climate models, which have difficulty in accurately representing the water vapour distribution in the atmosphere, and in describing its greenhouse effect, especially at the regional level.~~

~~Nevertheless, with regards to IWV in particular, the homogeneity of the reanalysis data has also been called into question by several studies (Bengtsson et al., 2004; Dessler and Davis, 2010; Schröeder et al., 2016). Schröeder et al. (2016) compared the IWV from three reanalyses (ERA-Interim; the Modern Era Retrospective analysis for Research and Applications, MERRA; and the Climate Forecast System Reanalysis, CFSR) with three satellite-based IWV data records (Hamburg Ocean Atmosphere~~
5 ~~Parameters and Fluxes from Satellite Data, HOAPS; Remote Sensing Systems, REMSS; NASA Water Vapour Project MEaSUREs program, NVAP-M), for the 1988-2008 period. They analysed anomaly differences relative to HOAPS for averages over the global ice-free oceans and found break-points, which mostly coincided with changes in the observing system. In addition, their trend estimates show poor consensus in the central Africa, the Sahara, and South America regions.~~

10 ~~set and ERA-Interim reanalysis with special focus on trends and interannual variability. Trenberth et al. (2005) analysed trends and variability of IWV over the period 1988-2001 and used radiosonde data from Ross and Elliott (1996, 2001) over land to evaluate the ECMWF reanalysis ERA-40 and National Centers for Environmental Prediction (NCEP) reanalyses. However, radiosondes were shown to be in less agreement with ERA-Interim than GPS and DORIS IWV (Bock et al., 2014). In this study, to add new insights in both the evaluation of ERA-interim reanalysis~~ and in the understanding of IWV trends and
15 variability, we separate the analysis into seasons, and consider trends and interannual variability of seasons. This analysis by seasons is rarely provided in other studies, although it helps to better identify regions with higher uncertainty and to understand the physical processes involved in different seasons (e.g.
20 the dynamical component which transports moisture strongly differs between winter and summer). Trenberth et al. (2011) separated January and July in their analysis of the representation of water and energy budget in ERA-I and MERRA and showed the importance to study the seasons separately. Compared to this study, we added the analysis of the GPS dataset and use the new version of MERRA.

25 This paper is organized as follows. Section 2 details the datasets and methods used. Section 3 reports on the means and variability found in the GPS and reanalyses data, for the 1995-2010 period. Section 4 focuses on the monthly and seasonal trends in GPS and ERA-Interim, for 1995-2010. In Section 5 we confront results of ERA-Interim and GPS to MERRA-2. In this section, the comparison between ERA-Interim and MERRA-2 was also extended to the 1980-2016 period and focused on two regions of intense trends: western Australia and north Africa/eastern Sahel. Section 6
30 summarizes and concludes the paper.

**2 Datasets and methods**

**2.1 Reanalysis data**

Reanalysis data from the ECMWF, ERA-Interim (Dee et al., 2011), and NASA, MERRA-2 (Gelaro et al., 2017), were extracted for the 1980-2016 period, on regular latitude-longitude grids, at their highest horizontal resolution (0.75° x 0.75° for ERA-Interim and 0.625° longitude x 0.5° latitude for MERRA-2). In this work, the two-dimensional (2D) distribution of IWV is investigated with reanalysis fields and with point observations from 104 GPS stations of the International GNSS (Global Navigation Satellite System) Service (IGS) network (Fig. 1). Because GPS heights and model surface heights in the reanalyses are not perfectly matched (see the GPS coordinates and ERA-Interim heights in the supplement Table S1), the IWV estimates were adjusted for the height differencedifferences using two different methods. In the 2D maps (e.g. Fig. 2), the monthly mean GPS IWV estimates were height corrected to match the nearest ERA-Interimreanalysis grid point, while for the computation of IWV differences (e.g. Fig. 34), a more elaborate interpolation method was used (described below). For the monthly mean IWV correction, specific humidity from the ERA-Interim pressure level data waswere integrated over the layer of atmosphere bounded by the model's surface height and the height of the GPS station. The ERA-Interim pressure level data containscontain a total of 37 levels between 1000 and 1 hPa, andamong which 27 levels lie between 1000 and 100 hPa. This ensures a good vertical sampling of the troposphere where most of the water vapour is located. Note that for the sake of consistency, the same pressure-level data (i.e. ERA-Interim) are used for correcting the MERRA-2 monthly mean IWV data shown in the maps. TheIn the case of ERA-Interim, the height differences between GPS stations and nearest model grid points range from -1457 m (at the SANT (Santiago, Chile) station) to +3167 m (at the MKEA (Mauna Kea, Hawaii) station). The negative height difference means GPS height is below the model surface. The mean IWV corrections for these two stations amount to -3.4 kg.m$^{-2}$ and 21.7 kg.m$^{-2}$, respectively. Globally, 102 out of the 104 stations have a correction smaller than 7.7 kg.m$^{-2}$ in absolute value and the 
[revised manuscript text omitted]

25    ~~occurs at mid and high latitudes due to the cooler oceans and land surface. Lower IWV observed at these latitudes is also explained by the limited moisture-holding capacity of the relatively cooler tropospheric air (Trenberth et al., 2007; Lorentz and DeWeaver, 2007). The rapid decrease of water vapour saturation pressure with altitude as predicted by Clausius-Clapeyron equation also explains the lower IWV contents over elevated land surfaces. Minimal IWV values are found over major mountain ranges (e.g. the Himalayas and the Andes cordillera). The lack of surface water is another strong limitation for~~
30

[revised manuscript text omitted]

~~Next, the GPS data and both reanalyses were used to study the trends in IWV over the period 1995-2010. Strong annual trends were found in both reanalyses. First, in ERA-Interim, significant moistening trends were observed over most of the tropical oceans and over the Arctic, while significant drying was observed in south-tropical eastern Pacific region, west of the United States and generally south of 60°S. Over land, significant positive trends were observed in northern South America, Central Africa, and Indonesia, over northern North America, Greenland, most of Europe and Siberia. Significant negative trends over land were observed over North Africa, Australia, Antarctica, central Asia, and most of the USA. These trends were compared with GPS and were found to be in general good agreement, but with opposite sign trend at some sites. Discrepancies at most of these sites were found to be due to gaps in the GPS time series (when time-matched series are compared, the agreement is improved) and discontinuities (some of which explained by reported GPS equipment changes), but drifts in the ERA-Interim reanalysis are also plausible.~~

~~The seasonal trends (for DJF and JJA) presented stronger absolute and relative trends. In some regions, trends can have opposite signs in winter and summer, which emphasizes the role of atmospheric circulation in IWV trends. The comparison of ERA-Interim and GPS seasonal trends is consistent with the annual trends. However, the differences are generally of larger magnitudes and a few more sites show trends with opposite signs. This is mainly due to the enhanced impact of time gaps for the short seasonal time series.~~

~~The results for MERRA-2 appear to be different from ERA-Interim over several parts of the globe, in particular over Indonesia and Indian Ocean, central Africa, Western (coastal) and Northern Africa, Central Asia and Antarctica (where there appears to be some uncertainty in all datasets). The trends for 1995-2010 were also compared with longer-term trends, for the 1980-2016 period. For both long and short term periods, important differences were found between ERA-Interim and MERRA2 over Africa and Antarctica. These areas correspond to areas with very few observations available for data assimilation, which suggest that physical processes might not be well represented. A more detailed investigation of the dynamics over the eastern Sahel in north Africa and western Australia was presented. We considered the wind at 925 hPa to assess the role of dynamics in the IWV and temperature trends and variability. Anomalies in the wind speed and direction were associated with differences in IWV anomalies, and differences in the winds for both reanalyses were found to enhance the differences in IWV trends.~~

[revised manuscript text omitted]

Bastin, S., Bock, O., Chiriaco, M., Drobinski, P.,Roehrig, R., Ahrens, B., Gallardo, C., Dominguez Alonso, M., and Li, L.: Evaluating the impact of humidity bias on light precipitation estimates in Med-CORDEX/Hymex simulations using long term GPS network and ground-based datasets, Submitted to Atmos. Chem. Phys. special issue.

Bauer, P.: 4D-Var assimilation of MERIS total column water-vapour retrievals over land, Q. J. Roy. Meteor. Soc., 135: 1852-1862, doi:10.1002/qj.509, 2009.

Bengtsson, L., Hagemann, S., and Hodges, K. I.: Can climate trends be calculated from reanalysis data?, J. Geophys. Res. Atmos., 109(D11), doi: 10.1029/2004JD004536, 2004.

[revised manuscript text omitted]

O'Gorman, P. A. and Muller, C. J.: How closely do changes in surface and column water vapor follow Clausius–Clapeyron scaling in climate-change simulations? Environ. Res. Lett. 5, 025207, 2010.

Power, S., Tseitkin, F., Torok, S., Lavery, B., and McAvaney, B.: Australian temperature, Australian rainfall, and the Southern Oscillation, 1910–1996: Coherent variability and recent changes, Aust. Meteorol. Mag., 47, 85–101, 1998.

Robertson, F.R., Bosilovich, M.G., Roberts, J.B., Reichle, R.H., Adler, R., Ricciardulli, L., Berg, W., and Huffman, G.J.: Consistency of Estimated Global Water Cycle Variations over the Satellite Era. J. Ross, R. J. and Elliott, W. P.: Tropospheric water vapour climatology and trends over North America: 1973-93, J. Climate, 9(12), 3561-3574, doi:10.1175/1520-0442(1996)009<3561:TWVCAT>2.0.CO;2, 1996.

Ross, R. J. and Elliott, W. P.: Radiosonde-based Northern Hemisphere tropospheric water vapour trends, J. Climate, 14(7), 1602-1612, doi:10.1175/1520-0442(2001)014<1602:RBNHTW>2.0.CO;2, 2001. d Climate, 27, 6135–6154, https://doi.org/10.1175/JCLI-D-13-00384.1, 2014.

Rousseeuw, P. J. and Leroy, A. M.: Robust Regression and Outlier Detection, Wiley Series in Probability and Mathematical Statistics, 516, Wiley, p. 67, 2003. ISBN 978-0-471-48855-2.

Ruti, P.M., Somot, S., Giorgi, F., Dubois, C., Flaounas, E., Obermann, A., Dell'Aquila, A., Pisacane, G., Harzallah, A., Lombardi, E., and Ahrens, B.: MED-CORDEX initiative for Mediterranean climate studies, B. Am. Meteorol. Soc., 97(7), 1187-1208, doi: 10.1175/BAMS-D-14-00176.1, 2016.

Sahany, S., Neelin, J. D., Hales, K., and Neale, R. B.: Temperature–moisture dependence of the deep convective transition as a constraint on entrainment in climate models, J. Atmos. Sci., 69(4), 1340-1358, doi: 10.1175/JAS-D-11-0164.1, 2012.

Schröder, M., Lockhoff, M., Forsythe, J. M., Cronk, H. Q., Vonder Haar, T. H., and Bennartz, R.: The GEWEX Water Vapour Assessment: Results from Intercomparison, Trend, and Homogeneity Analysis of Total Column Water Vapour, J. Appl. Meteorol. Clim., 55(7), 1633-1649, doi: 10.1175/JAMC-D-15-0304.1, 2016.

Semenov, V. and Bengtsson, L.: Secular trends in daily precipitation characteristics: greenhouse gas simulation with a coupled AOGCM, Clim. Dynam., 19(2), 123-140, doi: 10.1007/s00382-001-0218-4, 2002. Schneider, T., O'Gorman, P. A., and Levine, X. J.: Water vapor and the dynamics of climate changes, Rev. Geophys., 48, RG3001, doi:10.1029/2009RG000302, 2010.

Sen, P. K.: Estimates of the regression coefficient based on Kendall's tau, J. Am. Stat. Assoc., 63 (324): 1379–1389, JSTOR 2285891, MR 0258201, doi:10.2307/2285891, 1968.

Sherwood, S. C., Roca, R., Weckwerth, T. M., and Andronova, N. G.: Tropospheric water vapor, convection, and climate, Rev. Geophys., 48, RG2001, doi:10.1029/2009RG000301, 2010.

Theil, H.: A rank-invariant method of linear and polynomial regression analysis, I, II, III, Nederl. Akad. Wetensch., Proc., 53: 386–392, 521–525, 1397–1412, MR 0036489, doi: 10.1007/978-94-011-2546-8_20, 1950.

Thorne, P.W. and Vose, R.S.: Reanalyses Suitable for Characterizing Long-Term Trends. Bull. Amer. Meteor. Soc., 91, 353–362, https://doi.org/10.1175/2009BAMS2858.1, 2010.

Trenberth, K. E., Fasullo, J., and Smith, L.: Trends and variability in column-integrated atmospheric water vapour, Clim. Dynam., 24(7-8), 741-758, doi: 10.1007/s00382-005-0017-4, 2005.

Trenberth, K. E., Smith, L., Qian, T., Dai, A., and Fasullo, J.: Estimates of the global water budget and its annual cycle using observational and model data, J. Hydrometeorol., 8(4), 758-769, doi: 10.1175/JHM600.1, 2007.

Trenberth, K.E., Fasullo, J.T., and Mackaro, J.: Atmospheric Moisture Transports from Ocean to Land and Global Energy Flows in Reanalyses. J. Climate, 24, 4907–4924, https://doi.org/10.1175/2011JCLI4171.1, 2011.

Vey, S., Dietrich, R., Fritsche, M., Rülke, A., Steigenberger, P., and Rothacher, M.: On the homogeneity and interpretation of precipitable water time series derived from global GPS observations, J. Geophys. Res. Atmos., 114, D10101, doi: 10.1029/2008JD010415, 2009.

Visbeck, M. H., Hurrell, J. W., Polvani, L., and Cullen, H. M.: The North Atlantic Oscillation: past, present, and future. Proceedings of the National Academy of Sciences, 98(23), 12876-12877, https://doi.org/10.1073/pnas.231391598, 2001.

Wagner, T., Beirle, S., Grzegorski, M., and Platt, U.: Global trends (1996–2003) of total column precipitable water observed by Global Ozone Monitoring Experiment (GOME) on ERS-2 and their relation to near-surface temperature, J. Geophys. Res. Atmos., 111, D12102, doi: 10.1029/2005JD006523, 2006.

Wang, J., Zhang, L., and Dai A.: Global estimates of water-vapour-weighted mean temperature of the atmosphere for GPS applications, J. Geophys. Res., 110, D21101, doi:10.1029/2005JD006215, 2005.

Wang, J., and Zhang, L.: Systematic errors in global radiosonde precipitable water data from comparisons with ground-based GPS measurements. J. Climate, 21(10), 2218-2238, https://doi.org/10.1175/2007JCLI1944.1, 2008.

Wang, J., Dai, A., and Mears, C.: Global Water Vapour Trend from 1988 to 2011 and Its Diurnal Asymmetry Based on GPS, Radiosonde, and Microwave Satellite Measurements, J. Climate, 29(14), 5205-5222, doi: 10.1175/JCLI-D-15-0485.1, 2016.

Weatherhead, E. C., Reinsel, G. C., Wardle, R. and Smith, I.: Modeled response of the Australian monsoon to changes in land

[revised manuscript text omitted]

---

## Author Response (AR2)

The co-editor and referee's comments are presented followed by our responses in blue.

**Co-Editor :**

Dear Authors,

collecting the comments of the three different reviewers, it still remains a different task to give a recommendation for this manuscript. Overall, the manuscript provides important and interesting material, with which there is nothing wrong, but it provides just too much information, presented in a very descriptive way ("a work summary, rather than a synthesized scientific paper", as quoted by a reviewer). Therefore, I'm afraid that a reader will loose its interest when struggling through e.g. all the descriptions of the differences seen in the figures of global IWV means and trends. And this would really be a pity, given the workload put in the analysis. So, I follow the referee #1 that the manuscript is still too long and should be considerably shortened. Please follow the general comment of this referee regarding Sections 4 and 5. I will also give some specific indications on which parts could be significantly shortened later on. But furthermore, I would also go through your manuscript once more and ask yourself at every sentence the questions: "Is this sentence really necessary? Does it provide new information? Has the information provided not already been given at another location in the manuscript?"

Secondly, I also think that you did not take into account well enough the comment raised by Referee #2 (who also asked for a major revision, but unfortunately was unable to review the paper again during the summer break) about the focus of the paper. Yes, you rewrote parts of the introduction, shifting the focus to the IWV analysis based on the two (recent) reanalyses (page 2, line 33 - page 3, line 10), but this shift of focus is not consistently kept throughout the analysis. For instance, in Section 3, you should then primarily discuss the differences in the means and variability between the IWVs from the reanalyses, and in regions where you find discrepancies between both reanalyses, use the GPS dataset to understand and explain those discrepancies. Here, also the reference to the analysis done by Trenberth et al. 2011 is important. Now, in its present form, two third of the Section (pages 7 and 8) deals with GPS - ERA-Interim IWV differences, while the focus was said to lie on the reanalyses. These two pages should hence almost completely be dropped from the regular text, instead I would devote an entire appendix on GPS site specific findings (with a description (and explanation) of sites for which the means, standard deviation and trends deviate from these of the reanalyses). In this case, a reader with a GPS background will easily find his/her way to relevant information for his/her research. The same remark about the inconsistent focus applies to the trend description sections 4 and 5.1. As your focus is on the comparison of the reanalyses, it is really illogical to start the description the trends with the comparison between GPS and ERA-Interim (Section 4). From my point of view, it seems more logical to start the analysis with the comparison of the ERA-Interim and MERRA-2 trends for the 1980-2016 time period, and then constrain to the 1995-2010 time period in which the GPS dataset can be used as external comparison (as outlined on page 3, lines 5-6). But here again, I would move GPS site specific findings about trend differences with ERA-Interim (and why not with MERRA-2) to the dedicated Appendix.

We thank the editor for the constructive comments that helped improve the manuscript. The paper was significantly shortened keeping in mind these comments, including a reorganization of the subsections. The focus was first put on a general intercomparison between GPS and reanalyses for the longest common period between the datasets (with more station-specific comparisons between ERA-Interim

and GPS moved to an Appendix, as suggested). And then followed by a more extensive comparison between the two reanalyses, in term of trends, for an extended period (largest common period between reanalyses). We believe this new organization of the paper, aside from making it shorter, will also make it easier to read.

Now, I will give some specific comments, in particular highlighting the parts of the paper that can seriously be shortened to my opinion.
* page 1, lines 12-13: in the abstract you mention the link with ENSO, while this is only a secondary outcome of your paper (as a matter of fact, you have never shown it). So drop it.

The lines have been dropped from the manuscript.
* page 2, lines 5-13: also not very relevant for your analysis here.

These lines have been removed
* page 3, lines1-4: move to section 2.1

This has been done.

* page 4, lines 7-10: might be moved to the GPS site-specific appendix

They have been moved to Appendix 2, as suggested.
* page 6: lines 14-19: Is this information (mentioned, not shown) really important for your paper?

The information has been removed.
* page 6, lines 24-26: how does this finding compare with other studies (e.g. Trenberth et al. 2011? Perhaps you should mention this at this location).

The Trenberth et al. (2011) paper does not deal with MERRA-2 data, although it was used a reference throughout the paper.
* page 7, lines 4-25: this information might be shifted to the proposed appendix

It has been moved to Appendix 2, as suggested.
* page 7, line 31 - page 8, line 28: this information might be shifted to the proposed appendix, possibly just provide a summary of a few sentences in this part of your regular paper.

This has been done.
* Sect 4: move after Section 5.1 and shift the bulk of the text (especially page 9, line 15 - page 10, line 15) to the proposed appendix.
* page 11, lines 14-22: please be aware of some duplication of text with sections 5.1 and 5.2
* Sect 5.1: I would start this section with the paragraph on page 13, lines 19-33 (but only for the long period), which is really well written, a really good summary without too much obsolete information.

* page 12, line 15 - page 13, line 12: these paragraphs could be considerably be shortened! You just have to mention where the ERA-Interim and MERRA-2 trends are opposite and what the trends of the GPS sites in those regions are.

Regarding the points above, the paper has been reorganized in large part, so that the text has been reduced overall, with less repetition. In this new version we start by comparing the three datasets (GPS, ERA-Interim and MERRA-2) for the longest common period (1995-2010), and then extend the trend analysis to 1980-2016 for the ERA-Interim vs. MERRA-2 comparison. When comparing the

datasets, we focus on the differences found between reanalyses and attempt to compare it with GPS.
* page 16, line 32 - page 17, line 11: this paragraph is very hard to follow, as it is a mixture of IWV trends, T trends, IWV-T correlations, both for the short and long period and also for the seasons! Could you not highlight the most important findings instead of mentioning them all?

This has been re-written :

The trend in T anomalies also stops at around 2008 (Fig. 12a). Before that period, the temperature anomaly is increasing significantly, despite strong month-to-month variability. However, there is low/ negative correlation between IWV and T anomalies when considering the monthly time series (Fig. 12a). In JJA, the trends are strong and go on after 2008 (Fig. 12c). The correlation of anomalies for JJA between both reanalyses is quite good, both for IWV (around r = 0.67 for the short period and r = 0.63 for the longer period) and T (around r = 0.69 for both periods), although their amplitudes and trends are quite different. MERRA-2 presents an overall moistening trend, while ERA-Interim shows a drying (Fig. 7g,h and 12c). Simultaneously, the temperature trends are both positive and significant, thus not explaining the IWV trends according to C-C

* page 18, lines 5-12: In the beginning of the summary, you state that your focus are the modern reanalyses (line 3). But, in the following lines, you give the results about the comparison between GPS and ERA-interim.

These have been re-written :
The means and variability in IWV in the reanalyses were inter-compared and compared to ground-based GPS data for the 1995-2010 period. ERA-Interim was shown to exhibit a slight moist bias in the extra-tropics (~ 0.5 kg m-2) and a slight dry bias in the tropics in relation to both GPS and MERRA-2, which is consistent with other studies (Trenberth et al., 2011). Inter-annual variability in ERA-Interim is highly consistent with GPS, and in good agreement with MERRA-2. Differences were pointed out between GPS and reanalyses at only a few stations, mostly located in coastal regions and regions of complex topography, where representativeness errors put a limit to the comparison of gridded reanalysis data and point observations.

**Anonymous Referee #1**

General Comments

The revised manuscript is 46 pages long. The previous version was 43 pages long and I gave the recommendation to shorten the manuscript and focus on what is new. ( I did ask for the Appendix added in the updated version, but it is 1.5 pages long, and could be made significantly shorter.)

The revised manuscript has now been significantly shortened.

I think this is still a drawback in Sections 4 and 5. There is no need to "walk trough" different regions repeating in words what is already presented in the figures. I would prefer if the authors could focus on issues (in terms of trends or mean values) where there are significant disagreements, given the uncertainties of the means and the trends.

Following this suggestion, sections 4 and 5 have been merged and reduced, focusing on the significant similarities and disagreements between reanalyses, with GPS being used as an independent comparison.

Specific comments

page 25, Table 1: Does it make sense to divide the seasons into DFJ and JJA when you calculate mean values using data from sites spread over both the northern and the southern hemisphere?
The statistics in Table1 have now been separated by hemisphere.

Technical Corrections
The technical corrections have been applied.
page 1, line 12: 20% --> 20 %

page 3, lines 7 and 8:

page 7, lines 17 and 20: "space" between values and units is missing

page 17, line 12: 925hPa --> 925 hPa

page 18, line 9: 20% --> 20 %

page 46: Figure A3 --> Figure A1 ?

**Anonymous Referee #3**

The paper consists of an analysis of the trends and variability of integrated water vapor mainly in the period 1995-2010, mainly analyzing ERA-Interim product in comparison with reliable local data (GPS) and another reanalysis, MERRA-2. The manuscript is very well written, and very thorough both in all of its sections. Maybe it can feel dense because of its length, and a shortening could make it easier to read. The statistical treatment is very good to my knowledge. The topic is very interesting and of paramount importance.

I only have a few questions that I would like the authors to address:

1. Authors claim that the GPS time-series have inhomogeneities. I would like to know if they have considered any kind of statistical treatment to eliminate these inhomogeneities (when it is clear that they are due to GPS and not to ERA-Interim), to see if after correction they match ERA-interim trends.
Not yet. At this point the method of eliminating inhomogeneities uses ERA-Interim data, so that ERA-Interim could not be used as comparison. This is an ongoing reasearch topic.

2. Page 2, line 6: IPCC report is mentioned, but not the year, and it does not appear in references section.

The IPCC is no longer cited/ mentioned.

3. Page 2, lines 32-33: could the authors give references to these "several studies"?

Yes, a more thorough overview of the studies is now given (i.e. : Sherwood et al. (2010), Trenberth et al. (2005) among others)

4. Page 5, line 1: could the authors provide a references to NASA JPL's product?

Basic details on the operational GPS data processing procedure are described by Byun and Bar-Server (2009).

http://igscb.jpl.nasa.gov/pipermail/igsmail/2010/007488.html

5. Page 6, line 4: author claim that interoplation is done with the "4 surrounding grid points". Does this mean that they use the closest north, east, south, west grid points but not the grid point coincident with the GPS location?

GPS locations do not coincide with the reanalyses grid points, so we use the 4 gridpoints on the summit of the square enclosing the GPS station. This is a standard method for bi-linear interpolation.

6. Page 8, line 24-25: Since the test is not very efficient because of the low number of points, have authors considered using, for instance, monthly averages, in order to have more datapoints? Of course the time-series should be detrended first.

We didn't consider this option. It would give more data points but they would be noisier, so it is probable that their significance would not be higher.

7. Page 9, last line: Authors claim that "to be physically explained, such trends would imply a significant change in the regional and global water cycle". Could this be referenced, or explained?

The trend in ERA-Interim is -17% per decade, so -26% over the 15 years of the study. Such large trend has never been observed anywhere and is hard to explain.

8. I would suggest that Table S3 have significant trends in bold, or marked in any way to improve readability. But this is up to the authors to decide.

We provide the p-values so the reader can chose which confidence level he/she considers the most relevant.

9. Regarding lines 11-12 in page 15, I would like to know what other variables have been considered for "temperature proxy" and why T2m was chosen.

Actually, no other variables were tested, T2m was chosen from the beginning. In order to see the link between water vapour trends and surface temperature trends, especially in a context of climate change. This is following studies by, for instance, Trenberth et al. (2005) and Wang et al. (2016) that also looked at this relationship (cited in the paper).

10. Page 16, line 4: CMIP5 is firts used here, so the meaning of the acronym should be provided. A reference here would be appropiate as well.

The acronym is now defined in the manuscript.

11. Figures 12 and 15: The figures show the trend spatially averaged. Could the authors explain how the computation of spatial averages was performed?

The IWV anomaly values were averaged over the box, and a trend of the resulting time series was computed.

12. Table S3: p-values do not need so many decimal places, these should be rounded to 3 decimal places.

This has been done in the new version of the supplement.

[revised manuscript text omitted]

[Figure]

**Figure 1: Map showing the 104 GPS stations used in this study. The stations discussed in the text and the Annex are identified by their 4-character ID.**

[Figure]

[Figure]

**Figure 2: (a) Mean IWV for DJF 1995-2010 from ERA-Interim (shading) and GPS (filled circles), (b) same as (a) for JJA.**

[Figure]

[Figure]

**Figure 2 (continued): (c) Relative variability in % (standard deviation of the IWV series divided by its mean) for DJF 1995-2010, (d) Same as (c) for JJA.**

[Figure]

**Figure 3: (a) Difference of mean IWV estimates (MERRA-2 minus ERA-Interim) for DJF 1995-2010. The global mean difference is 0.35 kg.m-2 (0.94 %) and the standard deviation of the difference is 0.32 kg.m-2 (1.56 %). (b) Same as (a) for JJA. The global mean difference is 0.56 kg.m-2 (1.66 %) and the standard deviation of the difference is 0.37 kg.m-2 (1.67 %). (c) Difference of relative variability estimates (MERRA-2 variability minus ERA-Interim variability) for DJF 1995-2010. The global mean difference is -0.01 % and the standard deviation of the difference is 0.26 %. (d) same as (c) for JJA. The global mean difference is 0.28 % and the standard deviation of the difference is 0.42 %.**

[Figure]

[Figure]

**Figure 4: (a, b) Difference of mean IWV estimates (ERA-Interim minus GPS) for DJF and JJA 1995-2010; (c, d) same as (a, b) for MERRA-2 minus GPS. The monthly data were time-matched before seasonal means were computed.**

[Figure]

[Figure]

[revised manuscript text omitted]

---

## Author Response (AR3)

The co-editor's comments are presented followed by our responses in blue.

Co-Editor Decision: Publish subject to technical corrections (19 Oct 2018) by Roeland Van Malderen Comments to the Author:

I want to congratulate you with the large improvements made to the paper. To my opinion, the paper is now written much fluently, the focus clearer and it also better incorporates the remarks/suggestions given in the 5 different reviews. I only have some small suggestions for corrections/adjustments:

We thank the editor for the constructive comments that helped improve the manuscript, and the technical corrections have all been implemented. The rewritten passages are highlighted in more detail below. In addition, statistics for the trend differences between GPS and reanalyses were added to Table 1 for completeness. They were briefly commented on in the main text:

10

Statistics of differences in the trend estimates between the reanalyses and GPS are given in Table 1. The median differences are small (below  $\pm 1\%$  per decade) for both reanalyses and both hemispheres. The interquartile range of differences vary depending on the season and hemisphere but they are similar for both reanalyses. The monthly trends agree quite well in the northern hemisphere ( $\sim 2.6$  % per decade vs.  $\sim 4$  % per decade in the southern hemisphere), while the seasonal trends have larger errors in the winter hemispheres ( $\sim 7\%$  per decade).

15

\* on page 2, lines 3-4: this sentence has some issues. First, specify to what "This" refers to. Secondly, it's an evidence that IWV is highly variable due to the short residence time of water vapour in the atmosphere: you make two times the same statement here. Rewrite this sentence please.

**The sentence has been rewritten to: 20**

In addition, the short residence time of water vapour in the atmosphere makes its study in terms of variability and trends rather challenging.

\* page 2, line 29: a "valuable validation dataset" instead of "a valuable validation data"

- \* page 3, line 12: NASA is used here for the first time, so spell out (although everyone knows NASA).
- \* page 6, line 3: "zero" instead of "null"?
- \* page 6, line 25: "their values do not" instead of "their values don't"
- \* page 7, line 22: "Significant trend differences" instead of "Significant differences"
- \* page 7, line 28: spell out NOAA and SST
- 30 \* page 7, line 26 to page 8, line 11: it might be a good idea to mark the Maritime Continent and North Africa with boxes on the discussed plots.

**This has been done.**

- \* page 8, line 7: "extends southward" instead of "extents southward"
- \* page 8, line 8: "see also further on Fig. 8" 35
  - \* page 8, lines 21-22: You might change into "However, the amount of assimilated data over the Antarctica's ice

sheet may differ between the reanalyses, which is actually not documented."

\* page 8, lines 25-26: "A striking feature seen in both reanalyses is their relatively larger magnitude compared to the monthly trends." You might comment here if this might be related to the fact that for the seasonal trends, you calculate trends based on one value a year, and for the monthly trends, one value every month is used for the trend calculation.

This has been added as :

A striking feature seen in both reanalyses is their relatively larger magnitude compared to the monthly trends (Fig. 5), which could be due to the fact the monthly trends use one value per month, while the seasonal trends use only one value per year.

10

5

\* page 9, line 13: Trenberth et al. (2005) "suggested" instead of "suggests"

\* page 9, line 28- page 10, line 5. I got the impression that this discussion only takes ERA-Interim into account, not MERRA-2: you only refer to the ERA-interim subfigures of Fig 8 and at the beginning of the page 10, you note that "For JJA, all continental areas show a significant warming, with the exception of parts of Antarctica,

- 15 and a small area over northern Australia, where a cooling is also displayed, albeit not significant", a statement which is not true for MERRA-2. Please be more general in this paragraph. But also not too general, because at page 9, lines 30-31, you write: "Over land, all areas show an increase in temperature, except the high latitudes of the southern hemisphere". It should be noted that also over large parts of the Asian continent, the temperature decreased.
- 20 This has been rewritten to:

Figure 8 shows the seasonal IWV trends and temperature trends. In general, it is seen that over the oceans, the temperature trends have generally the same sign as the IWV trends (but opposite colours), as expected by Clausius-Clapeyron (C-C) theory, despite some small-scale differences. Over land, most areas show an increase in temperature, except the high latitudes of the southern hemisphere and large parts of the Asian continent. This

- 25 means that, except over Antarctica and parts of Asia, the drying observed in the afore-mentioned areas does not follow C-C theory. When we consider each season more closely, some areas indicate a cooling (Figs. 8a, b, e, f) consistent with a drying (Figs. 8c, d, g, h). This is observed over Antarctica (especially in ERA-Interim) and over Central Asia in DJF (especially in MERRA-2). For JJA, most continental areas show a significant warming, with the exception of parts of Antarctica, a small area over northern Australia, and regions in Central Asia,
- 30 where a cooling is also displayed. Thus the C-C scaling ratio is not a good proxy for humidity when considering seasonal and regional variabilities and trends due to the important role of dynamics which allow the advection of dry or wet air masses (e.g. over USA, South America, eastern Sahel, and South Africa in JJA).

\* page 10, line 30: The different IWV trend estimates between the two periods "are" instead of "is"

35 \* page 11, line 11: "their trends over and around Australia present different patterns (Fig 10)". Please specify which field and DJF/full time series.

This has been changed to:

Note that although the climatological means of zonal and meridional wind components are similar between ERA-Interim and MERRA-2, their trends over and around Australia present different patterns, especially in DJF (Fig.10e-h), likely explaining the different IWV trends between both reanalyses.

- 5 \* page 14, line 6: add "(" before see Appendix B).
  - \* page 14, line 27: "below" instead of "under"
  - \* page 15, line 1: add "with respect to the Theil-Sen method" at the end of the sentence.
  - \* page 15, line 11: "do not agree" instead of "don't agree"
  - \* page 15, line 18: "located above 1000 hPa" (or superior to) instead of "located below 1000 hPa".
- 10 In order to avoid misunderstandings, this has been changed to :

The reanalysis data are only extrapolated for stations located at pressure values over 1000 hPa (the lowest pressure level).

\* page 17, line 15: "coincides" instead of "coincide"

15 \* page 17, line 18: "do not seem" instead of "don't seem"

[revised manuscript text omitted]

Table 1: Statistics (median +/- interquartile range) for the differences (ERA-Interim minus GPS and MERRA-2 minus GPS) in mean and relative standard deviation of IWV at the 104 stations, divided by season and hemisphere.

25